# Quantum groups as global symmetries II. Coulomb gas construction

Barak Gabai[1], Victor Gorbenko[1], Jiaxin Qiao[1], Bernardo Zan[2] and Aleksandr Zhabin[1]

**1** Laboratory for Theoretical Fundamental Physics, Institute of Physics,
École Polytechnique Fédérale de Lausanne, Switzerland
**2** Department of Applied Mathematics and Theoretical Physics,
University of Cambridge, CB3 0WA, UK

## Abstract

We study a conformal field theory that arises in the infinite-volume limit of a spin chain with $U_q(sl_2)$ global symmetry. Most operators in the theory are defect-ending operators which allows $U_q(sl_2)$ symmetry transformations to act on them in a consistent way. We use Coulomb gas techniques to construct correlation functions and compute all OPE coefficients of the model in closed form, as well as to prove that the properties imposed by the quantum group symmetry are indeed satisfied by the correlation functions. In particular, we treat the non-chiral operators present in the theory. Free boson realization elucidates the origin of the defects attached to the operators. We also comment on the role of quantum group in generalized minimal models.

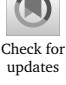

# 1 Introduction

In a companion paper [1], we considered an example of a CFT with the quantum group (QG) global symmetry which arises as a continuum limit of a certain spin chain that we called XXZ$_q$, a QG-invariant version of the periodic XXZ model discovered in [2]. Given the known spectrum of the theory, we solved for all the OPE coefficients using the conformal bootstrap method, stopping short of determining some signs. We also studied the consequences of QG symmetry in the continuum theory and confirmed some of the properties required by it, as we review below. In this paper, we use an alternative to the bootstrap method to give expressions for all the correlation functions in the XXZ$_q$ CFTs and make the appearance of the QG more manifest. We use a version of the Coulomb gas formalism [3,4], which was originally invented to compute the correlation functions of rational 2d CFTs with central charge $c < 1$, see also [5,6], as well as [7] for a pedagogical explanation.

The idea of the formalism is to start with a free boson field with special boundary conditions (a background charge $-2\alpha_0$ at infinity), build vertex operators, and construct correlation functions as linear combinations of integrated correlators of vertex operators. Operators that are being integrated are called screening charges. The key to the success of such a procedure is the fact that the functions constructed in this way satisfy differential equations that correlators of degenerate fields in minimal models satisfy (so-called BPZ equations [8]). We will call such objects kinematic functions. To get a correlation function of the theory one is interested in, one still needs to determine what is the proper linear combination of these kinematic functions by imposing monodromy conditions and crossing symmetry. For local CFTs, the monodromy condition is that the correlators are single-valued. In our version of the formalism, we inter-

pret certain integrated correlators as the correlators of operators of the theory with quantum group global symmetry. As explained in [1], operators are generically non-local and live at the end-points of defect lines; consequently, the monodromy conditions we need to impose are also different. The CFT we study has a local stress tensor and Virasoro symmetry and is local in this sense.

The idea to relate Coulomb gas formalism and quantum groups is not new. It was proposed originally in [9] and used extensively in subsequent publications summarized in [10]. While the general framework is the same, there are some important modifications that we had to introduce in the formalism that we describe below. The main novelty is that we are computing correlation functions of a UV complete theory, meaning that it also has a lattice definition, and as such we do not have any ambiguity in the definition of its correlation functions. This allows us to clearly separate the physical QG that acts as a global internal symmetry of the model. We also identify QGs that are present in the model, but do not serve as actual symmetries: they do not act neither on the Hilbert space nor on the operators of the model. These additional QGs are sometimes referred to as "hidden quantum group symmetries" [11], also present in generalized minimal models. We explain what this means in section 4.3.

In addition to correlation functions, free boson realization allows us to construct (some of the) QG generators, as well as the defect lines on which operators of the theory live. These lines consist of a combination of the integrated screening charges and the topological defect lines coming from the free boson CFT. As a result, we have a construction of lines which is not what we would like to call a microscopic construction of the defect lines. The reason is that Coulomb gas formalism itself, despite its appearance, is not a fully microscopic (Lagrangian) description of the model, see [12] for a recent discussion on this point. Nevertheless, definition of lines through Coulomb gas integrals is much more explicit than that presented in [1] and allows us to check all the properties that we need.

## 2 Constraints from quantum group symmetry

Let us start with a brief review of the constraints on correlation functions imposed by quantum group global symmetry. For a more detailed analysis see a companion paper [1], where two-dimensional QFTs with quantum group global symmetry were discussed. In this paper, we restrict the discussion to CFTs and, for simplicity, consider only the quantum group $U_q(sl_2)$.

The quantum group $U_q(sl_2)$ can be viewed as a deformation of the universal enveloping algebra $U(sl_2)$ with some deformation parameter, $q \in \mathbb{C}$. Its generators are usually called $E, F, H$ and satisfy the following commutation relations

$$[E, F] = \frac{q^H - q^{-H}}{q - q^{-1}}, \qquad q^H E q^{-H} = q^2 E, \qquad q^H F q^{-H} = q^{-2} F. \tag{1}$$

In the limit $q \to 1$ one recovers the commutation relations of $sl_2$. Considering correlation functions of several operators bring about the need to act on a tensor product of several quantum group representations. The action of the generators on two representations is constructed by applying the coproduct

$$\Delta(E) = E \otimes 1 + q^{-H} \otimes E, \qquad \Delta(F) = F \otimes q^H + 1 \otimes F, \qquad \Delta\left(q^H\right) = q^H \otimes q^H. \tag{2}$$

The action on several representations is given by applying the coproduct several times. The choice of the coproduct as in (2) is very natural from the Coulomb gas point of view, see section 3.

The quantum group $U_q(sl_2)$ being a global internal symmetry means that: (a) $U_q(sl_2)$ generators commute with the space-time symmetry generators; (b) correlation functions satisfy

$U_q(sl_2)$ Ward identities when the operators transform linearly under the quantum group action. We consider only the operators transforming in finite-dimensional representations. Thus, the operators are labeled by integer or half-integer spin $\ell$ and a number $m \in \{-\ell, \ldots, \ell\}$, which parametrizes the states in $(2\ell + 1)$-dimensional representation. We denote the operators as $\mathcal{O}_{i,\ell_i,m_i}$, where we keep the index $i$ because generically there might be several quantum group multiplets of the same spin. Our convention for the action of $U_q(sl_2)$ generators on operators is:

$$H \cdot \mathcal{O}_{i,\ell_i,m_i} = 2m_i \, \mathcal{O}_{i,\ell_i,m_i}, \qquad E \cdot \mathcal{O}_{i,\ell_i,m_i} = e_{m_i}^{\ell_i} \mathcal{O}_{i,\ell_i,m_i+1}, \qquad F \cdot \mathcal{O}_{i,\ell_i,m_i} = f_{m_i}^{\ell_i} \mathcal{O}_{i,\ell_i,m_i-1}, \quad (3)$$

where the matrix elements $e_m^\ell$ and $f_m^\ell$ are taken to be

$$\begin{aligned} f_m^\ell &= q^{m-1} \sqrt{[\ell+m]_q [\ell-m+1]_q}, \\ e_m^\ell &= q^{-m} \sqrt{[\ell-m]_q [\ell+m+1]_q}, \end{aligned} \qquad (4)$$

where $[n]_q = \frac{q^n - q^{-n}}{q - q^{-1}}$ denotes the $q$-deformed number.

The constraints imposed by $U_q(sl_2)$ global symmetry manifest themselves in Ward identities that restrict the correlation functions. A full list of independent Ward identities can be derived from the action of just $H$ and $F$. First of all, non-zero correlation functions satisfy the $U(1)$ charge conservation, $\sum_i m_i = 0$. Second, using the coproduct (2) the following non-trivial Ward identities can be derived,

$$\sum_{i=1}^{n} q^{2(m_{i+1}+m_{i+2}+\ldots+m_n)} f_{m_i}^{\ell_i} \langle \mathcal{O}_{1,\ell_1,m_1}(x_1) \ldots \mathcal{O}_{i,\ell_i,m_i-1}(x_i) \ldots \mathcal{O}_{n,\ell_n,m_n}(x_n) \rangle = 0, \qquad (5)$$

which provide the relations between correlation functions of zero total $U(1)$ charge. The solution to Ward identities gives $U_q(sl_2)$ invariant tensors. As it is argued in [1], the operators whose correlation functions satisfy these constraints are generically non-local and are attached to topological lines. In this paper we focus on CFT correlation functions, the constraints that we need can be summarized as follows:

- CFT two-point functions in canonical normalization are given by

$$\langle \mathcal{O}_{i,\ell_i,m_i}(0) \mathcal{O}_{j,\ell_j,m_j}(z,\bar{z}) \rangle = \delta_{ij} \begin{bmatrix} \ell_i & \ell_j & 0 \\ m_i & m_j & 0 \end{bmatrix}_q \frac{1}{z^{2h_i} \bar{z}^{2\bar{h}_i}}, \qquad (6)$$

  where $[\ldots]_q$ is the quantum Clebsch-Gordan coefficient and $h_i$, $\bar{h}_i$ are the conformal dimensions.

- The OPE between two operators in canonical normalization is constrained by the way basis vectors are multiplied in the tensor product of $U_q(sl_2)$ representations,

$$\begin{aligned} \mathcal{O}_{i,\ell_i,m_i}(z_1,\bar{z}_1) \mathcal{O}_{j,\ell_j,m_j}(z_2,\bar{z}_2) = \sum_{\ell=|\ell_i-\ell_j|}^{\ell_i+\ell_j} \sum_{k\in[\ell]} \frac{C_{ijk}}{z_{21}^{h_{ijk}} \bar{z}_{21}^{\bar{h}_{ijk}}} \begin{bmatrix} \ell_i & \ell_j & \ell \\ m_i & m_j & m_i+m_j \end{bmatrix}_q \\ \times \left( \mathcal{O}_{k,\ell,m_i+m_j}(z_1,\bar{z}_1) + \ldots \right). \end{aligned} \qquad (7)$$

  An additional sum over $k$ is allowed if several primaries of the same spin $\ell$ appear in the OPE. Here, $C_{ijk}$ is the OPE coefficient, which encodes the dynamical information and does not depend on $m_i$ and $m_j$, and $h_{ijk} \equiv h_i + h_j - h_k$. Virasoro descendants are kinematically fixed by conformal symmetry. The sum over $\ell$ effectively starts from $\ell = |m_1 + m_2|$ because the Clebsch-Gordan coefficients vanish otherwise.

- CFT three-point functions are given by

$$
\left\langle \mathcal{O}_i(z_1,\bar{z}_1)\mathcal{O}_j(z_2,\bar{z}_2)\mathcal{O}_k(z_3,\bar{z}_3)\right\rangle = C_{ijk} \begin{bmatrix} \ell_i & \ell_j & \ell_k \\ m_i & m_j & -m_k \end{bmatrix}_q \begin{bmatrix} \ell_k & \ell_k & 0 \\ -m_k & m_k & 0 \end{bmatrix}_q
$$
$$
\times \frac{1}{z_{21}^{h_{ijk}} z_{32}^{h_{jki}} z_{31}^{h_{ikj}}} \frac{1}{\bar{z}_{21}^{\bar{h}_{ijk}} \bar{z}_{32}^{\bar{h}_{jki}} \bar{z}_{31}^{\bar{h}_{ikj}}} .
\tag{8}
$$

Here $z_{ij} \equiv z_i - z_j$ and $\bar{z}_{ij} \equiv \bar{z}_i - \bar{z}_j$.

- The non-locality of operators manifests itself in the following rule for the permutation of operators

$$
\sum_{m_i',m_j'} [R_{\ell_j,\ell_i}]^{m_j' m_i'}_{m_j m_i} \mathcal{O}_{i,\ell_i,m_i'}(x)\mathcal{O}_{j,\ell_j,m_j'}(y) = \pm \mathcal{O}_{j,\ell_j,m_j}(y)\mathcal{O}_{i,\ell_i,m_i}(x),
\tag{9}
$$

where $R_{\ell_1,\ell_2}$ is the $R$-matrix of $U_q(sl_2)$, and the $\pm$ accounts for the bosonic/fermionic statistics of the operators. This rule for permutation of operators is called a braid locality condition and it is consistent with the action of the quantum group symmetries. Moreover, it constrains the allowed spin of operators to be

$$
\text{spin} = \frac{\ell(\ell+1)}{\pi i} \log q - \ell + \mathbb{Z} + \frac{1-(-1)^F}{4} .
\tag{10}
$$

**XXZ$_q$ CFT.** In this paper, we consider the Coulomb gas approach to constructing CFT correlation functions that respect the $U_q(sl_2)$ symmetry. Specifically, we apply this approach to a theory with $U_q(sl_2)$ global symmetry, which emerges in the continuum limit of the XXZ$_q$ spin chain, as described in [2] and [1]. In this limit, the spin chain flows to a CFT with quantum group symmetry, which we refer to as the XXZ$_q$ CFT. Below, we summarize some key details of this example.

First, the parameter $q$ in the quantum group is a defining parameter of the CFT. We introduce the notation:

$$
q = e^{i\pi \frac{\mu}{\mu+1}}, \qquad \mu > 0,
\tag{11}
$$

which is convenient for expressing the central charge of the theory:

$$
c = 1 - \frac{6}{\mu(\mu+1)} .
\tag{12}
$$

The spectrum of the theory consists solely of degenerate Virasoro primary fields and their descendants. The conformal dimensions of the Virasoro primary fields are in the Kac table [7] and are given by

$$
h_{r,s} = \frac{[r(\mu+1) - s\mu]^2 - 1}{4\mu(\mu+1)} ,
\tag{13}
$$

where $r$ and $s$ are integers. We denote the characters of the associated Virasoro representations as $\chi_{r,s}(\tau)$. The partition function of the XXZ$_q$ CFT is given by [13]

$$
Z(\tau,\bar{\tau}) = \sum_{\ell=0}^{\infty} (2\ell+1) \sum_{r=1}^{\infty} \chi_{r,2\ell+1}(\tau)\chi_{r,1}(\bar{\tau}) .
\tag{14}
$$

Primary operators in the theory correspond to states with conformal dimensions $(h_{r,2\ell+1}, h_{r,1})$, and are $(2\ell+1)$-fold degenerate. These operators transform under the spin-$\ell$ representation of

$U_q(sl_2)$. The sum over $\ell$ runs over integers, meaning that the operators in this theory possess only integer quantum group spin. We denote these operators as

$$W^m_{r,2\ell+1} \longleftrightarrow (h_{r,2\ell+1}, h_{r,1}), \tag{15}$$

where the index $m \in \{-\ell, \ldots, \ell\}$ is the quantum number that transforms under $U_q(sl_2)$.

Note that there is a closed chiral subsector of the theory, which consists of operators $W^m_{1,2\ell+1}$ with conformal dimensions $(h_{1,2\ell+1}, 0)$. The analysis of correlation functions and OPE coefficients is significantly simplified in this subsector.

The OPE between operators (15) is given by

$$
\begin{aligned}
W^{m_1}_{r_1,2\ell_1+1}(z_1,\bar{z}_1) W^{m_2}_{r_2,2\ell_2+1}(z_2,\bar{z}_2) = &\sum_{\ell=|\ell_1-\ell_2|}^{\ell_1+\ell_2} \sum_{\substack{r_3=|r_1-r_2|+1 \\ r_1+r_2+r_3 \text{ odd}}}^{r_1+r_2-1} \frac{C^{\text{XXZ}}_{(r_1,2\ell_1+1),(r_2,2\ell_2+1),(r_3,2\ell+1)}}{z_{21}^{h_{123}} \bar{z}_{21}^{\bar{h}_{123}}} \\
&\times \begin{bmatrix} \ell_1 & \ell_2 & \ell \\ m_1 & m_2 & m_1+m_2 \end{bmatrix}_q \left( W^{m_1+m_2}_{r_3,2\ell+1}(z_1,\bar{z}_1) + \ldots \right),
\end{aligned} \tag{16}
$$

where again we use the shorthand notation $h_{123} = h_1 + h_2 - h_3 = h_{r_1,s_1} + h_{r_2,s_2} - h_{r_3,s_3}$. For convenience we sometimes use the notation $s_i = 2\ell_i + 1$. Using the conformal bootstrap methods, the OPE coefficients were found to be equal to

$$\left( C^{\text{XXZ}}_{(1,s_1),(1,s_2),(1,s_3)} \right)^2 = \pm C^{\text{MM}}_{(1,s_1),(1,s_2),(1,s_3)}, \tag{17}$$

for the chiral subsector, and

$$\left( C^{\text{XXZ}}_{(r_1,s_1),(r_2,s_2),(r_3,s_3)} \right)^2 = \pm C^{\text{MM}}_{(r_1,s_1),(r_2,s_2),(r_3,s_3)} C^{\text{MM}}_{(r_1,1),(r_2,1),(r_3,1)}, \tag{18}$$

for generic operators. The coefficients $C^{\text{MM}}_{(r_1,s_1),(r_2,s_2),(r_3,s_3)}$ are the well-known OPE coefficients of minimal models, see [3–5, 14, 15]. In sections 3.6 and 4.2 we will test these results versus the Coulomb gas computations. The formula above can determine the value of $C^{\text{XXZ}}_{(r_1,s_1),(r_2,s_2),(r_3,s_3)}$ up to a phase that is an integer power of $i$. We will be able to resolve this ambiguity. The final relation between OPE coefficients with explicit phases is given as follows:

$$C^{\text{XXZ}}_{(r_1,s_1),(r_2,s_2),(r_3,s_3)} = e^{i\Theta(1,2,3)} \sqrt{C^{\text{MM}}_{(r_1,s_1),(r_2,s_2),(r_3,s_3)} C^{\text{MM}}_{(r_1,1),(r_2,1),(r_3,1)}}, \tag{19}$$

where the square roots are taken to be positive for $c$ close to (but smaller than) 1, and the phase factor $\Theta(1,2,3)$ is given by:[1]

$$
\begin{aligned}
\Theta(1,2,3) = &\frac{\pi}{4}\Big[(r_1-1)(s_1-1) + (r_2-1)(s_2-1) - (r_3-1)(s_3-1) \\
&\qquad - 2(r_1-1)(s_2-1) + (r_1+r_3-r_2-1)(s_2+s_3-s_1-1)\Big] \\
&+ \pi\Bigg[ D(r_1-1,s_1-1) + D(r_2-1,s_2-1) + D(r_3-1,s_3-1) \\
&\qquad + D\left(\frac{r_1+r_2-r_3-1}{2}, \frac{s_1+s_2-s_3-1}{2}\right) + \text{ cyclic permutations of (123)} \\
&\qquad + D\left(\frac{r_1+r_2+r_3-3}{2}, \frac{s_1+s_2+s_3-3}{2}\right)\Bigg],
\end{aligned} \tag{20}
$$

---

[1] In conformal field theory, the normalization of two-point functions allows for a residual ambiguity: $\mathcal{O}_i \to (-1)^{n_i}\mathcal{O}_i$ for integer $n_i$. As a result, the OPE coefficients can shift as $C_{ijk} \to (-1)^{n_i+n_j+n_k} C_{ijk}$. This implies that the third line in the expression for $\Theta(1,2,3)$ can be absorbed into a field redefinition of the form $W_{r,s} \to (-1)^{D(r-1,s-1)} W_{r,s}$.

where the function $D(m,n)$ is defined as

$$D(m,n) := \theta(n > 0)\left[\frac{1}{2}m(m+1) - \frac{1}{2}(m-n)(m-n+1)\,\theta(m > n)\right], \qquad (21)$$

and $\theta(a > b)$ is the indicator function for the condition $a > b$.

In (20), recall that the $s_i$ are odd integers in the $\text{XXZ}_q$ CFT. The first line determines the power of $i$, while the remaining lines contribute only to overall $\pm$ signs.

## 3 Coulomb gas construction of quantum group symmetry

In this section we propose a non-standard Coulomb gas integral, which produces correlation functions of $c < 1$ CFTs with a quantum group symmetry. The correlation functions directly satisfy the constraints imposed by the conformal invariance, as well as the constraints of the quantum group symmetry reviewed in section 2. In addition, this construction gives us an intuitive picture about how the operators in the theory look like: they are generically non-local operators and are attached to topological lines.[2]

The starting point of our construction is the fact that the quantum group generators can be realized using screening charges [9]. Then we identify the operators transforming in quantum group multiplets and construct their correlation functions in section 3.2. To some extent, a similar construction was done in [17]; we would like to emphasize that our construction is however different (see footnote 7). Then we proceed with our definition to compute the generic two-point functions in section 3.4, and generic OPE coefficients in section 3.5. The normalization of operators given upon us from the Coulomb gas integrals is slightly different from the canonical normalization of section 2. So, the normalization procedure is performed in section 3.6.

For simplicity, this section provides only a construction for operators of conformal dimension $(h_{1,s}, 0)$. That is, it covers the whole chiral subsector of $\text{XXZ}_q$ CFT. The construction of non-chiral operators is done by combining chiral and anti-chiral operators and is considered in section 4.

### 3.1 Review of the free boson theory with a background charge

Let us start with reviewing the basic notations of Coulomb gas formalism which we are using to construct the correlation functions respecting $U_q(sl_2)$ symmetry. We do not review the last step of the usual Coulomb gas approach, which is combining chiral and anti-chiral parts into correlation functions of local operators, as done in [3,4]. Instead, we introduce the necessary ingredients and move on to the construction of correlation functions of chiral operators with a quantum group symmetry in section 3.2.

We start with a chiral free field $\phi(z)$. The background charge effectively modifies the stress-energy tensor of the free boson to

$$T(z) = -\frac{1}{2} : (\partial\phi)^2 : +i\sqrt{2}\alpha_0\partial^2\phi. \qquad (22)$$

The OPE of the stress-energy tensor with itself gives the central charge of the theory,

$$c = 1 - 24\alpha_0^2. \qquad (23)$$

---

[2]These lines might satisfy fewer or different properties than topological defect lines, see e.g. [16].

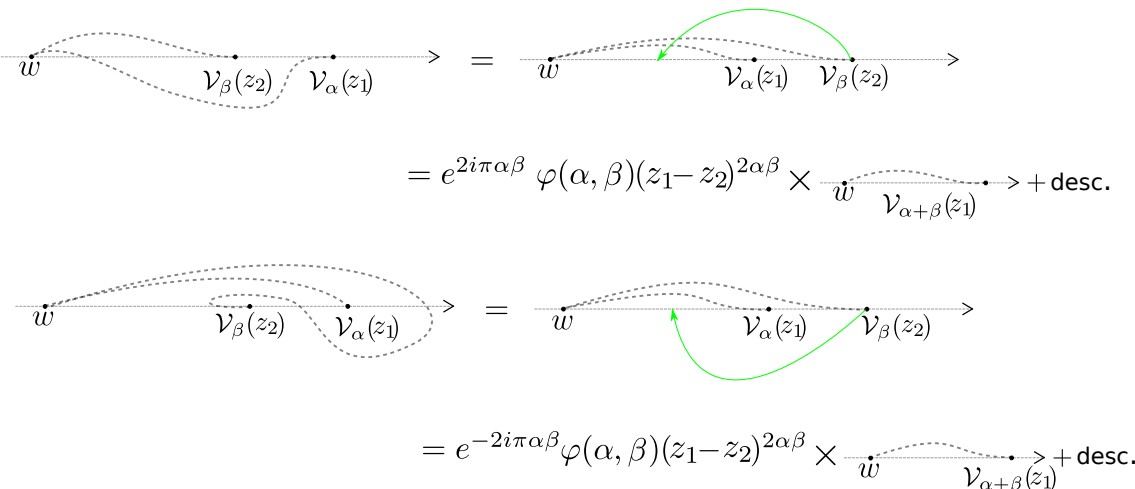

Figure 1: The operator and topological line configuration corresponding to the definition of the OPE in (26). The value of the OPE for any other configuration can be found by continuously deforming a configuration of this type.

Figure 2: Eq. (26) provides the OPE coefficient when the ordering in space coincides with the operator ordering. The coefficient for other configurations can be determined by smoothly deforming the configuration of figure 1, as shown in the examples given in this figure.

Identifying this with the standard parametrization of the central charge, given by (12), suggests the relation between $\alpha_0$ and $\mu$,

$$\alpha_0 = \frac{1}{\sqrt{4\mu(\mu+1)}}. \tag{24}$$

When $\alpha_0 = 0$, the primary operators in the theory are $\partial\phi$ and the vertex operators (exponentials of the free field $\phi(z)$). When $\alpha_0 \neq 0$, $\partial\phi$ is no longer a primary operator while the vertex operators remain primaries. Because of the background charge, the conformal dimensions of the vertex operators are shifted. This can be checked by computing the OPE of $T(z)T(w)$. As is standard in the Coulomb-Gas formalism, we treat the chiral and anti-chiral pieces of the theory separately. We denote the chiral vertex operators and their conformal dimensions as

$$\mathcal{V}_\alpha(z) =: e^{i\sqrt{2}\alpha\phi(z)}:, \qquad h_\alpha = \alpha^2 - 2\alpha_0\alpha. \tag{25}$$

The chiral free field $\phi$ and the chiral vertex operator $\mathcal{V}_\alpha(z)$ are non-local operators [18]. They are attached to topological defect lines, denoted in our figures by gray dashed lines (see figures 1 and 2, for the notations of the ordering see [1, section 3.1]), and their correlation functions are multi-valued. While we do not give a microscopic construction of these topological lines, all of their properties can be inferred from the OPE, which will be given shortly. Note that while these lines do play the role of keeping track of ordering, they are just one ingredient of the full quantum group lines (which will appear later on). We choose all the lines to originate at an arbitrary point $w$. We will see that it is a natural choice from the point of view of the rest of the construction.

The OPE of chiral vertex operators is given by

$$\mathcal{V}_\alpha(z_1)\mathcal{V}_\beta(z_2) = \varphi(\alpha,\beta)(z_2 - z_1)^{2\alpha\beta}\mathcal{V}_{\alpha+\beta}(z_1) + \dots, \tag{26}$$

$$
\left\langle w \quad \mathcal{V}_\alpha(z_1) \, \mathcal{V}_\beta(z_2) \right\rangle = e^{2i\pi\alpha\beta} \, \frac{\varphi(\beta,\alpha)}{\varphi(\alpha,\beta)} \left\langle w \quad \mathcal{V}_\alpha(z_1) \, \mathcal{V}_\beta(z_2) \right\rangle
$$

$$
\left\langle w \quad \mathcal{V}_\alpha(z_1) \, \mathcal{V}_\beta(z_2) \right\rangle = e^{-2i\pi\alpha\beta} \, \frac{\varphi(\beta,\alpha)}{\varphi(\alpha,\beta)} \left\langle w \quad \mathcal{V}_\alpha(z_1) \, \mathcal{V}_\beta(z_2) \right\rangle
$$

Figure 3: The phases relating different operator orderings.

where "+..." denotes the contribution from descendants of $\mathcal{V}_{\alpha+\beta}$.[3] Schematically, the OPE is represented in figure 1. The convention for the fractional power is that it is positive when $z_2 - z_1$ is positive. Here, $\varphi(\alpha,\beta)$ are phases constrained by OPE associativity, allowed because of the non-locality of the chiral vertex operators.[4] The $\varphi(\alpha,\beta)$ are not necessarily symmetric under the exchange $\alpha \leftrightarrow \beta$. In the literature these phases are typically referred to as cocycles (see for example [19, Chapter 8]). They are typically fixed by requiring the absence of monodromy when permuting the local operators that can be constructed from a product of non-local operators. It turns out that for OPE involving only operators of conformal dimensions $(h_{1,s}, 0)$, and as a subset those needed for the construction of the chiral subsector of the XXZ$_q$ CFT, we can set all the cocycles to 1. In section 4 we will discuss more general operators where non-trivial cocycles will make an appearance.

When the operator ordering does not coincide with the ordering in space, one has to smoothly deform (26) to obtain the expression for the OPE. Figure 2 provides us with the phase factors appearing after changing the space ordering of operators in the two possible ways.

Combining figures 1 and 2, we can compute the phases relating configurations with operators in different ordering. These are given in figure 3. These phases will be important later as a part of the construction of quantum group lines.

In the figures 1, 2, and 3 we were careful to draw the lines so that they always approach each operator with the same angle. This is important because the expectation value generally depends on the angle of approach (see for example [16]). In favor of readability of figures, we will not be careful with the angles of approach in the rest of this paper. The reader should understand the lines as always approaching operator insertions with the same angle as in this figure. This also applies to the operator at point $w$.

Correlators of the vertex operators defined above are used as a building block to construct correlation functions of $c < 1$ theories in the original approach of [4]. We are using the same building blocks to construct correlation functions in a theory with quantum group symmetry. There is one more important element of the construction, the screening operators. To proceed with the definition of the screening operators, notice that the quadratic equation $h_\alpha = 1$ has two solutions, thus this theory admits two dimension 1 vertex operators $\mathcal{V}_{\alpha_+}(z)$ and $\mathcal{V}_{\alpha_-}(z)$, where

$$
\begin{aligned}
\alpha_+ &= \alpha_0 + \sqrt{\alpha_0^2 + 1} = \sqrt{\frac{\mu+1}{\mu}}, \\
\alpha_- &= \alpha_0 - \sqrt{\alpha_0^2 + 1} = -\sqrt{\frac{\mu}{\mu+1}}.
\end{aligned}
\tag{27}
$$

---

[3]When $\alpha + \beta = 0$, $\partial\phi$ and its descendants appear as a part of the conformal multiplet of the identity operator, $\mathcal{V}_0 \equiv 1$.

[4]Note that the action of the lines on any of the operators can be reverse engineered starting from the expression for the OPE (26). If we restrict to correlators of only operators of dimensions $h_{1,s}$ or to correlators only operators of dimensions $h_{r,1}$, these lines can be taken to be $U(1)$ symmetry lines in the original free boson with a background charge theory.

Note that $\alpha_\pm$ satisfy the relations $\alpha_+\alpha_- = -1$ and $\alpha_+ + \alpha_- = 2\alpha_0$. The screening operators are defined as

$$S_+ = \oint dz\, \mathcal{V}_{\alpha_+}(z)\,, \qquad S_- = \oint dz\, \mathcal{V}_{\alpha_-}(z)\,. \tag{28}$$

The integrals are topological. Since the dimension of $\mathcal{V}_{\alpha_\pm}$ is 1, the screening operators are of dimension zero and commute with Virasoro algebra. In other words, $S_\pm$ do not change the conformal properties when they are inserted in correlation functions, but rather only change the $U(1)$ charge by one unit of $\sqrt{2}\alpha_\pm$ respectively. Thus, they are perfect candidates for being generators of some internal symmetry. Furthermore, one can show that when $\alpha_0 > 0$, $S_+$ and $S_-$ commute with each other if we choose proper integration contours. We summarize the properties of $S_\pm$ as follows:

$$[S_\pm, L_n] = 0\,, \qquad [Q, S_\pm] = \sqrt{2}\alpha_\pm S_\pm\,, \qquad [S_+, S_-] = 0 \text{ (if } \alpha_0 > 0)\,, \tag{29}$$

where $Q$ is the $U(1)$ charge:

$$Q = \frac{1}{2\pi} \oint dz\, \partial\phi(z)\,. \tag{30}$$

While in the standard Coulomb-Gas construction the only role of the screening charges is to absorb the left-over U(1) charge in order to produce a nonzero correlator, in our construction they are taken as proper generators of a global symmetry. Let us also mention the convenient parametrization of the dimensions of the degenerate fields from the Kac table (13), which can be expressed through $\alpha_\pm$:

$$h_{r,s} = h_\alpha\,, \qquad \text{for } \alpha = \alpha_{r,s} = \frac{1-r}{2}\alpha_+ + \frac{1-s}{2}\alpha_-\,. \tag{31}$$

In particular, until the end of this section we restrict ourselves to the operators with indices $r = 1, s = 2\ell + 1$, for which $\alpha_{1,2\ell+1} = -\ell\alpha_-$, where $\ell$ can be integer or half-integer. The relevant generator for the next subsections is $S_-$, while $S_+$ will only make an appearance in section 4. Hence, for now we only make use of the following set of correlators,

$$\left\langle \mathcal{V}_{k_1\alpha_-}(z_1)\dots\mathcal{V}_{k_n\alpha_-}(z_n) \right\rangle_{\text{CG}} = \prod_{i<j}(z_j - z_i)^{2k_i k_j \alpha_-^2}\,, \tag{32}$$

with integer or half-integer $k_i$'s. Here, as for the OPE, the RHS is defined to be positive when $i < j \Rightarrow z_i < z_j$. As mentioned above, the cocycle structure for these subset of generators is very simple, $\varphi(k_1\alpha_-, k_2\alpha_-) = 1$. A subscript "CG" stands for the terminology "Coulomb gas". The correlation function is non-zero only if the anomalous charge conservation is satisfied,[5]

$$\alpha_1 + \dots + \alpha_n = 2\alpha_0\,. \tag{33}$$

In the original Coulomb gas construction, there is some arbitrariness in the choice of vertex operators and screening charges. For example, one can sometimes (but not always) replace a vertex operator of charge $\alpha_{r,s}$ with the one of charge $2\alpha_0 - \alpha_{r,s}$, since both have the same conformal dimension. However, this does not imply that the corresponding operators $\mathcal{V}_\alpha$ and $\mathcal{V}_{2\alpha_0-\alpha}$ are equivalent, although they may coincide in specific correlation functions. A simple example is $\mathcal{V}_{2\alpha_0}$, which has conformal dimension zero, yet behaves like the identity operator only in very special cases, as we will discuss later in sections 3.2.1 and 4.1. In general, however, we regard $\mathcal{V}_\alpha$ and $\mathcal{V}_{2\alpha_0-\alpha}$ as distinct operators.

---

[5]A careful treatment of two-dimensional massless free scalar and its vertex operators leads to an extra factor of $\mu_{\text{IR}}^{\frac{1}{4\pi}(\alpha_1+\dots\alpha_n-2\alpha_0)^2}$ in the correlation function, where $\mu_{\text{IR}}$ is the infrared cut-off of the theory. So in the infrared limit $\mu_{\text{IR}} \to 0$, the anomalous charge conservation (33) is necessary for a non-zero correlation function.

## 3.2 Correlation functions of operators $(h_{1,s}, 0)$

Let us first focus only on the closed chiral subsector which consists of operators of conformal dimensions $(h_{1,2\ell+1}, 0)$, where $\ell \in \mathbb{Z}/2$. The analysis of these operators simplifies compared to more general cases: there is a single quantum group multiplet for each spin $\ell$, thus the fusion rules involve a simpler summation, solely over the second Kac index, rather than a double summation over both $r$ and $s$. The correlation functions in this case already demonstrate the main features we aim to present. Moreover, this covers the chiral subsector of $XXZ_q$ CFT. Therefore, we focus on this specific case in this section and discuss the general correlation functions of the theory in section 4.2.

Let us start with constructing an appropriate space of operators transforming under the quantum group. In particular, for a given $\ell$ one should find $2\ell + 1$ primary operators that form a spin-$\ell$ representation of $U_q(sl_2)$. Following [9,10], we utilize the vertex and screening operators to construct such a representation (we explain similarities and differences with similar approaches that appeared in the literature in footnote 7). We take vertex operators $\mathcal{V}_{\alpha_{1,2\ell+1}}$ and act on them with the screening operator $S_-$,

$$\mathcal{V}_{\alpha_{1,2\ell+1}}(x) \longrightarrow S_-^n \mathcal{V}_{\alpha_{1,2\ell+1}}(x) = \int_C dz_1 \dots dz_n \mathcal{V}_{\alpha_{1,2\ell+1}}(x) \mathcal{V}_{\alpha_-}(z_1) \dots \mathcal{V}_{\alpha_-}(z_n), \qquad (34)$$

without changing the conformal dimension of the operator. The action of $S_-$, of course, depends on the choice of the contour. We choose the contours as is shown in the left hand side of figure 4: the integration contours start and end at some point $w$ and surround the vertex operator $\mathcal{V}_{\alpha_{1,2\ell+1}}(x)$. Let us call this collection of contours as $C$. Below we will show that correlators do not depend on the $w$. The integrand may at first appear ambiguous, as the OPE's are multivalued functions. However, it is completely specified by (26) (or, equivalently, (32)). The interpretation of (32) is that the integrand is real and positive when all the vertex operators are located on the real line with the following order: $w < x < z_1 < \dots < z_n$.

Let us now show that for such a choice of vertex operators the integrals vanish for some large enough $n$, i.e. for a single operator $\mathcal{V}_{\alpha_{1,2\ell+1}}$ we are constructing a finite dimensional space of operators. To do this let us make some contour manipulations around the point $x$. We can use a different representation of a given integral so that all the contours start at $w$ and end at $x$. Let us call the new contour $L$. For the integral to be the same, an additional factor $A_L^C$ appears, see the right hand side of figure 4. The precise definition of the line integral representation, as well as a discussion about why its divergences do not lead to ambiguities in the finite value assigned to it, are given in appendix B. Here we provide the result of the computation:

$$[A_L^C]_n^\ell = \prod_{j=0}^{n-1} \left( e^{(2\ell-2j)i\pi\alpha_-^2} - e^{-2\ell i\pi\alpha_-^2} \right) = \prod_{j=0}^{n-1} \left( q^{2\ell-2j} - q^{-2\ell} \right), \qquad (35)$$

where we are using the notation $q = e^{i\pi\alpha_-^2}$. The reason for such notation is explained in section 3.3. It is clear that this factor vanishes for $n \geq 2\ell + 1$. This observation gives us $2\ell + 1$ primary operators with the same conformal dimension $h_{1,2\ell+1}$:

$$\mathcal{V}_{\alpha_{1,2\ell+1}} \xrightarrow{S_-} S_- \mathcal{V}_{\alpha_{1,2\ell+1}} \xrightarrow{S_-} S_-^2 \mathcal{V}_{\alpha_{1,2\ell+1}} \xrightarrow{S_-} \dots \xrightarrow{S_-} S_-^{2\ell} \mathcal{V}_{\alpha_{1,2\ell+1}} \xrightarrow{S_-} 0.$$

This is exactly the spin-$\ell$ representation of $U_q(sl_2)$ we want. Note that the termination of representations depends on the proper choice of parameter $\alpha$ of the vertex operator, thus on the conformal dimension. We interpret $\mathcal{V}_{\alpha_{1,2\ell+1}}$ as the highest-weight state $|\ell, \ell\rangle$; and $S_-$ as the lowering operator "$F$", which gives all the other states $|\ell, m\rangle$ in the multiplet. The charge $\alpha_{1,2\ell+1} + n\alpha_- = (-\ell + n)\alpha_-$ is proportional to the $U(1)$ charge:

$$V_{\ell,m} \equiv S_-^{\ell-m} \mathcal{V}_{\alpha_{1,2\ell+1}}, \qquad H \cdot V_{\ell,m} \equiv -\frac{2}{\alpha_-} \left[ \alpha_{1,2\ell+1} + (\ell-m)\alpha_- \right] V_{\ell,m} = 2m V_{\ell,m}, \qquad F = S_-. \quad (36)$$



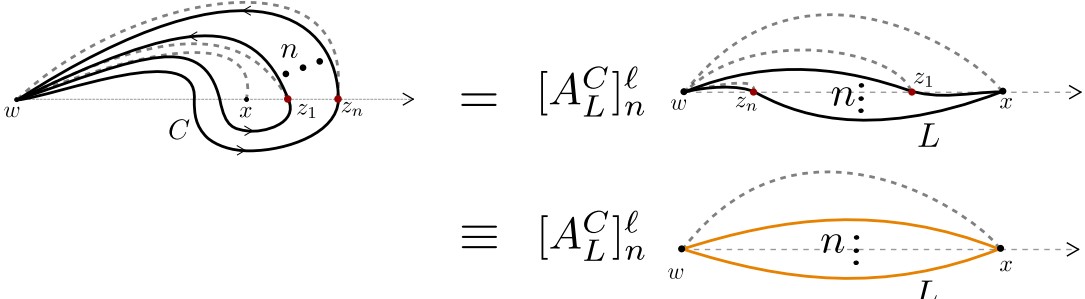

Figure 4: Dressing of a vertex operator $\mathcal{V}_{\alpha_{1,2\ell+1}}(x)$ with the screening operators defines an operator $V_{\ell,m}(x)$ transforming in a quantum group multiplet of spin-$\ell$. Note that the dashed topological lines are allowed to cross the integration contours (solid lines). Only dashed lines crossing each other and contours crossing each other are not allowed. The number of screening operators is equal to $n = \ell - m$. The RHS of the figure shows a different integral representation of the same operator. We have also introduced a new graphical notation in the second line of this figure. The orange contours stand in for the $z_1, \ldots, z_n$ integrals in the RHS of the first line. When we use the orange contour, it is always implied that the topological lines from the integrated points are configured exactly as in the RHS of the first line.

Note that with this identification we do not immediately get canonically normalized operators satisfying (3) and (6), instead, we are using the notation $V_{\ell,m}$. The first reason is that the relative normalizations of operators $V_{\ell,m}$ in the same multiplet are not as in (3). This can be fixed by rescaling the operators by appropriate products of $f_m^\ell$. The second reason is that the two-point functions of $V_{\ell,m}$ are not normalized under the convention of (6). Overall, one has to redefine the operators by a rescaling: $\mathcal{O}_{\ell,m} = \lambda_{\ell,m} V_{\ell,m}$. We will come back to the canonically normalized operators in section 3.6 after computing the Coulomb gas integrals for the two point functions. Nonetheless, the quantum group symmetry manifests itself in either normalization, and for now we focus on the operators $V_{\ell,m}$.

We do not know an explicit form of the raising operator $E$.[6] However, as we have emphasized in section 2, knowing $H$ and $F$ is sufficient to derive all the properties of correlation functions with quantum group symmetry.

Now let us proceed with defining the correlation functions of operators $V_{\ell,m}$. As it was pointed out in section 2, the most immediate consequence of the quantum group Ward identities is that the total $U(1)$ charge of the correlation function should be zero:

$$\left\langle V_{\ell_1,m_1}(x_1)\ldots V_{\ell_n,m_n}(x_n)\right\rangle \neq 0 \implies m_1 + m_2 + \ldots + m_n = 0. \tag{37}$$

This means that the charges of the vertex operators $\mathcal{V}_{-\ell_i\alpha_-}$ and screening vertex operators $\mathcal{V}_{\alpha_-}$ sum up to zero. However, in the Coulomb gas formalism we need an extra charge $2\alpha_0$ to satisfy the anomalous charge conservation (33), otherwise the correlation function vanishes. For this reason, we propose the following construction of the correlation functions:

$$\left\langle V_{\ell_1,m_1}(x_1)\ldots V_{\ell_n,m_n}(x_n)\right\rangle$$
$$\equiv \int_\Gamma dz^{(1)}\ldots dz^{(n)}\left\langle \mathcal{V}_{2\alpha_0}(w)\mathcal{V}_{-\ell_1\alpha_-}(x_1)\mathcal{V}_{\alpha_-}\left(z^{(1)}\right)\ldots \mathcal{V}_{-\ell_n\alpha_-}(x_n)\mathcal{V}_{\alpha_-}\left(z^{(n)}\right)\right\rangle_{\text{CG}} \tag{38}$$
$$\equiv \int_\Gamma dz^{(1)}\ldots dz^{(n)} f\left(w, x_1, \ldots, x_n, z_1^{(1)}, \ldots, z_{\ell_1-m_1}^{(1)}, \ldots, z_{\ell_n-m_n}^{(n)}\right).$$

---

[6]See [10] for a proposal for how $E$ could be constructed.

Here $\langle\ldots\rangle_{\text{CG}}$ denotes the Coulomb-gas correlation function (32). We also introduced the shorthand notations for all the integration variables, i.e. for each $k \in \{1,\ldots,n\}$,

$$dz^{(k)} \equiv \prod_{j_k=1}^{\ell_k - m_k} dz_{j_k}^{(k)}, \qquad \mathcal{V}_{\alpha_-}\left(z^{(k)}\right) \equiv \prod_{j_k=1}^{\ell_k - m_k} \mathcal{V}_{\alpha_-}\left(z_{j_k}^{(k)}\right). \tag{39}$$

We insert an operator $\mathcal{V}_{2\alpha_0}$ of conformal dimension 0 at the point $w$. All the integration contours start and end at $w$. The full contour $\Gamma$ is shown in figure 5 (there we put all $x_k$ on the real axis for convenience).[7] Anomalous charge conservation (33) immediately imposes $m_1 + \cdots + m_n = 0$. Lastly, the integrand is in general a multivalued function: using an expression (32) for the Coulomb gas correlation function, we get

$$f(w, x_1, \ldots, x_n, z_1, \ldots, z_N) = \left(\prod_{1 \leqslant i < j \leqslant n} (x_j - x_i)^{2\ell_i \ell_j \alpha_-^2}\right)\left(\prod_{i=1}^{n}\prod_{k=1}^{N}(\pm(z_k - x_i))^{-2\ell_i \alpha_-^2}\right)$$
$$\times \left(\prod_{1 \leqslant r < s \leqslant N} (z_s - z_r)^{2\alpha_-^2}\right)\left(\prod_{k=1}^{N}(z_k - w)^{2\alpha_-^2 - 2}\right) \tag{40}$$
$$\times \left(\prod_{i=1}^{n}(x_i - w)^{2\ell_i(1 - \alpha_-^2)}\right).$$

Where the "$\pm$" sign is chosen to be a "$+$" if $z_k$ is to the right of $x_i$ in (38), and "$-$" otherwise. Note that one outcome is that the integrand is real and positive when all the vertex operators are on the real line with the following order:

$$w < x_1 < z_1^{(1)} < \ldots < z_{\ell_1 - m_1}^{(1)} < x_2 < z_1^{(2)} < \ldots < x_n < \ldots < z_{\ell_n - m_n}^{(n)}. \tag{41}$$

The integral (38) is convergent when $\text{Re}(\alpha_-^2) > \frac{1}{2}$. For other generic values of $\alpha_-$, the integral is unambiguously defined by analytic continuation from the convergent regime.

In appendix C we define the action of Virasoro generators on the operators $V_{\ell,m}$ and show that they commute with quantum group generators.

The above construction defines a collection of CFT correlation functions which respect $U_q(sl_2)$ symmetry in the following sense:

1. (Well-definedness) The correlation functions do not depend on $w$.

2. (BPZ) They satisfy the BPZ differential equations.

3. (Quantum group symmetry) They satisfy all the Ward identities of the quantum group $U_q(sl_2)$ with $q = e^{i\pi\alpha_-^2}$.

4. (OPE) The $n$-point function can be expanded in terms of $(n-1)$-point functions via OPE.

---

[7]The Coulomb gas integral (38) and figure 5 are similar to the ones appearing in [17]. We would like to point out some advantages of our approach: because of the insertion of the operator $\mathcal{V}_{2\alpha_0}$ at the point $w$, correlation functions satisfy the anomalous charge conservation law (33). Moreover, as we will see in section 3.2.1, the definition (38) with inserted $\mathcal{V}_{2\alpha_0}(w)$ is independent of $w$ for any choice of $m_1, \ldots, m_n$. Whereas in [17] only the combination of integrals which corresponds to the highest-weight states is independent of $w$.

Let us also mention the distinction between our approach and [9, 10]. While we are following these references to construct finite dimensional representations of the quantum group using the screening charges, the papers do not give a concrete proposal for the construction of correlators. These references also provide a construction of a chiral theory with 2 quantum groups. However, our construction of section 4.1 differs by a crucial choice of cocycle. Namely, an additional twisting of the 2 quantum group symmetries is not present in our case. Finally, neither of the publications gave a proposal for the non-chiral operators, which we will give in section 4.2.

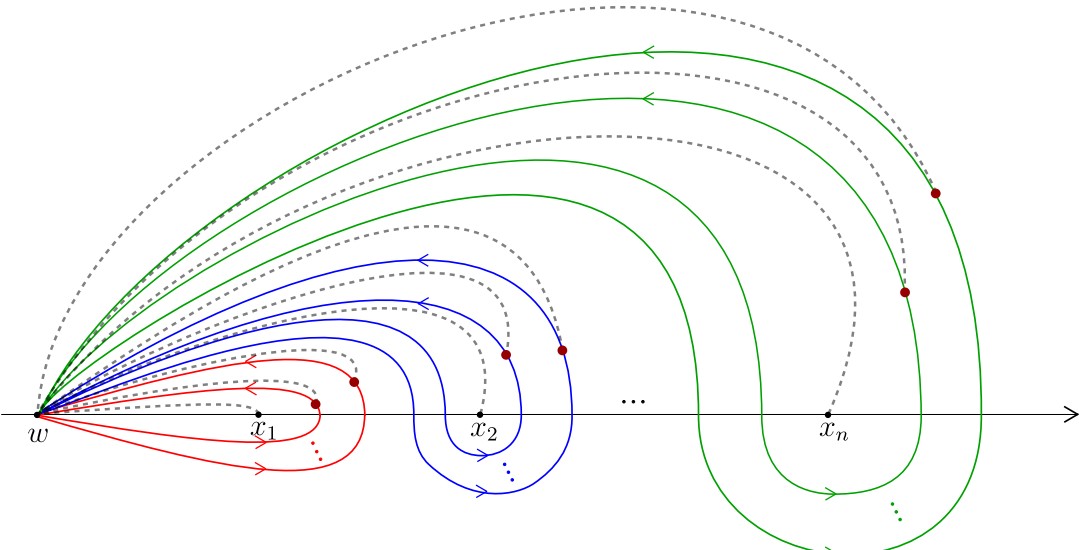

Figure 5: Integration contour $\Gamma$ for the definition of correlation function (38). The black dots denote the positions of the vertex operators $\mathcal{V}_{2\alpha_0}(w)$ and $\mathcal{V}_{-\ell_k \alpha_-}(x_k)$. The red, blue and green curves depict the integration contours for the screening charges around $x_1$, $x_2$, and $x_n$ respectively, with each contour corresponding to a single screening charge. The color coding for contours encircling different operators will be omitted in future figures. The integrand $f(w, x_1, \ldots, x_n, z_1^{(1)}, \ldots, z_{\ell_1 - m_1}^{(1)}, \ldots, z_{\ell_n - m_n}^{(n)})$ in (38) is real and positive when all operators are positioned on the real axis and ordered in the same way as in (41), as shown in the figure. The value of the integrand for other configurations can be obtained by analytic continuation.

We prove properties 1 and 2 for the chiral subsector of the theory in the following subsections 3.2.1-3.2.2. We then show that the correlation functions satisfy quantum group Ward identities in section 3.3 as well as show that this construction provides a realization of topological lines introduced in a companion paper [1]. We postpone the proof of property 4 to section 3.5. Let us explain why these properties are considered necessary and anticipated within our framework. Property 1 is essential as we aim to construct an $n$-point function of operators $\left\langle V_{\ell_1, m_1}(x_1) \ldots V_{\ell_n, m_n}(x_n) \right\rangle$ rather than an $(n+1)$-point function including also an operator $\mathcal{V}_{2\alpha_0}(w)$. Naively one can try to avoid this discussion by defining the $n$-point function as the $w \to \infty$ limit of the expression above. While that may be sufficient for a generic QFT, without $w$-independence the resulting $n$-point functions would not be invariant under special conformal transformations. Moreover, $w$-independence suggests we can think of the point $w$ as a sort of topological junction, whose properties are yet to be determined. Conveniently, $w$-independence significantly simplifies the direct computations of correlation functions: this is demonstrated in section 3.4, where setting $w = \infty$ makes expressions derived from (38) more tractable. Property 2 (BPZ) is required for the CFT if we are aiming to describe, for example, the XXZ$_q$ CFT, where operators are in the Kac table and in the partition function (14) the null states are explicitly subtracted in the characters $\chi_{r,s}$. Traditionally, global symmetries are evidenced by inserting a closed-contour integral of the current operator, $\oint j$, into the correlator and showing that: (1) the outcome remains unchanged under continuous deformation of the contour (conservation law), and (2) it vanishes as the contour is extended to infinity (the vacuum state is a singlet of the symmetry algebra). We use similar argument to show property 3. Finally, property 4 asserts that the integral (38) not only defines conformal kinematic functions, but also satisfies the defining properties of CFT correlation functions: it is determined by a universal set of data (spectrum and three-point structure constants) [20–22].

### 3.2.1 Independence of $w$

We would like to show that the Coulomb gas integral (38) does not dependent on $w$, the insertion point of $\mathcal{V}_{2\alpha_0}$. By eq. (25), $\mathcal{V}_{2\alpha_0}$ has conformal dimension 0, which suggests that it should behave in the same way as the identity operator. However, it is not always the case in non-unitary CFTs (which is our case). In fact, as indicated by eq. (40), the expression for a generic integrand does exhibit the dependence on $w$. We show that this apparent dependence on $w$ is effectively eliminated once all integrals over the positions of screening operators $\mathcal{V}_{\alpha_-}(z_k)$ are performed.

The $n$-point function (38), viewed as an $(n+1)$-point function involving $w$ and $x_k$, is invariant under global conformal transformation. So the $w$-independence of one- and two-point functions follows from the fact that they are kinematically fixed by conformal invariance:

$$\langle V_{\ell,m}(x)\rangle = \frac{const\ \delta_{\ell,0}\delta_{m,0}}{(w-x)^0}\,, \qquad \langle V_{\ell,m}(x_1)V_{\ell,-m}(x_2)\rangle = \frac{const}{(w-x_1)^0(w-x_2)^0(x_1-x_2)^{2h_{1,2\ell+1}}}\,, \tag{42}$$

where the powers of zero indicate that these correlators do not depend on the position $w$. For higher-point functions, the $w$-independence follows from the following lemma:

**Lemma 3.1.** *Let $\ell_1, \ell_2, \ldots, \ell_n$ ($n \geqslant 1$) be a collection of positive integers or half-integers such that their sum $N = \ell_1 + \ell_2 + \ldots \ell_n$ is an integer. The function $f$ defined in (40) satisfies the following identity*

$$\frac{\partial f}{\partial w} = \sum_{r=1}^{N} \frac{\partial}{\partial z_r}(g_r f)\,, \tag{43}$$

*where the function $g_r$ is defined by*

$$g_r(w,x_1,\ldots,x_n,z_1,\ldots,z_N) = -\left(\prod_{i=1}^{n}\left(\frac{z_r-x_i}{w-x_i}\right)^{2\ell_i}\right)\left(\prod_{\substack{s=1\\s\neq r}}^{N}\left(\frac{w-z_s}{z_r-z_s}\right)^2\right)\,. \tag{44}$$

We leave the proof of lemma 3.1 to appendix D. Let us see how this lemma implies the $w$-independence of the integral (38). We restrict our analysis to configurations where $w$ and all $x_i$ are distinct ($w \neq x_i$ for any $i$, and $x_i \neq x_j$ for any $i \neq j$). We first consider $\alpha_-$ in the regime $\text{Re}(\alpha_-^2) > 1$, the same conclusion will be extended to generic values of $\alpha_- \in \mathbb{C}$ through analytic continuation.

By lemma 3.1, we have:

$$\frac{\partial}{\partial w}\int_\Gamma dz_1 \ldots dz_N\, f(w,x_1,\ldots,x_n,z_1,\ldots,z_\ell) = \int_\Gamma dz_1 \ldots dz_N\, \frac{\partial f}{\partial w}$$

$$= \int_\Gamma dz_1 \ldots dz_N \sum_{r=1}^{N} \frac{\partial}{\partial z_r}(g_r f) \tag{45}$$

$$= \sum_{r=1}^{N}\int_\Gamma dz_1 \ldots \widehat{dz_r} \ldots dz_N (e^{i\varphi_r} - e^{i\varphi_r'})g_r f\Big|_{z_r=w}\,.$$

In the last line, the notation $\widehat{dz_r}$ indicates omission of the integral over $z_r$, and $e^{i\varphi_r}$ and $e^{i\varphi_r'}$ represent phase factors derived from the analytic properties of $g_r$ and $f$. Given that by (40) and (44), the function $g_r f$ vanishes when $z_r = w$ (recall that we are considering the regime $\text{Re}(\alpha_-^2) > 1$), the last line of (45) is equal to zero. This illustrates that the integral (38) does not depend on $w$ when $\text{Re}(\alpha_-^2) > 1$.

For other values of $\alpha_-$, the integral (38) is regularized by analytic continuation from the regime $\mathrm{Re}(\alpha_-^2) > 1$. Then $w$-independence holds for any $\alpha_- \in \mathbb{C}$.

Before concluding this subsection, we would like to comment on the relation between the $w$-independence observed here and the use of operators $\mathcal{V}_{2\alpha_0-\alpha}$ inside correlation functions in [3,4]. Suppose the correlation function (38) includes a lowest-weight operator $V_{\ell,-\ell}$. Since we have shown that the correlation function is independent of the position $w$, we can move the insertion point of $\mathcal{V}_{2\alpha_0}(w)$ to coincide with that of $V_{\ell,-\ell}$. This effectively yields a vertex operator with charge $2\alpha_0 + \ell\alpha_- \equiv 2\alpha_0 - \alpha_{1,2\ell+1}$. This observation explains the statement about the difference between vertex operators $\mathcal{V}_\alpha$ and $\mathcal{V}_{2\alpha_0-\alpha}$ made at the end of section 3.1: the insertion of a single $\mathcal{V}_{2\alpha_0-\alpha}$ operator can be interpreted (up to a constant factor) as putting an operator $\mathcal{V}_{2\alpha_0}$ and a lowest-weight operator together. Apart from this very special case, in our construction, generically $\mathcal{V}_\alpha$ and $\mathcal{V}_{2\alpha_0-\alpha}$ are distinct operators.

### 3.2.2 BPZ equations

In 2D unitary minimal model CFTs, the Belavin-Polyakov-Zamolodchikov (BPZ) equations play a pivotal role in describing the differential properties of correlation functions involving degenerate primary fields [8]. These equations arise when the conformal weights of the operators involved are in the Kac table. Since the operators $V_{\ell,m}$ considered in our analysis are in the Kac table, it suggests that their correlation functions

$$\langle V_{\ell_1,m_1}(x_1)\dots V_{\ell_n,m_n}(x_n)\rangle,$$

are likely candidates for satisfying the BPZ equations.

However, the non-unitary nature of the CFTs we are studying here introduces complexities not typically encountered in unitary theories. In non-unitary settings, the BPZ equations may not universally apply due to the fact that zero-norm states are not always null states.[8] Nevertheless, we would like to show that BPZ equations still apply to our particular case, i.e.,

$$\langle V_{\ell_1,m_1}(x_1)\dots\big(\mathcal{L}_{1,2\ell_k+1}V_{\ell_k,m_k}(x_k)\big)\dots V_{\ell_n,m_n}(x_n)\rangle = 0, \qquad (k=1,2,\dots,n), \qquad (46)$$

where the BPZ operator $\mathcal{L}_{1,2\ell_k+1}$ is given by

$$\mathcal{L}_{r,s} = \sum_{N=1}^{rs} \sum_{\substack{p_1 \geqslant \dots \geqslant p_N \geqslant 1 \\ p_1+\dots+p_N=rs}} A_{p_1,\dots,p_N}\, \mathcal{L}_{-p_1}\mathcal{L}_{-p_2}\dots\mathcal{L}_{-p_N}, \qquad (47)$$

where

$$\mathcal{L}_{-p}^{(k)} = \sum_{i \neq k}\left[\frac{(p-1)h_{\alpha_i}}{(x_i-x_k)^p} - \frac{1}{(x_i-x_k)^{p-1}}\frac{\partial}{\partial x_i}\right], \qquad (48)$$

and the explicit form of the coefficients $A_{p_1,\dots,p_N}$ is unimportant for this proof. Here $\mathcal{L}_{-p}^{(k)}$ means that the Virasoro generator $\mathcal{L}_{-p}$ acts on the operator at $x_k$.

The argument mainly relies on the following property of vertex operators (unscreened ones):

$$\mathcal{L}_{r,s}\mathcal{V}_{\alpha_{r,s}} \equiv 0 \qquad (r,s=1,2,3,\dots). \qquad (49)$$

This equation is a consequence of [23, theorem 3.2] (see also [24, proposition 3.2]). It is valid as long as the anomalous charge conservation law (33) is satisfied in the correlation function. The non-trivial part of this result is that it holds for any value of the central charge, including both unitary and non-unitary cases. We would like to point out that for each $(r,s)$, there is

---

[8]In unitary theories, one can argue that zero-norm states are always null states using the Cauchy-Schwarz inequality. But this argument fails in the case of non-unitary theories.

also another unscreened vertex operator $\mathcal{V}_{\alpha_{-r,-s}}$ having the same conformal dimension as $\mathcal{V}_{\alpha_{r,s}}$. However, acting with the BPZ operator on it leads to a primary operator $\mathcal{L}_{r,s}\mathcal{V}_{\alpha_{-r,-s}}$ which does not vanish in general. We will take (49) as a starting point and show (46) using an argument similar to [17].

Consider the integrand (40) of the correlation function (38). The vertex operators that constitute the integrand are summarized as follows:

When we act with the BPZ operator on $\mathcal{V}_{-\ell_k\alpha_-}(x_k)$, each $L_{-p}$ gives the following differential operator, denoted as $\widetilde{\mathcal{L}}_{-p}^{(k)}$:

$$\widetilde{\mathcal{L}}_{-p}^{(k)} = \mathcal{L}_{-p}^{(k)} + \sum_{r=1}^{N} \frac{\partial}{\partial z_r}\left[-\frac{1}{(z_r - x_k)^{p-1}}\right] - \frac{1}{(w - x_k)^{p-1}}\frac{\partial}{\partial w}, \tag{50}$$

where $\mathcal{L}_{-p}^{(k)}$ is exactly the same as in (48). If we use $\widetilde{\mathcal{L}}_{-p}^{(k)}$ to build the BPZ differential operator $\widetilde{\mathcal{L}}_{1,2\ell_k+1}$ by the same rule as $\mathcal{L}_{1,2\ell_k+1}$ is built out of $\mathcal{L}_{-p}^{(k)}$, then from (49) we know that acting with $\widetilde{\mathcal{L}}_{1,2\ell_k+1}$ on the integrand gives zero. However, for eq. (46) we need the BPZ differential operator $\mathcal{L}_{1,2\ell_k+1}$. It remains to show that the difference between $\widetilde{\mathcal{L}}_{1,2\ell_k+1}$ and $\mathcal{L}_{1,2\ell_k+1}$ does not contribute after we integrate out the $z_k$ variables of the screening charges.

To show this, we notice that in (50), $\mathcal{L}_{-p}^{(k)}$ only interacts with all $x_i$ ($i \neq k$), the second term only interacts with all $z_r$, and the last term only interacts with $w$ (so they mutually commute). This observation implies that the difference between $\mathcal{L}_{1,2\ell_k+1}$ and $\widetilde{\mathcal{L}}_{1,2\ell_k+1}$ has the following form:

$$\widetilde{\mathcal{L}}_{1,2\ell_k+1} = \mathcal{L}_{1,2\ell_k+1} + \sum_{r=1}^{N} \frac{\partial}{\partial z_r}\mathcal{A}_r + \mathcal{B}\frac{\partial}{\partial w}, \tag{51}$$

where, in the right-hand side of the equation, the first term corresponds to picking only $\mathcal{L}_{-p}^{(k)}$ in (50), the second term corresponds to picking at least one $\frac{\partial}{\partial z_r}$ term in (50), and the last term corresponds to picking only $\mathcal{L}_{-p}^{(k)}$ and at least one $\frac{\partial}{\partial w}$. $\mathcal{A}_r$ is a (finite-order) differential operator of the form

$$\mathcal{A}_r = \mathcal{A}_r(x, z, w, \partial_x, \partial_z, \partial_w),$$

and $\mathcal{B}$ is a (finite-order) differential operator of the form

$$\mathcal{B} = \mathcal{B}(x, w, \partial_x, \partial_w).$$

Now let us start with sufficiently large positive $\alpha_-^2$. In eq. (51), acting with the second term on the integrand gives a total derivative and does not contribute to the integral.[9] The third term in (51) does not involve $z_k$, so it acts directly on the integral. In section 3.2.1, we have shown that the integral is $w$-independent, so the last term in (51) also gives zero. Therefore, BPZ equation (46) is true for sufficiently large positive $\alpha_-^2$, and also true for any $\alpha_- \in \mathbb{C}$ by analytic continuation.

Table 1: Table of vertex operators that appear in the integrand of (38).

| Vertex operator | U(1) charge in $\alpha_{r,s}$ notation | Conformal dimension |
|---|---|---|
| $\mathcal{V}_{-\ell_i\alpha_-}(x_i)$ | $\alpha_{1,2\ell_i+1}$ | $h_{1,2\ell_i+1} = \alpha_-^2\ell_i(\ell_i+1) - \ell_i$ |
| $\mathcal{V}_{\alpha_-}(z_k)$ | $\alpha_{1,-1}$ | $h_{1,-1} = 1$ |
| $\mathcal{V}_{2\alpha_0}(w)$ | $\alpha_{-1,-1}$ | $h_{-1,-1} = 0$ |

---

[9]Here we used the condition that $\alpha_-^2$ is sufficiently large, so the boundary terms vanish.

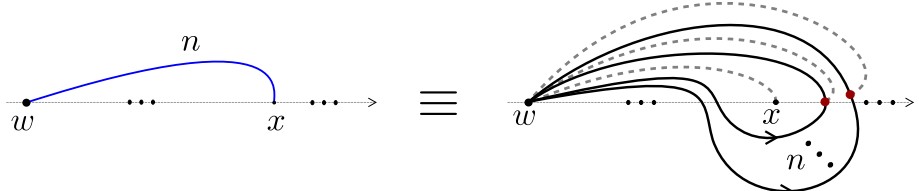

Figure 6: In this figure we introduce a shorthand graphical notation for the operators $V_{\ell,m}(x)$. The blue line represents the highest-weight operator $\mathcal{V}_{-\ell\alpha_-}$ together with all $F$ contours acting on it.

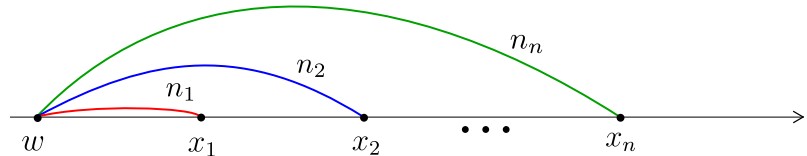

Figure 7: This figure shows correlation functions $\langle V_{\ell_1,m_1}(x_1)\ldots V_{\ell_n,m_n}(x_n)\rangle$ redrawn in the graphical notation of figure 6. The colors of the topological lines attached to operators match with the colors of the contours introduced in figure 5.

### 3.3 Ward identities and the construction of defect ending operators

In this section we derive the Ward identities for the correlation functions defined by (38), as well as show how the construction presented in this paper fits into the general framework of [1]. The Ward identities can be formulated as follows:

$$\langle X \cdot \left( V_{\ell_1,m_1}(x_1)\ldots V_{\ell_n,m_n}(x_n) \right) \rangle = 0, \tag{52}$$

where $X = H, E, F$ are the generators of $U_q(sl_2)$. As mentioned above, we do not have an explicit expression of the generator $E$. Nevertheless, in practice, using $H$ and $F$ we can generate all the independent Ward identities.

First of all, to make the figures simpler let us introduce a shorthand graphical notations for the operators $V_{\ell,m}(x)$, see figure 6. A single blue line encodes all the contours associated with the operator $V_{\ell,m}(x)$. With this notation we can identify the operators $V_{\ell,m}(x)$ with the defect-ending operators introduced in the companion paper [1]. The correlation functions of these operators can now also be redrawn in a schematic way, as it is shown in figure 7. Such notation makes clear that the Coulomb gas construction presented here gives a particular realization of lines studied in our companion paper [1], see for example figures in section 3 there. The refinement of the construction, however, is that now we discuss CFT correlation functions and the topological lines end at a single point $w$. As it was shown in section 3.2.1, the correlation functions are independent on the position of the point $w$.

Now, let us derive the Ward identities for $H$ and $F$. The rule of $H$-action on the operators (36) gives:

$$H \cdot \left( V_{\ell_1,m_1}(x_1)\ldots V_{\ell_n,m_n}(x_n) \right) = 2(m_1 + \ldots + m_n)V_{\ell_1,m_1}(x_1)\ldots V_{\ell_n,m_n}(x_n). \tag{53}$$

So the Ward identity for $H$ is

$$(m_1 + \ldots + m_n)\langle V_{\ell_1,m_1}(x_1)\ldots V_{\ell_n,m_n}(x_n)\rangle = 0. \tag{54}$$

This equality is an immediate consequence of the anomalous charge conservation law (33).

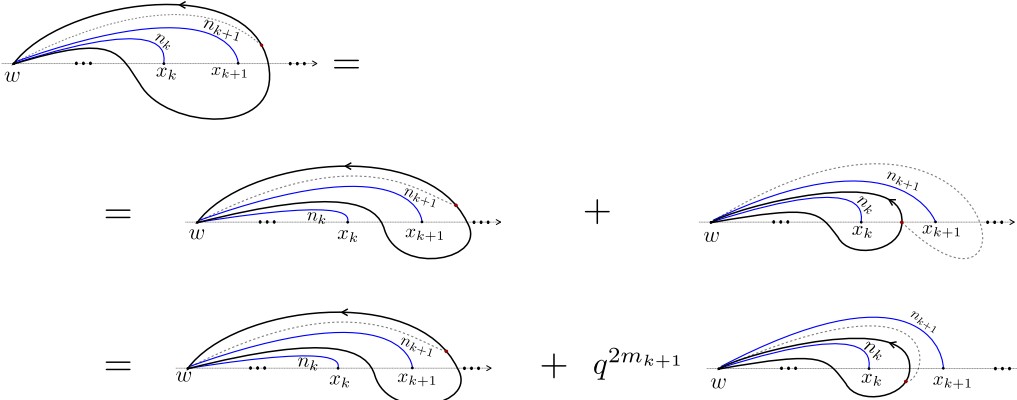

Figure 8: The action of $F$ on two operators is given by a correct $U_q(sl_2)$ coproduct: $\Delta(F) = 1 \otimes F + F \otimes q^H$. We get a phase of $q^{2m_{k+1}}$ from the reordering of the screening charge and the operator at $x_{k+1}$. Here we use the graphical notation introduced in figure 6. With this graphical notation the figure provides a concrete realization of the $U_q(sl_2)$ coproduct, which was introduced in figure 6 of [1].

To derive the Ward identity for $F$, we first need to show that the action of $F$ on several operators is governed by the $U_q(sl_2)$ coproduct, $\Delta(F) = F \otimes q^H + 1 \otimes F$. Recall that $F$ is identified with the contour integral of the screening charge $\int dz \mathcal{V}_{\alpha_-}(z)$. Fortunately, such an identification together with the monodromy properties of the integrand (40) automatically guarantees the correct quantum group coproduct. Let us show it by acting with $F$ on two operators. Figure 8 represents the action $F \cdot \left( V_{\ell_k, m_k}(x_k) V_{\ell_{k+1}, m_{k+1}}(x_{k+1}) \right)$, which corresponds to surrounding both operators with the $F$ line. Now we deform the contour to express the result of the action in terms of the basis correlation functions of figure 7. The first term on the RHS represents the part of the contour that goes around only the second operator $V_{\ell_{k+1}, m_{k+1}}(x_{k+1})$. For this contribution, we get immediately the same ordering as in the definition, and it contributes as $1 \otimes F$ would. As for the second term, one gets an extra phase from changing the free boson operator ordering (of which we keep track using the dotted lines, see figure 3). Note that the reordering needs to be done with all operators that constitute the operator $V_{\ell_{k+1}, m_{k+1}}(x_{k+1})$, which itself is a composite operator: it consists of a highest-weight vertex operator $\mathcal{V}_{-\ell_{k+1}\alpha_-}$ and $(\ell_{k+1} - m_{k+1})$ integrated vertex operators $\mathcal{V}_{\alpha_-}$.

Let us determine the phase. According to figure 3, for each of the vertex operators $\mathcal{V}_\beta$ that constitute the operator $V_{\ell_{k+1}, m_{k+1}}(x_{k+1})$ we get a phase of $e^{-2i\pi\alpha_-\beta}$. These sum up to

$$e^{-2i\pi\alpha_-^2(\ell_{k+1} - m_{k+1})}e^{-2i\pi\alpha_-^2(-\ell_{k+1})} = e^{2i\pi\alpha_-^2 m_{k+1}} = q^{2m_{k+1}}. \tag{55}$$

Where we have defined $q$ to be

$$q = e^{i\pi\alpha_-^2}. \tag{56}$$

This is exactly the action of $q^H$ on the operator $V_{\ell_2, m_2}$. Then the second term on the RHS of figure 8 gives a term $F \otimes q^H$ to the coproduct. Finally, we obtained the correct $U_q(sl_2)$ coproduct for generator $F$.

Our conventions for the choice of contours, ordering of operators and reality conditions were designed in a way that the choice of $q$ as in (56) together with $\alpha_-$ given by (27) coincides with the lattice $q$ given by (11). Also the conventions have been chosen so that the action of a screening charge reproduces the coproduct as in (2), rather than any other choice of coproduct that is compatible with the quantum group.

The generalization of the action of $F$ on $n > 2$ operators is straightforward. First, one has to surround all the operators with a contour integral of a screening charge (see figure 9). Then

one deforms the contour into a sum of basis contours, defined by figure 5. Finally, the phases come from reordering of the free boson operators and a choice of $q$ as in (56). The phases determine the coproduct to be

$$\Delta^{n-1}(F) = \sum_{i=1}^{n} 1_1 \otimes 1_2 \otimes \ldots \otimes 1_{i-1} \otimes F_i \otimes q^{H_{i+1}} \otimes q^{H_{i+2}} \otimes \ldots \otimes q^{H_n}. \qquad (57)$$

Now, when we have a proper action of $F$ on any number of operators, we can easily derive the Ward identities. Formally we have

$$\left\langle F \cdot \left(V_{\ell_1,m_1}(x_1) \ldots V_{\ell_n,m_n}(x_n)\right)\right\rangle = \oint dz \left\langle \mathcal{V}_{\alpha_-}(z) V_{\ell_1,m_1}(x_1) \ldots V_{\ell_n,m_n}(x_n)\right\rangle, \qquad (58)$$

where the integration contour surrounds all the other operators (see figure 9). Here we need to be careful because the correlation functions in the Coulomb gas formalism are multivalued functions in general, so superficially we cannot close the contour of integral (58). However, thanks to the anomalous conservation law (33), the integrand in the RHS of eq. (58) does not vanish only when $m_1 + \ldots + m_n = 1$. In this case, the corresponding integrand is

$$f(w, x_1, \ldots, x_n, z_1, \ldots, z_N) \times (z-w)^{2(\alpha_-^2-1)} \prod_{k=1}^{N}(z-z_k)^{2\alpha_-^2} \prod_{i=1}^{n}(z-x_i)^{-2\ell_i\alpha_-^2}, \qquad (59)$$

where the function $f$ is the same as the one in (40). Here $N = \ell_1 + \ldots + \ell_n - 1$ to satisfy the condition $m_1 + \cdots + m_n = 1$. In eq. (59), the total power of $z$ is equal to

$$2(\alpha_-^2 - 1) + 2N\alpha_-^2 - 2\alpha_-^2 \sum_{i=1}^{n} \ell_i = -2.$$

Consequently, (59) is analytic in $z$ in the domain

$$|z| > \max\{|w|, |x_1|, \ldots, |x_n|, |z_1|, \ldots, |z_n|\}.$$

Therefore, we are allowed to close the contour of $z$ and continuously deform the contour without changing the value of the integral. Because of the $z^{-2}$ behavior of the integrand, we deform the contour of $z$ to an infinitely large circle and conclude that the contour integral is equal to zero. This leads to the Ward identify for $F$:

$$\left\langle F \cdot \left(V_{\ell_1,m_1}(x_1) \ldots V_{\ell_n,m_n}(x_n)\right)\right\rangle = \oint dz \left\langle \mathcal{V}_{\alpha_-}(z) V_{\ell_1,m_1}(x_1) \ldots V_{\ell_n,m_n}(x_n)\right\rangle = 0. \qquad (60)$$

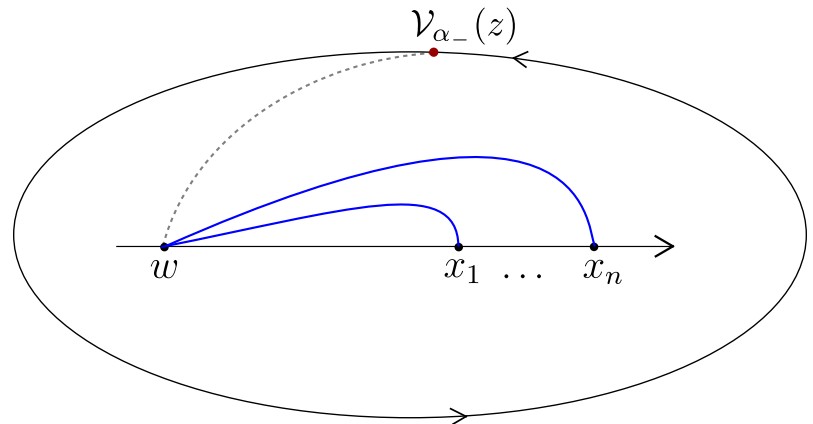

Figure 9: Contour integral representing the action $F \cdot \left(V_{\ell_1,m_1}(x_1) \ldots V_{\ell_n,m_n}(x_n)\right)$.

As demonstrated above, (60) holds in a nontrivial way when $m_1 + \ldots + m_n = 1$ and trivially holds when $m_1 + \ldots + m_n \neq 1$.

Now let us rewrite the contour integral (60) as a sum of correlation functions. Using (57) we immediately get the Ward identities for the correlation functions of the Coulomb gas normalized operators $V_{\ell,m}$:

$$\sum_{i=1}^{n} q^{2(m_{i+1}+m_{i+2}+\ldots+m_n)}\left\langle V_{\ell_1,m_1}(x_1)\ldots V_{\ell_i,m_i-1}(x_i)\ldots V_{\ell_n,m_n}(x_n)\right\rangle = 0\,. \tag{61}$$

After the canonical normalization of operators which is discussed in section 3.6, the Ward identities become exactly (5).

In summary, we have derived the Ward identities (52) for $X = H, F$, given by eqs. (54) and (61). The quantum group parameter $q$ is given by $q = e^{i\pi\alpha_-^2}$. These are the full set of independent Ward identities for $U_q(sl_2)$ global symmetry.[10] Moreover, the graphical notation used in figures 6, 7, 8 and 9 allows us to interpret the operators $V_{\ell,m}$ as defect-ending operators and provides a concrete realization of a general discussion of a companion paper [1].

### 3.4 Computations of two-point functions

We defined the correlation functions of a generic number of chiral operators $V_{\ell,m}(z)$ of generic quantum group spin $\ell$ in section 3.2. Now, we would like to compute the two-point functions explicitly starting from the definition (38). In the computations we use the $w$-independence property, which significantly simplifies the integrals. We show that the two-point functions of operators of any spin $\ell$ scale as $\sim 1/(y-x)^{2h}$ for the corresponding scaling dimension $h$. This is exactly the behaviour we expect from CFT correlators. However, because of the quantum group structure, we cannot normalize all the non-zero 2-point functions to 1. For example, the Ward identities (61) imply $\langle V_{1,1}V_{1,-1}\rangle = q^2\langle V_{1,-1}V_{1,1}\rangle = -\langle V_{1,0}V_{1,0}\rangle$. In the Coulomb gas construction above we do not choose the normalization. Nevertheless, we know that it must be consistent with the Ward identities.

**Spin-1/2**

Let us start with the computation of the spin-1/2 two-point functions. In fact, we do not have spin-1/2 operators in the spectrum of $XXZ_q$ spin chain (14), to which we want to apply the results. However, it turns out that the Coulomb Gas construction of section 3.2 gives well defined correlators also for half integer spin. Since correlators of spin-1/2 operators involve the simplest integrals, we start by evaluating them as a warm-up.

There are exactly two non-zero spin-1/2 two-point functions. We start with the computation of $\left\langle V_{\frac{1}{2},-\frac{1}{2}}V_{\frac{1}{2},\frac{1}{2}}\right\rangle$ directly from the definition (38). The picture corresponding to this correlation function is given by the LHS of figure 10. One can read off the corresponding integral,

$$\left\langle V_{\frac{1}{2},-\frac{1}{2}}(x)V_{\frac{1}{2},\frac{1}{2}}(y)\right\rangle = (y-x)^{\frac{\alpha_-^2}{2}}(x-w)^{1-\alpha_-^2}(y-w)^{1-\alpha_-^2}\int_C dz(z-x)^{-\alpha_-^2}(y-z)^{-\alpha_-^2}(z-w)^{2\alpha_-^2-2}\,. \tag{62}$$

According to the definition, the convention is that the integrand is real for $w < x < z < y$. The first step is to rewrite the integral over the contour $C$ as a line integral, i.e. the integral from

---

[10]This can be seen from the following counting. Let $\mathbb{V} = \mathbb{V}_{\ell_1} \otimes \ldots \otimes \mathbb{V}_{\ell_n}$ be the tensor product representation of $U_q(sl_2)$ for generic $q$. If we decompose $\mathbb{V}$ into irreducible representations, each irreducible representation contains one state with charge 0, i.e. $m_1 + m_2 + \ldots + m_n = 0$. Similarly, each nontrivial irreducible representation contains one state with charge 1. Using the Ward identities for $H$ and $F$, the number of independent correlation functions is equal to the number of charge-0 states minus the number of charge-1 states, which is equal to the number of trivial representations in the decomposition of $\mathbb{V}$, i.e. the number of $U_q(sl_2)$-invariant tensors.

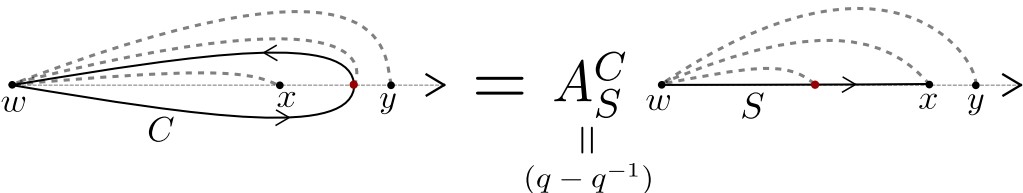

Figure 10: Correlation function of spin-1/2 operators $\left\langle V_{\frac{1}{2},-\frac{1}{2}}(x)V_{\frac{1}{2},\frac{1}{2}}(y)\right\rangle$. The dashed topological line configuration is such that when the integrated points are on the red dots the integrand is real.

$w$ to $x$ (see figure 10). The contour manipulation involved yields an overall constant factor, $A_S^C$, defined in appendix B. In this case it is equal to $[A_S^C]_1^{\frac{1}{2}} = (q-q^{-1})$.[11] Let us denote the line integral corresponding to the RHS of figure 10 as $\left\langle V_{\frac{1}{2},-\frac{1}{2}}(x)V_{\frac{1}{2},\frac{1}{2}}(y)\right\rangle_S$.

Then we can write

$$\left\langle V_{\frac{1}{2},-\frac{1}{2}}(x)V_{\frac{1}{2},\frac{1}{2}}(y)\right\rangle = [A_S^C]_1^{\frac{1}{2}}\left\langle V_{\frac{1}{2},-\frac{1}{2}}(x)V_{\frac{1}{2},\frac{1}{2}}(y)\right\rangle_S, \tag{63}$$

where

$$\left\langle V_{\frac{1}{2},-\frac{1}{2}}(x)V_{\frac{1}{2},\frac{1}{2}}(y)\right\rangle_S = (y-x)^{\frac{\alpha_-^2}{2}}(x-w)^{1-\alpha_-^2}(y-w)^{1-\alpha_-^2}\int_w^x dz(x-z)^{-\alpha_-^2}(y-z)^{-\alpha_-^2}(z-w)^{2\alpha_-^2-2}. \tag{64}$$

Note that for the line integral the terms of the integrand are organized in such an order that it is real for $w < z < x < y$. Now we use the $w$-independence property: the integral simplifies significantly if we send $w \to -\infty$. In this case we can get rid of all the brackets containing $w$:

$$\left\langle V_{\frac{1}{2},-\frac{1}{2}}(x)V_{\frac{1}{2},\frac{1}{2}}(y)\right\rangle_S = (y-x)^{\frac{\alpha_-^2}{2}}\int_{-\infty}^x dz(x-z)^{-\alpha_-^2}(y-z)^{-\alpha_-^2}. \tag{65}$$

The integral is convergent for $\frac{1}{2} < \alpha_-^2 < 1$, i.e. it is convergent for all physical values of $\alpha_-^2$. To evaluate the integral we perform the change of variables $z = x - (y-x)\tau$, so $\tau \in [0, +\infty)$:

$$\left\langle V_{\frac{1}{2},-\frac{1}{2}}(x)V_{\frac{1}{2},\frac{1}{2}}(y)\right\rangle_S = \underbrace{(y-x)^{1-\frac{3\alpha_-^2}{2}}}_{\text{correct scaling}}\underbrace{\int_0^{+\infty} d\tau \cdot \tau^{-\alpha_-^2}(1+\tau)^{-\alpha_-^2}}_{=I,\text{ numerical factor}}. \tag{66}$$

We obtain the correct scaling behaviour times some normalization factor $I$. Let us compute this factor. There are no singularities along the integration path and we can safely perform the change of variables $p = \frac{\tau}{1+\tau}$, where $p \in [0, 1]$. The inverse mapping is $\tau = \frac{p}{1-p}$.

$$I = \int_0^1 dp \cdot p^{-\alpha_-^2}(1-p)^{2\alpha_-^2-2}. \tag{67}$$

The expression for $I$ is exactly the integral expression for the Euler beta function (A.2) with parameters $a = 1-\alpha_-^2$, $b = 2\alpha_-^2 - 1$:

$$I = B(1-\alpha_-^2, 2\alpha_-^2 - 1) = \frac{\Gamma(1-\alpha_-^2)\Gamma(2\alpha_-^2-1)}{\Gamma(\alpha_-^2)}. \tag{68}$$

---

[11]Note that since for a spin-1/2 two-point function there is just a single integral, the contour manipulation is equivalent to the one in figure 4. In other words this is a special case when the constants $[A_S^C]_1^{\frac{1}{2}}$ and $[A_L^C]_1^{\frac{1}{2}}$ defined in appendix B coincide. This will not be the case for higher spins.

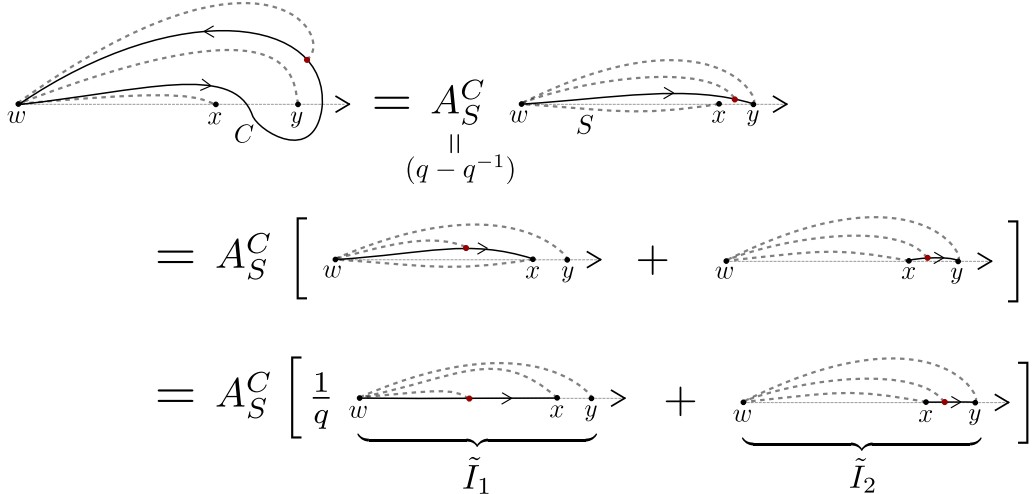

Figure 11: Correlator of spin-1/2 operators $\left\langle V_{\frac{1}{2},\frac{1}{2}}(x)V_{\frac{1}{2},-\frac{1}{2}}(y)\right\rangle$. The dashed topological line configuration tells us what phase corresponds to the integrand. The topological line configurations in the third row are such that the red points denote configurations of variables for which the integrand is real.

Combining everything together we get the result for the two-point function:

$$\left\langle V_{\frac{1}{2},-\frac{1}{2}}(x)V_{\frac{1}{2},\frac{1}{2}}(y)\right\rangle = (q-q^{-1})\cdot(y-x)^{1-\frac{3\alpha_{-}^2}{2}}\cdot\frac{\Gamma(1-\alpha_{-}^2)\Gamma(2\alpha_{-}^2-1)}{\Gamma(\alpha_{-}^2)}\,. \tag{69}$$

The second non-zero two-point function is $\left\langle V_{\frac{1}{2},\frac{1}{2}}(x)V_{\frac{1}{2},-\frac{1}{2}}(y)\right\rangle$. If we just wanted to know its value, we could simply use the Ward identity combined with the result above (69). However, since this is a warmup, we evaluate it from first principles as well. As before, we start with the definition (38). It provides us with the LHS of figure 11, from which we can write the integral expression:

$$\left\langle V_{\frac{1}{2},\frac{1}{2}}(x)V_{\frac{1}{2},-\frac{1}{2}}(y)\right\rangle = (y-x)^{\frac{\alpha_{-}^2}{2}}(x-w)^{1-\alpha_{-}^2}(y-w)^{1-\alpha_{-}^2}\int_{\widetilde{C}}dz(z-x)^{-\alpha_{-}^2}(z-y)^{-\alpha_{-}^2}(z-w)^{2\alpha_{-}^2-2}\,. \tag{70}$$

In this expression the integrand is real when $w < x < y < z$. We can start by writing the integral over the contour $\widetilde{C}$ as a line integral from $w$ to $y$ (see first line of figure 11). We get the same coefficient as we got when we converted the contour to a line integral in the computation above, $[A_S^C]_1^{\frac{1}{2}} = (q-q^{-1})$. Then, we can split it into two pieces and reinterpret it as the sum of two line integrals – one from $w$ to $x$, and another from $x$ to $y$. We would like to import the result of the previous section for the first segment, from $w$ to $x$. However, we have to be a bit careful since the topological line configuration is not the same. This gives rise to the extra factor of $1/q$ between the second and third lines of figure 11. We denote the line integral in the region $[w,x]$ as $\widetilde{I}_1$ and in the region $[x,y]$ as $\widetilde{I}_2$. Then $\widetilde{I}_1$ is exactly the integral we computed before, $\widetilde{I}_1 = \left\langle V_{\frac{1}{2},-\frac{1}{2}}(x)V_{\frac{1}{2},\frac{1}{2}}(y)\right\rangle_S$, and we are left with computing $\widetilde{I}_2$.

Then we can write

$$\left\langle V_{\frac{1}{2},\frac{1}{2}}(x)V_{\frac{1}{2},-\frac{1}{2}}(y)\right\rangle = [A_S^C]_1^{\frac{1}{2}}(q^{-1}\widetilde{I}_1 + \widetilde{I}_2)\,. \tag{71}$$

Let us now compute $\widetilde{I}_2$. In the limit $w \to -\infty$ we again get a significant simplification of the integrand

$$\widetilde{I}_2 = (y-x)^{\frac{\alpha_{-}^2}{2}}\int_x^y dz(z-x)^{-\alpha_{-}^2}(y-z)^{-\alpha_{-}^2}\,. \tag{72}$$

To evaluate the integral we perform the change of variables $z = (1-t)x + ty$:

$$\widetilde{I}_2 = \underbrace{(y-x)^{1-\frac{3\alpha_-^2}{2}}}_{\text{correct scaling}} \underbrace{\int_0^1 dt \cdot t^{-\alpha_-^2}(1-t)^{-\alpha_-^2}}_{=\widetilde{I}, \text{ numerical factor}} . \tag{73}$$

Computation of the numerical factor again yields the Euler Beta function (A.2) with parameters $a = b = 1 - \alpha_-^2$:

$$\widetilde{I} = \frac{\Gamma(1-\alpha_-^2)^2}{\Gamma(2-2\alpha_-^2)} . \tag{74}$$

Combining everything together, from (71) we get

$$\begin{aligned}
\left\langle V_{\frac{1}{2},\frac{1}{2}}(x) V_{\frac{1}{2},-\frac{1}{2}}(y) \right\rangle &= [A_S^C]_1^{\frac{1}{2}}(y-x)^{1-\frac{3\alpha_-^2}{2}}\left( q^{-1}\frac{\Gamma(1-\alpha_-^2)\Gamma(2\alpha_-^2-1)}{\Gamma(\alpha_-^2)} + \frac{\Gamma(1-\alpha_-^2)^2}{\Gamma(2-2\alpha_-^2)} \right) \\
&= [A_S^C]_1^{\frac{1}{2}}(y-x)^{1-\frac{3\alpha_-^2}{2}}\frac{\Gamma(1-\alpha_-^2)\Gamma(2\alpha_-^2-1)}{\Gamma(\alpha_-^2)} \\
&\quad \times \underbrace{\left[ e^{-\pi i\alpha_-^2} + \frac{\Gamma(1-\alpha_-^2)\Gamma(\alpha_-^2)}{\Gamma(2-2\alpha_-^2)\Gamma(2\alpha_-^2-1)} \right]}_{\xi} .
\end{aligned} \tag{75}$$

To simplify the factor $\xi$ we use the relations (A.1) for the $\Gamma$-functions so that

$$\begin{aligned}
\Gamma(1-\alpha_-^2)\Gamma(\alpha_-^2) &= \frac{\pi}{\sin(\pi\alpha_-^2)}, \\
\Gamma(2-2\alpha_-^2)\Gamma(2\alpha_-^2-1) &= \frac{\pi}{\sin\left(\pi(2\alpha_-^2-1)\right)},
\end{aligned} \tag{76}$$

then

$$\xi = e^{-i\pi\alpha_-^2} - 2\cos(\pi\alpha_-^2) = -e^{i\pi\alpha_-^2} = -q . \tag{77}$$

Finally, we get the expression for the second non-zero two-point correlation function:

$$\left\langle V_{\frac{1}{2},\frac{1}{2}}(x) V_{\frac{1}{2},-\frac{1}{2}}(y) \right\rangle = -q(q-q^{-1}) \cdot (y-x)^{1-\frac{3\alpha_-^2}{2}} \cdot \frac{\Gamma(1-\alpha_-^2)\Gamma(2\alpha_-^2-1)}{\Gamma(\alpha_-^2)} . \tag{78}$$

Here we simplified the expression for the two-point correlation function so that it is easy to compare it with the first non-zero spin-1/2 correlation function (69).

*Remark* 3.2. Note that from dimensional analysis one expects the same scaling behaviour of the correlators. From Coulomb gas formalism the conformal dimensions of operators $V_{\frac{1}{2},\pm\frac{1}{2}}$ are equal to $h = \frac{3}{4}\alpha_-^2 - \frac{1}{2}$, so that the coordinate dependence should look like

$$\left\langle V_{\frac{1}{2},\pm\frac{1}{2}}(x) V_{\frac{1}{2},\mp\frac{1}{2}}(y) \right\rangle \sim \frac{1}{(y-x)^{2h}} = (y-x)^{1-\frac{3}{2}\alpha_-^2} , \tag{79}$$

which is true for both spin-1/2 two-point functions (69) and (78).

*Remark* 3.3. Using the explicit expressions for the two-point correlation functions (69) and (78) one can check that they satisfy the Ward identities:

$$q\left\langle V_{\frac{1}{2},-\frac{1}{2}}(x) V_{\frac{1}{2},\frac{1}{2}}(y) \right\rangle + \left\langle V_{\frac{1}{2},\frac{1}{2}}(x) V_{\frac{1}{2},-\frac{1}{2}}(y) \right\rangle = 0 , \tag{80}$$

which is exactly what is expected from (61)

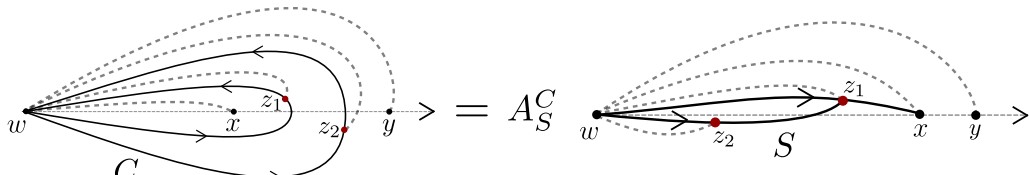

Figure 12: Correlator of spin-1 operators $\langle V_{1,-1}(x)V_{1,1}(y)\rangle$. The integrand is real when both red points are on the real line.

**Spin-1**

Before considering the generic spin-$\ell$ two-point function let us provide an explicit computation for a spin-1 two-point function. After this computation, it will be easy to generalize the result to arbitrary $\ell$. We do not compute all the non-zero two-point functions of spin-1 (there are three of them). Instead, we provide the calculation in the simplest case of "lowest-weight – highest-weight" configuration of operators. The rest of the non-zero two-point functions can be in principle computed directly or obtained using the quantum group Ward identities (61). One should also note that for spin $\geq 1$ some of the intermediate steps in the computation involve formally divergent integrals and one has to regularize them. However, these integrals can be defined unambiguously via analytic continuation. For a discussion of the regularization of these integrals see appendix B.

Let us compute the correlation function $\langle V_{1,-1}(x)V_{1,1}(y)\rangle$. We call it the "lowest-weight – highest-weight" configuration of operators because the first operator corresponds to a lowest-weight state in the spin-1 quantum group multiplet and is annihilated by $F$, similarly, the second operator is of highest weight. We start from the definition (38), which gives us the LHS of figure 12 and a corresponding integral expression:

$$\langle V_{1,1}(x)V_{1,-1}(y)\rangle = (y-x)^{2\alpha_-^2}(x-w)^{2-2\alpha_-^2}(y-w)^{2-2\alpha_-^2} \tag{81}$$
$$\times \int_C dz_1 dz_2 \Big[ (z_2-z_1)^{2\alpha_-^2}(z_1-x)^{-2\alpha_-^2}(y-z_1)^{-2\alpha_-^2}$$
$$\times (z_1-w)^{2\alpha_-^2-2}(z_2-x)^{-2\alpha_-^2}(y-z_2)^{-2\alpha_-^2}(z_2-w)^{2\alpha_-^2-2} \Big].$$

Here the integrand is real for $w < x < z_1 < z_2 < y$. The first step is the same as for the spin-1/2 two-point function, we rewrite the integral over the contour $C$ as a line integral (see the RHS of figure 12), such that the integration domain is $w < z_2 < z_1 < x$. At this point one can notice the difference with the integral representation on the RHS of figure 4: the integration domain for $z_2$ is different. For the purpose of computing the two-point function we are using the contour manipulation as shown in figure 12. The transition to such a representation yields a coefficient of $A_S^C$ (see appendix B), which in this case is equal to

$$[A_S^C]_2^1 = (q^2 - q^{-2})^2. \tag{82}$$

Let us denote the line integral as $\langle V_{1,-1}(x)V_{1,1}(y)\rangle_S$. In the $w \to -\infty$ limit the line integral simplifies to

$$\langle V_{1,-1}(x)V_{1,1}(y)\rangle_S = (y-x)^{2\alpha_-^2} \int_{-\infty}^x dz_1 \int_{-\infty}^{z_1} dz_2 (z_1-z_2)^{2\alpha_-^2}(x-z_1)^{-2\alpha_-^2}$$
$$\times (y-z_1)^{-2\alpha_-^2}(x-z_2)^{-2\alpha_-^2}(y-z_2)^{-2\alpha_-^2}, \tag{83}$$

where the integrand is now defined to be real for $w < z_2 < z_1 < x < y$. Next, we perform the change of variables $x - z_1 = t_1$ and $x - z_2 = t_2$, and then $\tau_1 = \frac{t_1}{y-x}, \tau_2 = \frac{t_2}{y-x}$, which results

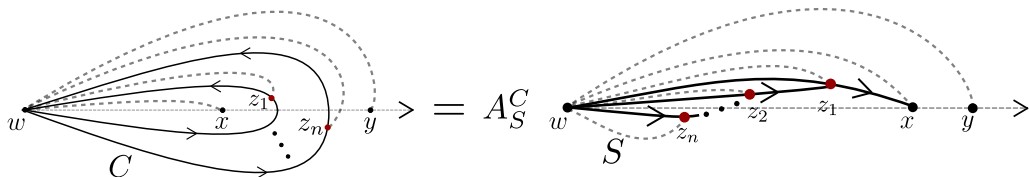

Figure 13: Correlator of spin-$\ell$ operators $\langle V_{\ell,-\ell}(x)V_{\ell,\ell}(y)\rangle$. The red points denote a configuration of variables for which the integrand is real.

in,

$$\langle V_{1,-1}(x)V_{1,1}(y)\rangle_S \tag{84}$$
$$= \underbrace{(y-x)^{2-4\alpha_-^2}}_{\text{correct scaling}} \underbrace{\int_0^{+\infty} d\tau_1 \int_{\tau_1}^{+\infty} d\tau_2 (\tau_2-\tau_1)^{2\alpha_-^2} \tau_1^{-2\alpha_-^2} \tau_2^{-2\alpha_-^2}(1+\tau_1)^{-2\alpha_-^2}(1+\tau_2)^{-2\alpha_-^2}}_{=I,\text{ numerical factor}}.$$

As before, we are left with computing the numerical factor $I$. To compute it we perform the change of variables $p_1 = \frac{\tau_1}{1+\tau_1}, p_2 = \frac{\tau_2}{1+\tau_2}$:

$$I = \int_0^1 dp_1 \int_{p_1}^1 dp_2 (p_2-p_1)^{2\alpha_-^2} p_1^{-2\alpha_-^2} p_2^{-2\alpha_-^2}(1-p_1)^{2\alpha_-^2-2}(1-p_2)^{2\alpha_-^2-2}, \tag{85}$$

which is exactly the well known Selberg integral defined in (A.3) with parameters $\alpha = -\beta = 1-2\alpha_-^2$, $\gamma = \alpha_-^2$:

$$I = \frac{1}{2}S_2(1-2\alpha_-^2, 2\alpha_-^2-1, \alpha_-^2). \tag{86}$$

Note that while the Selberg integral is formally divergent for the physical values of the parameters, the finite piece is unambiguous and can be extracted by analytic continuation from the region of parameters where the integral converges. It is given by a product of Gamma functions (A.3). Finally we get the expression for the spin-1 two-point function

$$\langle V_{1,-1}(x)V_{1,1}(y)\rangle = (q^2-q^{-2})^2 \cdot (y-x)^{2-4\alpha_-^2} \cdot \frac{1}{2}S_2(1-2\alpha_-^2, 2\alpha_-^2-1, \alpha_-^2). \tag{87}$$

Any other spin-1 two-point function is related to the one above by Ward identities (61).

**Spin-$\ell$**

Now we are ready to consider the generic spin-$\ell$ two-point function. We provide explicit computations only for the "lowest-weight – highest-weight" configuration of operators. Then, any other non-zero two-point functions can be found using the quantum group Ward identities.

Let us compute the correlation function $\langle V_{\ell,-\ell}(x)V_{\ell,\ell}(y)\rangle$. We start directly from a line integral as shown in the RHS of figure 13. The integration domain for such integral is $\mathcal{D} = \{z_1, \ldots, z_n : w < z_n < \ldots < z_1 < x\}$. It gives

$$\langle V_{\ell,-\ell}(x)V_{\ell,\ell}(y)\rangle = [A_S^C]_{2\ell}^\ell \langle V_{\ell,-\ell}(x)V_{\ell,\ell}(y)\rangle_S. \tag{88}$$

The expression for the line integral directly in the $w \to -\infty$ limit is

$$\langle V_{\ell,-\ell}(x)V_{\ell,\ell}(y)\rangle_S \tag{89}$$
$$= (y-x)^{2\ell^2\alpha_-^2} \int_{-\infty}^x dz_1 \int_{-\infty}^{z_1} dz_2 \ldots \int_{-\infty}^{z_{n-1}} dz_n \prod_{i<j}(z_i-z_j)^{2\alpha_-^2} \prod_{k=1}^n (x-z_k)^{-2\ell\alpha_-^2}(y-z_k)^{-2\ell\alpha_-^2},$$

so that the integrand is real for $w < z_n < \ldots < z_1 < x < y$. As for the spin-1 case, the line integral is formally divergent, but its finite piece is unambiguous. Then, we change integration variables to $x - z_k = t_k$. Remembering that $n = 2\ell$ we get:

$$
\left\langle V_{\ell,-\ell}(x) V_{\ell,\ell}(y) \right\rangle_S \tag{90}
$$
$$
= (y-x)^{-2\ell^2 \alpha_-^2} \int_0^{+\infty} dt_1 \int_{t_1}^{+\infty} dt_2 \ldots \int_{t_{n-1}}^{+\infty} dt_n \prod_{i<j} (t_j - t_i)^{2\alpha_-^2} \prod_{k=1}^{n} t_k^{-2\ell\alpha_-^2} \left(1 + \frac{t_k}{y-x}\right)^{-2\ell\alpha_-^2}.
$$

The second change of variables $\tau_k = \frac{t_k}{y-x}$ gives:

$$
\left\langle V_{\ell,-\ell}(x) V_{\ell,\ell}(y) \right\rangle_S \tag{91}
$$
$$
= \underbrace{(y-x)^{2\ell[1-\alpha_-^2(\ell+1)]}}_{\text{correct scaling}} \underbrace{\int_0^{+\infty} d\tau_1 \int_{\tau_1}^{+\infty} d\tau_2 \ldots \int_{\tau_{n-1}}^{+\infty} d\tau_n \prod_{i<j}(\tau_j - \tau_i)^{2\alpha_-^2} \prod_{k=1}^{n} \tau_k^{-2\ell\alpha_-^2}(1+\tau_k)^{-2\ell\alpha_-^2}}_{=I, \text{ numerical factor}}.
$$

We are left with computation of the numerical factor $I$. After the change of variables $p_k = \frac{\tau_k}{1+\tau_k}$ we get

$$
I = \int_0^1 dp_1 \int_{p_1}^1 dp_2 \ldots \int_{p_{n-1}}^1 dp_n \prod_{i<j}(p_j - p_i)^{2\alpha_-^2} \prod_{k=1}^{n} p_k^{-2\ell\alpha_-^2}(1-p_k)^{2\alpha_-^2-2}, \tag{92}
$$

which is the Selberg integral (A.3) with parameters $\alpha = 1 - 2\ell\alpha_-^2, \beta = 2\alpha_-^2 - 1, \gamma = \alpha_-^2$. Hence, the numerical factor is given by

$$
I = \frac{1}{(2\ell)!} S_{2\ell}(1 - 2\ell\alpha_-^2, 2\alpha_-^2 - 1, \alpha_-^2). \tag{93}
$$

Combining everything together we get an explicit expression for the generic spin-$\ell$ two-point function for the "lowest-weight – highest-weight" configuration of operators:

$$
\left\langle V_{\ell,-\ell}(x) V_{\ell,\ell}(y) \right\rangle = \frac{[A_S^C]_{2\ell}^\ell}{(2\ell)!} S_{2\ell}(1 - 2\ell\alpha_-^2, 2\alpha_-^2 - 1, \alpha_-^2) \cdot (y-x)^{2\ell[1-\alpha_-^2(\ell+1)]}, \tag{94}
$$

where the expression for $[A_S^C]_{2\ell}^\ell$ is computed in appendix B,

$$
[A_S^C]_{2\ell}^\ell = \prod_{j=0}^{2\ell-1} \left(q^{2\ell-2j} - q^{-2\ell}\right) \prod_{k=1}^{2\ell} \frac{1-q^{2k}}{1-q^2}. \tag{95}
$$

The numerical factors in front of the coordinate dependence in (94) are inherited from the Coulomb gas definition (38). It is impossible to normalize all operators such that the two-point functions take the form $1/(y-x)^{2h}$, because they have to satisfy the Ward identities. However, we will normalize them to a canonical form in section 3.6. Let us plug in all the factors in (94). After simplification, we get the final result for the two-point function of lowest-weight – highest-weight configuration of operators:

$$
\left\langle V_{\ell,-\ell}(x) V_{\ell,\ell}(y) \right\rangle = \left[ \frac{(2\pi i)^{2\ell}}{(2\ell)!} \prod_{k=1}^{2\ell} \frac{\Gamma\left((k+1)\alpha_-^2 - 1\right)\Gamma(-\alpha_-^2)}{\Gamma\left(k\alpha_-^2\right)^2 \Gamma(-k\alpha_-^2)} \right] \frac{1}{(y-x)^{2\ell[\alpha_-^2(\ell+1)-1]}}. \tag{96}
$$

*Remark* 3.4. Note that from dimensional analysis one expects the same scaling behaviour of the correlators. From Coulomb gas formalism we expect the conformal dimensions of operators $V_{\ell,\pm\ell}$ to be

$$
h_{1,2\ell+1} = \alpha_{1,2\ell+1}^2 - 2\alpha_0 \alpha_{1,2\ell+1} = \ell[-1 + \alpha_-^2(\ell+1)], \tag{97}
$$

so that the coordinate dependence of two-point function should look like

$$\left\langle V_{\ell,-\ell}(x)V_{\ell,\ell}(y)\right\rangle \sim \frac{1}{(y-x)^{2h}} = (y-x)^{2\ell[1-\alpha_-^2(\ell+1)]}, \tag{98}$$

which is exactly what we get from (94).

## 3.5 Computing OPE coefficients for Coulomb gas operators

The arguments presented in sections 3.2.1, 3.2.2 and 3.3 establish that the correlation functions of $V_{\ell,m}$ are (a) well-defined ($w$-independent), (b) satisfy BPZ equations and (c) respect $U_q(sl_2)$ symmetry. However, these properties are also satisfied by kinematical functions. The question is: how do we know that eq. (38) defines actual CFT correlation functions rather than kinematical functions? The answer to this question is that the functions defined by (38) can be computed using operator product expansion (OPE). We consider OPE between two neighboring operators $V_{\ell_k,m_k}(x_k)$ and $V_{\ell_{k+1},m_{k+1}}(x_{k+1})$:

$$V_{\ell_k,m_k}(x_k)\, V_{\ell_{k+1},m_{k+1}}(x_{k+1}) = \sum_{\ell=|m_k+m_{k+1}|}^{\ell_k+\ell_{k+1}} \frac{\widetilde{C}^{(\ell,m_k+m_{k+1})}_{(\ell_k,m_k),(\ell_{k+1},m_{k+1})}}{(x_{k+1}-x_k)^{h_{1,2\ell+1}+h_{1,2\ell_{k+1}+1}-h_{1,2\ell+1}}} \left[V_{\ell,m_k+m_{k+1}}(x_k)+\ldots\right], \tag{99}$$

with $\widetilde{C}^{(\ell,m_k+m_{k+1})}_{(\ell_k,m_k),(\ell_{k+1},m_{k+1})}$ being the corresponding OPE coefficient. Note that in contrast to the notations used in (7), the Coulomb gas OPE coefficient also carries the information about the quantum Clebsch-Gordan coefficients and the normalization of operators. The descendant terms denoted by "..." in the expansion are fixed by conformal invariance.

In this subsection, we would like to justify the above expansion in the correlation functions. Since the correlation functions are conformal, it suffices to show that the coefficient $\widetilde{C}^{(\ell,m_k+m_{k+1})}_{(\ell_k,m_k),(\ell_{k+1},m_{k+1})}$ is universal, regardless of other operator insertions. We will demonstrate this universality in 4 steps. The argument will also provide an algorithm to compute the OPE coefficients. Our method allows the computation of generic OPE coefficients, and was tested for various values of $m$ and $\ell$ both for self consistency (with quantum group symmetry) and versus the result of crossing (17). While for self consistency checks one may want to work with general $m$ and $\ell$, extraction of the dynamical information only requires the computation of one representative OPE coefficient for every choice of 3 multiplets. Then, the other OPE coefficients can be obtained using the quantum group Ward identities for the Coulomb gas three-point functions (61). To be specific, we will see that the (quite cumbersome) final formula drastically simplifies for the OPE coefficient of the form $\widetilde{C}^{(\ell_3,\ell_3)}_{(\ell_1,\ell_3-\ell_2),(\ell_2,\ell_2)}$ (one generic operator and two highest-weight operators).

### 3.5.1 Contour manipulations

In this subsection we deform the contour integrals in the LHS of eq. (99) to the contour integrals in the RHS. We realize the deformation in four steps. For convenience, below we will not be explicit in the position dependence of the operators. Nonetheless, we should keep in mind that the operator $V_{\ell_k,m_k}$ is inserted at $x_k$.

**Step 1**
In the first step our goal is to trade all of the contours acting on the second operator with contours acting on either only the first operator or the two together (see figure 14). Since the screening charges acting on the two operators together can be taken to act after the OPE, this will effectively reduce the problem to the computation of the OPE between one generic operator and one highest-weight operator.

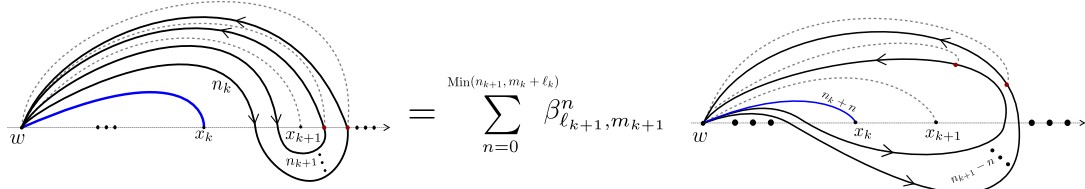

Figure 14: The first step of contour manipulation. The blue contours encode all contours involved in one operator and are defined in figure 6. We trade the contours that go around $x_{k+1}$ with contours either around $x_k$ or around both operators. Here $n_k = (\ell_k - m_k)$ is the number of lines going around point $x_k$ and defining the operator $V_{\ell_k, m_k}$, same for $n_{k+1}$. Contours that envelop both operators can be commuted with the OPE and as a result are easier to deal with.

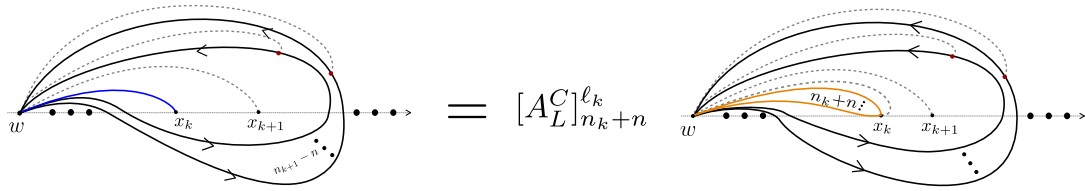

Figure 15: Working term by term in the RHS of eq. (101) and leaving the contours acting on both operators together untouched, we replace the contours acting on the operator at $x_k$ with linear contours (See figure 4 for a definition of the orange contour and figure 6 for a definition of the blue contours).

In practice, these contour moves are equivalent to using the coproduct of $F$, given by $\Delta(F) = F \otimes q^H + 1 \otimes F$. Applying it to the operators $V_{\ell_k, m_k+1} V_{\ell_{k+1}, m_{k+1}}$, we have

$$V_{\ell_k, m_k} V_{\ell_{k+1}, m_{k+1}} = F\left(V_{\ell_k, m_k} V_{\ell_{k+1}, m_{k+1}+1}\right) - q^{2(m_{k+1}+1)} V_{\ell_k, m_k-1} V_{\ell_{k+1}, m_{k+1}+1}. \tag{100}$$

We see that in the RHS of the equation, the $m$ index of the $k+1$-th operator is increased by 1. In terms of the contours it means that the second operator has one contour less going around it. Repeating this procedure for $(\ell_{k+1} - m_{k+1})$ times, we get

$$V_{\ell_k, m_k} V_{\ell_{k+1}, m_{k+1}} = \sum_{n=0}^{\min(\ell_{k+1}-m_{k+1}, m_k+\ell_k)} \beta^n_{\ell_{k+1}, m_{k+1}} F^{\ell_{k+1}-m_{k+1}-n}\left(V_{\ell_k, m_k-n} V_{\ell_{k+1}, \ell_{k+1}}\right). \tag{101}$$

Since the coefficients that appear in (100) only depend on the quantum numbers of $V_{\ell_k, m_k}$ and $V_{\ell_{k+1}, m_{k+1}}$, so do the coefficients $\beta^n_{\ell_{k+1}, m_{k+1}}$ in (101). In terms of contour manipulations, (101) is expressed in figure 14. The coefficients $\beta^n_{\ell_{k+1}, m_{k+1}}$ can be given in terms of the generating function,

$$\beta^n_{\ell, m} = \frac{1}{n!}\left[\partial_u^n \prod_{j=m}^{\ell-1}\left(1 - u q^{2(j+1)}\right)\right]_{u=0}. \tag{102}$$

This finishes the first step.

**Step 2**

In the second step we deform the contours that go around the point $x_k$ and constitute the operator $V_{\ell_k, m_k-n}$ to the line integral representation – a collection of straight contours from $w$ to $x_k$, as shown in figure 15. Recall that we have already done this step for a single operator in order to show the termination of representations (see figure 4). These contour deformations can be

done one by one from the innermost contour to the outermost contour. Technical details and the constants relating different representations are given in appendix B. The contribution of this step to the OPE is a factor of $[A_L^C]_{\ell_k-m_k+n}^{\ell_k}$, to each of the terms in eq. (101) ($n$ is the same as in (101)). This factor is given by eq. (35).

**Step 3**

In the previous steps we reduced the contour integral to the two types of lines: the lines going from $w$ to $x_k$ and the lines surrounding both operators (see the RHS of figure 15). In the third step, we would like to replace the lines from $w$ to $x_k$ either by the action of $F$ on both operators together, or by the lines stretching between $x_k$ and $x_{k+1}$. In the figures representing this step we are omitting $F$ lines acting on the two operators together. To perform this step, we repeatedly use the following contour manipulation identity: assume there are $h$ line integrals between $w$ and $x_k$ and $j$ line integrals between $x_k$ and $x_{k+1}$, then (see figure 4 for the topological line configuration implied by the orange contours),

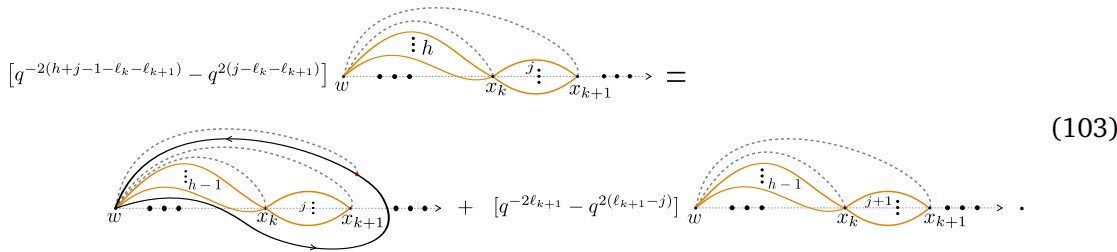

$$(103)$$

Using this contour manipulation we get the following identification,

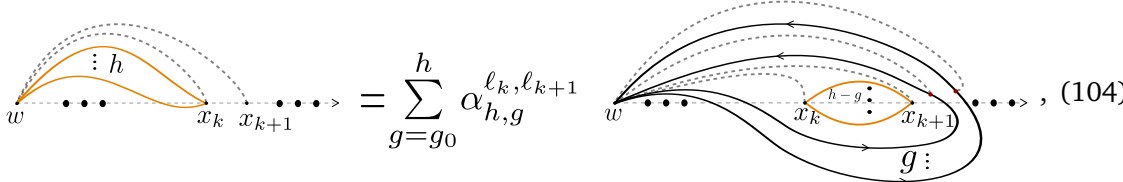

$$(104)$$

with $g_0 \equiv \max(0, h - 2\ell_{k+1})$.[12] Recall that this is the treatment of a single term from the RHS of (101) (or figure 14), with the identification

$$h = n_k + n = \ell_k - m_k + n. \tag{105}$$

As we will see in step 4, terms with different values of $h - g = \ell_k - m_k + n - g$ contribute to different primaries appearing in the OPE. Each primary is accompanied by the action of $g + n_{k+1} - n = g + \ell_{k+1} - m_{k+1} - n$ lowering operators $F$. Considering all the terms from the RHS of (101), there will be many terms with the same $h - g$, and we will collect all of them in the end of this subsection. Note that the resulting coefficients only make sense when they are computed for finite operators, that is when the resulting operator has $|m| \leq \ell$. For now, let us proceed with computing the constants $\alpha_{h,g}^{\ell_k,\ell_{k+1}}$.

In the case of $g = 0$, the contour manipulation (103) yields the coefficient

$$\alpha_{h,0}^{\ell_k,\ell_{k+1}} = \prod_{j=0}^{h-1} \frac{q^{-2\ell_{k+1}} - q^{2(\ell_{k+1}-j)}}{q^{2(1+\ell_k+\ell_{k+1}-h)} - q^{-2(\ell_k+\ell_{k+1}-j)}}. \tag{106}$$

Expressions for the subleading terms, $\alpha_{h,g}^{\ell_k,\ell_{k+1}}$, can be computed using the following recursive procedure.

---

[12] Here we use the fact that $\alpha_{h,g}^{\ell_k,\ell_{k+1}} = 0$ for $g < h - 2\ell_{k+1}$. This is because when $g < h - 2\ell_{k+1}$, there must be one recursion step with $j = 2\ell_{k+1}$ in eq. (103). If we want to further increase $j$, the prefactor of the second term in the RHS of eq. (103) vanishes: $q^{-2\ell_{k+1}} - q^{2(\ell_{k+1}-j)} = 0$.

Let us define $G_{h,j}^{(g)}$ through the recursion relation[13]

$$G_{h,j}^{(g)} = \alpha_1(h,j)\, G_{h-1,j}^{(g-1)} + \alpha_2(h,j)\, G_{h-1,j+1}^{(g)}, \tag{107}$$

together with the initial conditions

$$G_{0,j}^{(0)} = 1, \qquad \text{and} \quad G_{h,j}^{(g)} = 0, \quad \forall g > h. \tag{108}$$

Here, we have defined,

$$
\begin{aligned}
\alpha_1(h,j) &\equiv \left[ q^{-2(h+j-1-\ell_k-\ell_{k+1})} - q^{2(j-\ell_k-\ell_{k+1})} \right]^{-1}, \\
\alpha_2(h,j) &\equiv \frac{q^{-2\ell_{k+1}} - q^{2(\ell_{k+1}-j)}}{q^{2(1+\ell_k+\ell_{k+1}-h-j)} - q^{-2(\ell_k+\ell_{k+1}-j)}},
\end{aligned}
\tag{109}
$$

where we omitted the dependence on $\ell_k$ and $\ell_{k+1}$ to simplify notation. With this definition it follows from (106) that,

$$\alpha_{h,g}^{\ell_k,\ell_{k+1}} = G_{h,0}^{(g)}. \tag{110}$$

**Step 4**

We reduced the contour integrals to the lines between $x_k$ and $x_{k+1}$ and the lines going around both points, see the RHS of (104). Now, let us work with a single term in the RHS of (104). Due to conformal symmetry, we only need to compute the OPE coefficient of primary operators. As a consequence of anomalous charge conservation in the chiral free boson theory (33), the configuration under consideration will only contribute to one primary operator and its Virasoro descendants. In other words, the OPE limit corresponds to taking $\mathcal{V}_{-\ell_k\alpha_-}(x_k)$, $\mathcal{V}_{-\ell_{k+1}\alpha_-}(x_{k+1})$ and $(h-g)$ integrated operators $\mathcal{V}_{\alpha_-}$ very close to each other. Therefore, only the representation with spin

$$\ell = \ell_k + \ell_{k+1} - (h-g), \tag{111}$$

can appear in the contribution of a specific value of $(h-g)$. That means that we only need to compute the most singular term in the OPE. A simple consequence is that for any interaction with any of the $F$ lines wrapping both $x_k$ and $x_{k+1}$ or with any of the rest of the operators in the correlator we can simply take the leading term in the Taylor expansion of $x_{k+1}$ around $x_k$. Now we are left with the task of computing the OPE of the two operators connected by $(h-g)$ lines, see the LHS of figure 16. Using the OPE of vertex operators given by (26), we obtain the formula which corresponds to the middle of figure 16

$$
\left( \int_L dz_1 \dots dz_{h-g} \prod_{i=1}^{h-g}(z_i - x_k)^{-2\ell_k\alpha_-^2} \prod_{j=1}^{h-g}(x_{k+1} - z_j)^{-2\ell_{k+1}\alpha_-^2} \prod_{i<j}(z_i - z_j)^{2\alpha_-^2} \right)
$$
$$
\times (x_{k+1} - x_k)^{2\ell_k\ell_{k+1}\alpha_-^2}\, \mathcal{V}_{-(\ell_k+\ell_{k+1}-(h-g))\alpha_-}(x_k). \tag{112}
$$

As mentioned above, different values of $(h-g)$ contribute to different primaries appearing in the OPE. Starting from (112), we perform a change of variables $z_i = (1-t_i)x_k + t_i x_{k+1}$, so that $t_i \in [0,1]$:

$$
\underbrace{\left( \int_L dt_1 \dots dt_{h-g} \prod_{i=1}^{h-g}(t_i)^{-2\ell_k\alpha_-^2}(1-t_i)^{-2\ell_{k+1}\alpha_-^2} \prod_{i<j}(t_i - t_j)^{2\alpha_-^2} \right)}_{const}
$$
$$
\times (x_{k+1} - x_k)^{(h-g)+[2\ell_k\ell_{k+1}-2\ell_k(h-g)-2\ell_{k+1}(h-g)+(h-g)^2-(h-g)]\alpha_-^2}\, \mathcal{V}_{-\ell\alpha_-}(x_k). \tag{113}
$$

---

[13]Note that here we should treat the $G$'s as finite quantities. In other words, we do not recursively evaluate $G$ when it multiplies a vanishing quantity. The graphical meaning of $G_{h,j}^{(g)}$ is the coefficient when we expand the contour integral with $h$ $(w-x_k)$-lines, $j$ $(x_k-x_{k+1})$-lines and 0 screening charges (the picture in the LHS of (103)) into the contour integrals with 0 $(w-x_k)$-lines, $h+j-g$ $(x_k-x_{k+1})$-lines and $g$ screening charges.

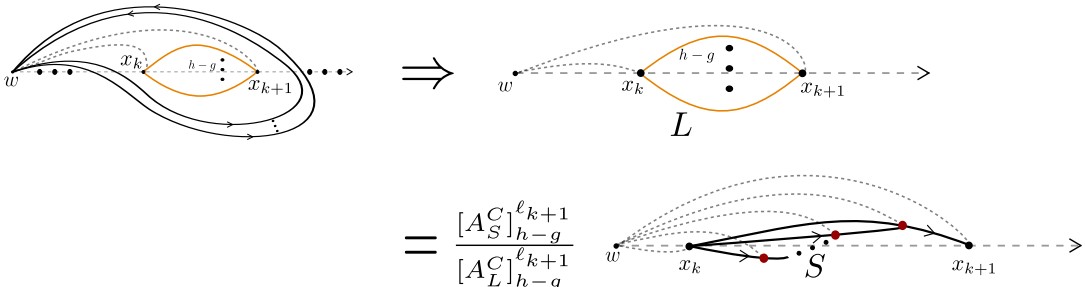

Figure 16: We start from the configuration on the left. The $F$ lines acting on the whole configuration together and other operators in the correlator do not have any effect on this step. We trade the straight contours from $x_k$ to $x_{k+1}$ by ordered line contours. This brings about an overall multiplicative factor that is computed in appendix B.

One can verify that under the identification (111), the power of $(x_{k+1} - x_k)$ is exactly equal to

$$-(h_{1,2\ell_k+1} + h_{1,2\ell_{k+1}+1} - h_{1,2\ell+1}),$$

where the conformal dimensions in the Coulomb gas notations are given in (97). This is exactly the power we expect to appear in the OPE (99). Including the subleading terms in the Taylor expansion for the interactions with all the other operators that are not present in the figure (including all $F$ lines acting on the configurations, the operator at $w$, and other operators in the correlator) gives subleading terms that are fixed by conformal symmetry.

We are left with the task of computing the constant that multiplies the vertex operator and the coordinate dependence in (113). The first step in evaluating this expression is to represent the same quantity as an ordered integration, see the RHS of figure 16. Here the integration domain is $\mathcal{D} = \{t_1, \ldots, t_{(h-g)} : 0 < t_{(h-g)} < \ldots < t_1 < 1\}$. The relevant computation is carried out in the appendix B. It gives us a factor of (B.7)

$$\frac{[A_S^C]_{h-g}^{\ell_{k+1}}}{[A_L^C]_{h-g}^{\ell_{k+1}}} \equiv \Sigma_{h-g} = \prod_{k=1}^{h-g} \frac{1-q^{2k}}{1-q^2}. \tag{114}$$

Here the dependence on $\ell_{k+1}$ disappears from the RHS. The ordered integral over the contour $S$ given by the expression above is precisely the well known Selberg integral (A.3) up to a factor of $(h-g)!$. The parameters of the Selberg integral are given by $\alpha = 1 - 2\ell_k \alpha_-^2$, $\beta = 1 - 2\ell_{k+1}\alpha_-^2$, $\gamma = \alpha_-^2$. So, the constant in (113) is given by,

$$\frac{\Sigma_{h-g}}{(h-g)!} S_{h-g}(1 - 2\ell_k \alpha_-^2, 1 - 2\ell_{k+1}\alpha_-^2, \alpha_-^2). \tag{115}$$

Now, we recall that there are $(g + \ell_{k+1} - m_{k+1} - n)$ lowering operators $F$ that surround the vertex operator $\mathcal{V}_{-\ell\alpha_-}(x_k)$. The number of lines determine the quantum number $m$ of the operator $V_{\ell,m}$ appearing in the OPE, i.e. the number of lines is equal to $(\ell - m)$. Taking into account the value of $\ell$, (111), and the expression for $h$, (105), one can verify that,

$$m = m_k + m_{k+1}, \tag{116}$$

as one would expect in the OPE. If one substitutes the expression for $h$ into the formula for $\ell$, one gets $\ell = \ell_{k+1} + m_k - n + g$, that is, the spin of the operator depends on the numbers $n, g$ over which we have a summation in the steps 1 and 3 respectively. Then $\ell$ takes values in between

$$|m_k + m_{k+1}| \leq \ell \leq \ell_k + \ell_{k+1}, \tag{117}$$

which, again, matches with our expectations (99).

Finally, putting together all of the ingredients from all of the steps, one gets a final expression for the OPE,

$$
\begin{aligned}
\widetilde{C}^{(\ell,m)}_{(\ell_k,m_k),(\ell_{k+1},m_{k+1})} = \sum_{n=0}^{\min(\ell_{k+1}-m_{k+1},m_k+\ell_k)} \beta^n_{\ell_{k+1},m_{k+1}}[A^C_L]^{\ell_k}_{\ell_k-m_k+n} \\
\times \sum_{g=g_0}^{\ell_k-m_k+n} \frac{\alpha^{\ell_k,\ell_{k+1}}_{t+g,g}}{t!}\Sigma_t S_t(1-2\ell_k\alpha^2_-,1-2\ell_{k+1}\alpha^2_-,\alpha^2_-)\delta_{\ell_k-m_k+n-g,t},
\end{aligned}
\tag{118}
$$

where we used the notation $t \equiv (h-g) = \ell_k+\ell_{k+1}-\ell$ and $g_0 = \max(0,\ell_k-m_k-2\ell_{k+1}+n)$. Let us provide the references to all the notations used in the formula: the coefficients $\beta^n_{\ell_{k+1},m_{k+1}}$ are defined in (102), $A^C_L$ are defined in (35), $\alpha^{\ell_k,\ell_{k+1}}_{h,g}$ are defined via a generating function in (110), the formula for $\Sigma_t$ is given in (114), finally the expression for the Selberg integral is given in (A.3). This formula represents the generic OPE coefficient, and we have used it to perform various consistency checks. For example, we checked that such OPE coefficients are consistent with the Ward identities (61). However, it is quite cumbersome and the dynamical information can be found by considering one representative for each choice of three multiplets (since the Ward identities (61) relate all such representative OPE coefficients). In the next subsection we present a choice of representative that simplifies the expression significantly.

### 3.5.2 Computing the simplest OPE coefficients

For the purpose of recovering all of the dynamical information, it is sufficient to compute the OPE coefficient for a special case where one of the initial operators and also the final operator are highest weight. Equivalently, we are after the 3-point function in which 2 out of 3 operators are highest weight. Specifically, we will take the first operator (inserted at $x_k$) to be arbitrary and the second operator (inserted at $x_{k+1}$) to be highest weight, i.e. initially it is not surrounded by any lines. In that case, step 1 trivializes, $n_{k+1} = \ell_{k+1} - m_{k+1} = 0$. The coefficient appearing in the step 2 is equal to $[A^C_L]^{\ell_k}_{\ell_k-m_k}$. Finally, since we are interested in the case when the resulting operator is highest weight (and hence has no $F$ lines acting on it), we pick out the $g = 0$ term from step 3. The final result simplifies to,

$$
\widetilde{C}^{(m+\ell_{k+1},m+\ell_{k+1})}_{(\ell_k,m),(\ell_{k+1},\ell_{k+1})} = [A^C_S]^{\ell_k}_{\ell_k-m}\frac{\alpha^{\ell_k,\ell_{k+1}}_{\ell_k-m,0}}{(\ell_k-m)!}S_{\ell_k-m}(1-2\ell_k\alpha^2_-,1-2\ell_{k+1}\alpha^2_-,\alpha^2_-).
\tag{119}
$$

Note that here we used that $[A^C_L]^{\ell_k}_{\ell_k-m_k}\Sigma_{\ell_k-m} = [A^C_S]^{\ell_k}_{\ell_k-m_k}$; the expression for $A^C_S$ is obtained by combining (B.3) and (B.7):

$$
[A^C_S]^\ell_n = \prod_{k=1}^n \frac{(1-q^{2k})\left(q^{2(\ell-k+1)}-q^{-2\ell}\right)}{1-q^2},
\tag{120}
$$

the coefficient $\alpha^{\ell_k,\ell_{k+1}}_{\ell_k-m,0}$ is given in (106) and the expression for the Selberg integral is given in (A.3).

Let us simplify the notation: $\ell_1 \equiv \ell_k$, $\ell_2 \equiv \ell_{k+1}$, and $\ell_3 \equiv \ell_{k+1}+m$. Then, (119) can be rewritten as

$$
\widetilde{C}^{(\ell_3,\ell_3)}_{(\ell_1,\ell_3-\ell_2),(\ell_2,\ell_2)} = [A^C_S]^{\ell_1}_{\ell_1+\ell_2-\ell_3}\frac{\alpha^{\ell_1,\ell_2}_{\ell_1+\ell_2-\ell_3,0}}{(\ell_1+\ell_2-\ell_3)!}S_{\ell_1+\ell_2-\ell_3}(1-2\ell_1\alpha^2_-,1-2\ell_2\alpha^2_-,\alpha^2_-).
\tag{121}
$$

Combining all the factors and using the $\Gamma$-function identities (A.1), the explicit expression for this OPE coefficient becomes

$$\widetilde{C}^{(\ell_3,\ell_3)}_{(\ell_1,\ell_3-\ell_2),(\ell_2,\ell_2)} = \frac{(2\pi i)^{\ell_1+\ell_2-\ell_3}}{(\ell_1+\ell_2-\ell_3)!} \prod_{k=1}^{\ell_1+\ell_2-\ell_3} \frac{\Gamma\big((\ell_1+\ell_2+\ell_3-k+2)\alpha_-^2 - 1\big)\Gamma(-\alpha_-^2)}{\Gamma\big((2\ell_1-k+1)\alpha_-^2\big)\Gamma\big((2\ell_2-k+1)\alpha_-^2\big)\Gamma(-k\alpha_-^2)}.$$
(122)

This expression is consistent with the fusion rules of the $U_q(sl_2)$ representations. For a non-vanishing $\widetilde{C}^{(\ell_3,\ell_3)}_{(\ell_1,\ell_3-\ell_2),(\ell_2,\ell_2)}$, the condition $|\ell_1-\ell_2| \leqslant \ell_3 \leqslant \ell_1+\ell_2$ must hold. The result in (122) also generalizes the computations performed for two-point functions in section 3.4. Specifically, the OPE coefficient between the operators $V_{\ell,-\ell}$, $V_{\ell,\ell}$, and the identity reduces to the two-point function calculation in (96). For such a choice of operators, we have

$$\widetilde{C}^{(0,0)}_{(\ell,-\ell),(\ell,\ell)} = \frac{(2\pi i)^{2\ell}}{(2\ell)!} \prod_{k=1}^{2\ell} \frac{\Gamma\big((k+1)\alpha_-^2 - 1\big)\Gamma(-\alpha_-^2)}{\Gamma\big(k\alpha_-^2\big)^2 \Gamma(-k\alpha_-^2)},$$
(123)

which matches precisely with the prefactor in the final result for the two-point function (96).

## 3.6 Canonical normalization and comparison of OPE coefficients

As explained in section 2, representation theory provides us with a canonical normalization for operators in a theory with quantum group symmetry. For CFTs, the correlation functions of such operators are determined by the rules in eqs. (6) and (8). This is the normalization in which the OPE coefficients were found using crossing symmetry, see (17). However, this is not the normalization that naturally comes out from the Coulomb gas formalism. This can be seen, for example, from the two-point functions of one highest-weight and one lowest-weight operators computed in section 3.4. Another difference is that in the Coulomb gas formalism the action of the lowering operator $F$ was defined as $V_{\ell,m-1} = FV_{\ell,m}$. To match with the canonically normalized operators, the action of $F$ should yield a coefficient $f^\ell_m$ given by (4). Since we have explicitly computed all two-point functions (94), we can canonically normalize the operators. This will allow us to compare the resulting Coulomb gas OPE coefficients (118) with the ones found by crossing symmetry (17). The argument below works for both integer and half-integer spin $\ell$.

To convert the results of this section to canonical normalization, we start by relating the highest-weight operators defined in the two normalizations with an unknown coefficient $\mathcal{N}_\ell$,

$$V_{\ell,\ell} = \mathcal{N}_\ell W^\ell_{1,2\ell+1}.$$
(124)

Then the repeated action of $F$ on both sides gives

$$V_{\ell,m} = \mathcal{N}_\ell W^m_{1,2\ell+1} \prod_{m'=m+1}^{\ell} f^\ell_{m'}.$$
(125)

The explicit form of the $f^\ell_m$'s is given in (4). Now, the unknown factors $\mathcal{N}_\ell$ can be found straightforwardly by comparing 2-point functions computed in the two normalizations. Namely, in (94) an explicit expression was given for all the two-point functions of one lowest-weight and one highest-weight operators in the Coulomb gas normalization. On the other hand, the two-point functions in canonical normalization are given in (6). Translating the words into formulas, we have

$$\langle V_{\ell,-\ell}(0)V_{\ell,\ell}(1)\rangle = \mathcal{N}_\ell^2 \left(\prod_{m=-\ell+1}^{\ell} f^\ell_m\right)\langle W^{-\ell}_{1,2\ell+1}(0)W^\ell_{1,2\ell+1}(1)\rangle.$$
(126)

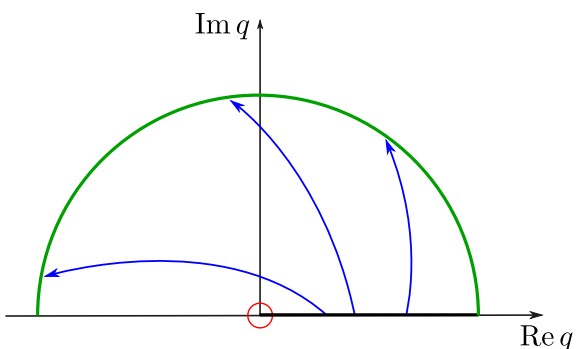

Figure 17: The half-disc $|q| < 1$ with $\text{Im}(q) > 0$, excluding $q = 0$. $\left([n]_q\right)^\gamma$ is positive for $q \in (0, +\infty)$.

Using equation (6) for the RHS and the fact that $\left\langle V_{\ell,-\ell}(0) V_{\ell,\ell}(1) \right\rangle = \widetilde{C}^{(0,0)}_{(\ell,-\ell),(\ell,\ell)}$, the above equation determines $\mathcal{N}_\ell^2$ to be

$$\mathcal{N}_\ell^2 = \frac{\widetilde{C}^{(0,0)}_{(\ell,-\ell),(\ell,\ell)}}{\left(\displaystyle\prod_{m=-\ell+1}^{\ell} f_m^\ell\right)\begin{bmatrix} \ell & \ell & 0 \\ -\ell & \ell & 0 \end{bmatrix}_q} \,. \tag{127}$$

Here, the factor $\widetilde{C}^{(0,0)}_{(\ell,-\ell),(\ell,\ell)}$ is given in eq. (123), $f_m^\ell$ is defined in eq. (4), and $[\ldots]_q$ represents the quantum Clebsch-Gordan coefficient (see [1, eq. (2.24)]). By substituting the explicit values of these quantities into the above equation, we obtain:

$$\mathcal{N}_\ell^2 = \frac{(-2\pi i q)^{2\ell} \sqrt{[2\ell+1]_q}}{(2\ell)! \, [2\ell]!} \prod_{k=1}^{2\ell} \frac{\Gamma\left((k+1)\alpha_-^2 - 1\right)\Gamma(-\alpha_-^2)}{\Gamma\left(k\alpha_-^2\right)^2 \Gamma(-k\alpha_-^2)} \,. \tag{128}$$

Here, $(n)!$ denotes the standard factorial of $n$, and $[n]!$ denotes the $q$-deformed factorial.

The expression (128) of $\mathcal{N}_\ell$ has a multi-valued factor $\sqrt{[2\ell+1]_q}$ factor, which comes from the Clebsch-Gordan coefficient in (127). In this paper, we always choose the branch as follows. When $q$ is real and positive, the $q$-numbers are always positive. There, we choose the principal branch for the real power of the $q$-number:

$$\left([n]_q\right)^\gamma > 0, \qquad \forall q > 0, \quad n \in \mathbb{Z}_+, \quad \gamma \in \mathbb{R}. \tag{129}$$

Then we analytically continue $\left([n]_q\right)^\gamma$ to the upper half disc (with $q = 0$ excluded):

$$\text{UHD} := \{q \in \mathbb{C} : |q| < 1, \ \text{Im}(q) \geq 0, \ q \neq 0\} \,. \tag{130}$$

See figure 17. $\left([n]_q\right)^\gamma$ for $q$ on the upper unit circle, i.e. $q = e^{i\theta}$ with $0 \leq \theta \leq \pi$, is obtained by taking the limit from the upper half disc. Under this convention, we have

$$\log([n]_q) = \log(n) - (n-1)\pi i, \quad \text{when } q = -1, \tag{131}$$

for the $q$-number-related quantities. Then the value of $\mathcal{N}_\ell^2$ at $q = -1$ ($c = 1$) is given by

$$\mathcal{N}_\ell^2\big|_{q=-1} = (2\pi)^{2\ell} \sqrt{2\ell+1}, \tag{132}$$

which is strictly positive for all $\ell = 0, 1/2, 1, 3/2, \ldots$ There is still a "$\pm$" ambiguity for $\mathcal{N}_\ell$, which corresponds to the field redefinition $V_{\ell,m} \to -V_{\ell,m}$. We fix it to be positive at $q = -1$:

$$\mathcal{N}_\ell\big|_{q=-1} = (2\pi)^\ell (2\ell+1)^{1/4}. \tag{133}$$

This fixes the normalizaton factor $\mathcal{N}_\ell$ in the whole upper half disc.

Now, let us derive the general OPE coefficients in the chiral subsector of $\text{XXZ}_q$ under the canonical normalization, namely $C^{\text{XXZ}}_{(1,2\ell_1+1)(1,2\ell_2+1)(1,2\ell_3+1)}$, which are defined in (16). In this case, (16) simplifies to a single sum:

$$
W^{m_1}_{1,2\ell_1+1} W^{m_2}_{1,2\ell_2+1} = \sum_{\ell_3=|m_1+m_2|}^{\ell_1+\ell_2} C^{\text{XXZ}}_{(1,2\ell_1+1),(1,2\ell_2+1),(1,2\ell_3+1)} \begin{bmatrix} \ell_1 & \ell_2 & \ell_3 \\ m_1 & m_2 & m_1+m_2 \end{bmatrix}_q (W^{m_1+m_2}_{1,2\ell_3+1} + \dots).
$$
(134)

Here, the dots represent the terms corresponding to the Virasoro descendants, and the dependence on coordinates is omitted for simplicity.

On the other hand, the OPE of the Coulomb-gas operators is given in (99). To align it with (134), we rewrite it as

$$
V_{\ell_1,m_1} V_{\ell_2,m_2} = \sum_{\ell_3=|\ell_1-\ell_2|}^{\ell_1+\ell_2} \widetilde{C}^{(\ell_3,m_1+m_2)}_{(\ell_1,m_1),(\ell_2,m_2)} V_{\ell_3,m_1+m_2}.
$$
(135)

By matching (134) with (135) using the relation (125), we obtain

$$
C_{(1,2\ell_1+1),(1,2\ell_2+1),(1,2\ell_3+1)} = \frac{\mathcal{N}_{\ell_3} \left( \prod_{m=m_1+m_2+1}^{\ell_3} f^{\ell_3}_m \right) \widetilde{C}^{(\ell_3,m_1+m_2)}_{(\ell_1,m_1),(\ell_2,m_2)}}{\mathcal{N}_{\ell_1} \mathcal{N}_{\ell_2} \left( \prod_{m=m_1+1}^{\ell_1} f^{\ell_1}_m \right) \left( \prod_{m=m_2+1}^{\ell_2} f^{\ell_2}_m \right) \begin{bmatrix} \ell_1 & \ell_2 & \ell_3 \\ m_1 & m_2 & m_1+m_2 \end{bmatrix}_q},
$$
(136)

where $|m_1+m_2| \leqslant \ell_3$. Note that $C^{\text{XXZ}}_{(1,2\ell_1+1),(1,2\ell_2+1),(1,2\ell_3+1)} = C_{(1,2\ell_1+1),(1,2\ell_2+1),(1,2\ell_3+1)}$ when all $\ell_i$ are integers. The RHS of (136) does not depend on the quantum numbers $m_i$. Therefore, to compute $C_{(1,2\ell_1+1)(1,2\ell_2+1)(1,2\ell_3+1)}$, we can freely choose $m_1$ and $m_2$, provided that $|m_1+m_2| \leqslant \ell_3$. As we have seen before, selecting $m_1 = \ell_3 - \ell_2$ and $m_2 = \ell_2$, i.e. taking two out of three operators to be highest weight, significantly simplifies the computation:

$$
C_{(1,2\ell_1+1),(1,2\ell_2+1),(1,2\ell_3+1)} = \frac{\mathcal{N}_{\ell_3} \widetilde{C}^{(\ell_3,\ell_3)}_{(\ell_1,\ell_3-\ell_2),(\ell_2,\ell_2)}}{\mathcal{N}_{\ell_1} \mathcal{N}_{\ell_2} \left( \prod_{m=\ell_3-\ell_2+1}^{\ell_1} f^{\ell_1}_m \right) \begin{bmatrix} \ell_1 & \ell_2 & \ell_3 \\ \ell_3-\ell_2 & \ell_2 & \ell_3 \end{bmatrix}_q}.
$$
(137)

The main simplification arises because all the ingredients in this expression are products, making the final result a product formula as well. We leave the detailed derivation to appendix E and present only the result here:

$$
\begin{aligned}
C_{(1,2\ell_1+1),(1,2\ell_2+1),(1,2\ell_3+1)} = &\frac{1}{[2\ell_3]!} \sqrt{\frac{[\ell_1-\ell_2+\ell_3]! \, [\ell_2-\ell_1+\ell_3]! \, [\ell_1+\ell_2+\ell_3+1]!}{\left([2\ell_1+1]_q [2\ell_2+1]_q [2\ell_3+1]_q\right)^{1/2} [\ell_1+\ell_2-\ell_3]!}} \\
&\times \prod_{k=1}^{\ell_1+\ell_2-\ell_3} \frac{\Gamma\left((\ell_1+\ell_2+\ell_3-k+2)\alpha_-^2-1\right)}{\Gamma\left((2\ell_1-k+1)\alpha_-^2\right) \Gamma\left((2\ell_2-k+1)\alpha_-^2\right) \Gamma(1-k\alpha_-^2)} \\
&\times \sqrt{\left(\prod_{k=1}^{2\ell_1} \frac{\Gamma\left(k\alpha_-^2\right)^2 \Gamma(1-k\alpha_-^2)}{\Gamma\left((k+1)\alpha_-^2-1\right)}\right) \left(\prod_{k=1}^{2\ell_2} \frac{\Gamma\left(k\alpha_-^2\right)^2 \Gamma(1-k\alpha_-^2)}{\Gamma\left((k+1)\alpha_-^2-1\right)}\right)} \\
&\times \sqrt{\prod_{k=1}^{2\ell_3} \frac{\Gamma\left((k+1)\alpha_-^2-1\right)}{\Gamma\left(k\alpha_-^2\right)^2 \Gamma(1-k\alpha_-^2)}}.
\end{aligned}
$$
(138)

In this expression, all the phase ambiguities have been fixed above. In particular, at $q = -1$ we have

$$C_{(1,2\ell_1+1),(1,2\ell_2+1),(1,2\ell_3+1)}\Big|_{q=-1} = \sqrt{\frac{(\ell_1+\ell_2-\ell_3)!(\ell_1+\ell_3-\ell_2)!(\ell_2+\ell_3-\ell_1)!(\ell_1+\ell_2+\ell_3+1)!}{(2\ell_1)!(2\ell_2)!(2\ell_3)!\sqrt{(2\ell_1+1)(2\ell_2+1)(2\ell_3+1)}}}$$

$$\times \prod_{k=1}^{\ell_1+\ell_2-\ell_3} \frac{\Gamma(k)\Gamma(\ell_1+\ell_2+\ell_3-k+1)}{\Gamma(2\ell_1-k+1)\Gamma(2\ell_2-k+1)}. \tag{139}$$

It is positive and symmetric under permutations of $\ell_1$, $\ell_2$ and $\ell_3$.

Now, let us verify and improve the relation (17) for the OPE coefficients found by crossing symmetry. The expression for the minimal-model OPE coefficients $C^{MM}_{(1,s_1),(1,s_2),(1,s_3)}$ is given in [5]:[14]

$$C^{MM}_{(1,s_1),(1,s_2),(1,s_3)} = \alpha_-^{2(s_1+s_2-s_3-2)} \sqrt{\frac{\gamma(\alpha_-^2-1)\gamma(s_1-\alpha_-^{-2})\gamma(s_2-\alpha_-^{-2})\gamma(-s_3+\alpha_-^{-2})}{\gamma(1-\alpha_-^{-2})\gamma(-1+s_1\alpha_-^2)\gamma(-1+s_2\alpha_-^2)\gamma(1-s_3\alpha_-^2)}} \tag{140}$$

$$\times \prod_{k=1}^{\frac{s_1+s_2-s_3-1}{2}} \left[ \gamma(k\alpha_-^2)\gamma\left(-1+(k+s_3)\alpha_-^2\right)\gamma\left(1+(k-s_1)\alpha_-^2\right)\gamma\left(1+(k-s_2)\alpha_-^2\right) \right],$$

where

$$\gamma(x) \equiv \frac{\Gamma(x)}{\Gamma(1-x)}. \tag{141}$$

(140) involves square root functions, so it is also multi-valued. Here, we fix the ambiguity in the upper half disc (130) by choosing the principal branch of the full square root around $q = -1$ ($\alpha_-^2 = 1$). Under this convention, the minimal-model OPE coefficients are positive at $q = -1$.

Since we have derived the explicit expression, the "±" ambiguity found in (17) can now be resolved as follows:

$$\left(C_{(1,2\ell_1+1),(1,2\ell_2+1),(1,2\ell_3+1)}\right)^2 = C^{MM}_{(1,2\ell_1+1),(1,2\ell_2+1),(1,2\ell_3+1)}, \tag{142}$$

where $\ell_i$'s are integers or half integers. When all $\ell_i$ are integers we get exactly the expression for chiral OPE coefficients of XXZ$_q$ CFT. We leave the technical details to appendix E.

A byproduct of (142) is that the chiral OPE coefficient $C_{(1,2\ell_1+1),(1,2\ell_2+1),(1,2\ell_3+1)}$ is symmetric under permutations of $\ell_i$'s. This is because (a) $C_{(1,2\ell_1+1),(1,2\ell_2+1),(1,2\ell_3+1)}$ is symmetric at $q = -1$ as demonstrated above, and (b) $C^{MM}_{(1,2\ell_1+1),(1,2\ell_2+1),(1,2\ell_3+1)}$ is symmetric for any $q$ on the upper unit circle. This property is consistent with the prediction from the crossing symmetry and the braid locality condition in [1].

After canonically normalizing the operators, we can verify that they satisfy the $R$-matrix identity (9) with plus sign in the RHS (For explicit formulas see appendix A of [1]). For this to work it was important that their spins are consistent with the selection rule (10).

## 4 More general quantum groups and construction of non-chiral operators in XXZ$_q$ CFT

The XXZ$_q$ partition function exhibits non-chiral operators of conformal dimensions $(h_{r,s}, h_{r,1})$, which transform in spin-$\ell$ representation of the quantum group (with $\ell = \frac{s-1}{2}$), see (15). In

---

[14]The final result of the minimal model OPE coefficients is not explicit in [5]. Here we use a slightly simplified version from [25].

order to compute correlators including these operators we have to extend the Coulomb gas approach of section 3 in the following way. First, we consider chiral operators of dimension $(h_{r,s}, 0)$, which transform under two quantum groups, one is associated with the index $r$ and the other one with the index $s$. In subsection 4.1 we show that this chiral theory with two quantum group symmetries is consistent on its own, and explain how to define its correlators. Second, in subsection 4.2 we combine it with anti-chiral operators of dimension $(0, h_{r,1})$ to produce correlators of generic operators in the $XXZ_q$ CFT. To do this we need to pick a particular combination of operators, or a "projection" which eliminates the quantum groups acting on the index $r$ in both chiral and anti-chiral components.

It turns out that if we combine the $(h_{r,s}, 0)$ chiral theory with a more general, $(0, h_{r,s})$, anti-chiral theory we can construct projections that realize more theories. In particular, in subsection 4.3, we show how to construct generalized diagonal minimal models with the help of one such projection, by using an approach similar to [26]. The appearance of two different $6j$-symbols in the crossing kernel of operators, sometimes referred to in the literature as a "hidden quantum group symmetry", is a direct consequence of this construction.

## 4.1 Chiral theory with two quantum groups

As emphasized above, the first step of the construction of the non-chiral operators of the $XXZ_q$ theory is to define a chiral theory with two quantum group symmetries. Recall that in section 3.2 we identified the screening charge $S_-$ with the quantum group generator $F$, see (36). Then, by studying the action of this generator on two operators we identified that $q = e^{i\pi\alpha_-^2}$, see figure 8. In this section we relabel these, $F_-$ and $q_-$. An identical procedure can be repeated for the generator $S_+$. This will give rise to a second quantum group with the identification $F_+ = S_+$ and $q_+ = e^{i\pi\alpha_+^2}$.

**Construction of operators.** We extend the chiral theory defined in section 3.2 by including vertex operators $\mathcal{V}_{\alpha_{r,s}}$ from the Kac table (31) with any positive integers $r$ and $s$ (in contrast to the restriction $r = 1$ of section 3.2). The spins $\ell^+$ and $\ell^-$ are related to the Kac indices through the relations $r = 2\ell^+ + 1$ and $s = 2\ell^- + 1$. We then act on the vertex operators with any number of screening charges $S_-$ or $S_+$,

$$\mathcal{V}_{\alpha_{2\ell^++1, 2\ell^-+1}} \longrightarrow S_+^m S_-^n \mathcal{V}_{\alpha_{2\ell^++1, 2\ell^-+1}}.$$

As before, the action of the screening charges is defined by the choice of contours, see figure 18. Note that the contours are chosen exactly as they were in the single quantum group case (see LHS of figure 4). The screening charges commute for $\alpha_0 > 0$, thus we can treat the contours

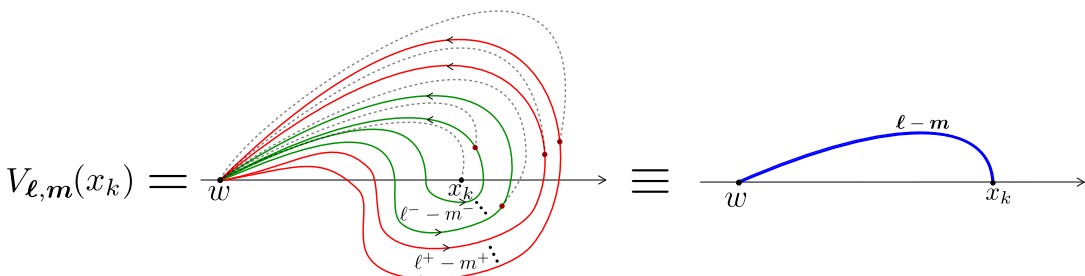

Figure 18: This figure defines the contours for $S_+$ (red lines) and $S_-$ (green lines) appearing in the definition of the operator (143). The order in which they are drawn does not matter since $[S^+, S^-] = 0$. On the RHS, we have introduced a shorthand graphical notation for the operator at $V_{\ell, m}$.

corresponding to $S_-^n$ and the contours corresponding to $S_+^m$ separately. For both types of the contours the same analysis as in figure 4 can be done. It yields a coefficient $[A_L^C]_m^{\ell^+}[A_L^C]_n^{\ell^-}$, which vanishes for either $m \geqslant 2\ell^+ + 1$ or $n \geqslant 2\ell^- + 1$. So, it is natural to identify the operators that transform under the action of two screening charges as

$$V_{\boldsymbol{\ell},\boldsymbol{m}} \equiv S_+^{\ell^+ - m^+} S_-^{\ell^- - m^-} \mathcal{V}_{\alpha_{2\ell^+ + 1, 2\ell^- + 1}}, \qquad F_\pm = S_\pm, \tag{143}$$

where $\boldsymbol{\ell} = (\ell^+, \ell^-)$ and $\boldsymbol{m} = (m^+, m^-)$. As before, the quantum numbers are in the range $-\ell^\pm \leqslant m^\pm \leqslant \ell^\pm$, and the operator $V_{\boldsymbol{\ell},\boldsymbol{m}}$ transforms in the $(2\ell^+ + 1)(2\ell^- + 1)$-dimensional representation.

As mentioned around eq. (26), since the chiral vertex operators in the free boson theory are not local, it allows us to assign values to their OPE at each ordering separately (nontrivial choices are commonly referred to in the literature as cocycles). These choices are encoded in (26) in the form of the phases $\varphi(\alpha, \beta)$. The choice of cocycles which gives rise to a theory with two independent quantum group symmetries (as we will see shortly after the derivation of the coproduct) is,[15]

$$\varphi(k_1^- \alpha_- + k_1^+ \alpha_+, k_2^- \alpha_- + k_2^+ \alpha_+) = i^{4k_1^+ k_2^-}. \tag{144}$$

Importantly, this expression ensures the associativity of the OPE. Now, we can use the OPE (26) to derive the $n$-point correlation function of vertex operators with general integer or half-integer $k^\pm$,

$$\left\langle \mathcal{V}_{\alpha_1}(z_1) \ldots \mathcal{V}_{\alpha_n}(z_n) \right\rangle_{\text{CG}} = \prod_{i<j} (z_i - z_j)^{2\alpha_i \alpha_j} \varphi(\alpha_i, \alpha_j). \tag{145}$$

**Construction of correlators.** Next, we define the correlation functions of operators $V_{\boldsymbol{\ell},\boldsymbol{m}}$. We generalize the definition (38) to include 2 quantum group labels,

$$\left\langle V_{\boldsymbol{\ell}_1,\boldsymbol{m}_1}(x_1) \ldots V_{\boldsymbol{\ell}_n,\boldsymbol{m}_n}(x_n) \right\rangle \equiv \int_\Gamma \prod_{j=1}^n dz^{(j)} du^{(j)} \Big\langle \mathcal{V}_{2\alpha_0}(w) \cdot \mathcal{V}_{\alpha_1}(x_1) \mathcal{V}_{\alpha_-}\big(z^{(1)}\big) \mathcal{V}_{\alpha_+}\big(u^{(1)}\big) \ldots$$

$$\times \mathcal{V}_{\alpha_n}(x_n) \mathcal{V}_{\alpha_-}\big(z^{(n)}\big) \mathcal{V}_{\alpha_+}\big(u^{(n)}\big) \Big\rangle_{\text{CG}}. \tag{146}$$

Here, $\alpha_j = -\ell_j^- \alpha_- - \ell_j^+ \alpha_+$; recall that $\mathcal{V}_{\alpha_-}\big(z^{(i)}\big)$ and $\mathcal{V}_{\alpha_+}\big(u^{(i)}\big)$ stand for products of screening charges, see (39). The integral contains two types of the contours corresponding to $F_\pm$, for each type the contour is exactly the same as in figure 5.

These correlation functions satisfy the same properties as listed in section (3.2): do not depend on the position of $w$; satisfy Ward identities with respect to the group $U_{q_+}(sl_2) \otimes U_{q_-}(sl_2)$; satisfy BPZ differential equations; OPE can be used inside correlation functions. In the rest of this section we derive the coproduct, comment on the $w$-independence property, provide the result of the computation of OPE coefficients, and explain how to translate these results into the canonical normalization. We do not provide the generalization of the proof of the BPZ equation to the case of 2 quantum groups. However, it is also expected to hold.

**Coproduct and Ward identities** Now one can derive the coproduct $\Delta(F_\pm)$ with the cocycles (144) using exactly the same procedure as described in section 3.3. Similarly to figure 8, we act on two operators $V_{\boldsymbol{\ell}_k,\boldsymbol{m}_k}(x_k) V_{\boldsymbol{\ell}_{k+1},\boldsymbol{m}_{k+1}}(x_{k+1})$ with integral operators $\int dz \mathcal{V}_{\alpha_\pm}(z)$. However,

---

[15]The trivial choice of cocycle, $\varphi = 1$, also leads to a consistent theory. However, the symmetry algebra of that theory will have a twisting of the commutation relations, which spoils the independence of the two quantum groups. For more details about the arising algebra structure see [10]. We do not proceed with this choice because it does not allow for a consistent projection to XXZ$_q$ operators.

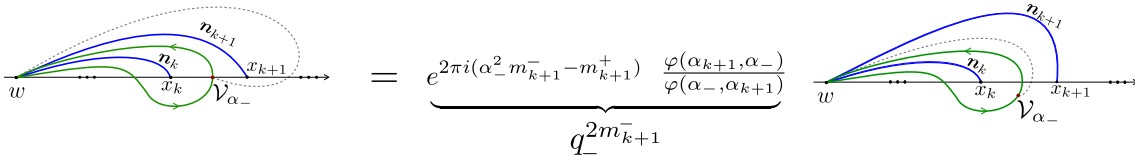

Figure 19: The nontrivial piece in the computation of the coproduct for $\Delta(F_-)$ in the presence of operators charged under $U_{q_+}(sl_2)$. Other steps in the computation are identical to figure 8. Here we can see the importance of the cocycles $\varphi$.

the computation of the phase, appearing from the reordering of operators, is different now. This is where the non-trivial cocycles (144) are important. Let us show what changes in the computation of the phase in the derivation of $\Delta(F_-)$, see figure 19.

The phase is determined from exchanging the ordering of $\mathcal{V}_{\alpha_-}$ and all the operators that constitute $V_{\ell_{k+1}, m_{k+1}}$, which are: $\mathcal{V}_{\alpha_{k+1}}$, $(\ell^-_{k+1} - m^-_{k+1})\, \mathcal{V}_{\alpha_-}$ operators and $(\ell^+_{k+1} - m^+_{k+1})\, \mathcal{V}_{\alpha_+}$ operators. According to figure 3, we determine the phase to be

$$e^{2\pi i \left(\alpha^2_- m^-_{k+1} - m^+_{k+1}\right)} \frac{\varphi(\alpha_{k+1}, \alpha_-)}{\varphi(\alpha_-, \alpha_{k+1})} = e^{2\pi i \alpha^2_- m^-_{k+1}} \underbrace{(-1)^{-2m^+_{k+1} - 2\ell^+_{k+1}}}_{=1} = q_-^{2m^-_{k+1}}, \tag{147}$$

where from (144) the cocycles are given by $\varphi(\alpha_{k+1}, \alpha_-) = (-1)^{-2\ell^+_{k+1}}$ and $\varphi(\alpha_-, \alpha_{k+1}) = 1$. Note that the relation $(-1)^{-2m^+_{k+1} - 2\ell^+_{k+1}} = 1$ is always satisfied, because $\ell^+_{k+1} - m^+_{k+1}$ is an integer.[16]

A similar analysis can be done for the coproduct of $\Delta(F_+)$, which allows us to write

$$\Delta(F_\pm) = F_\pm \otimes q_\pm^{H_\pm} + 1 \otimes F_\pm. \tag{148}$$

Thus, we conclude that the symmetry group that transforms the operators $V_{\ell, m}$ is the product of two quantum groups

$$U_{q_+}(sl_2) \otimes U_{q_-}(sl_2), \qquad q_\pm = e^{i\pi \alpha^2_\pm}. \tag{149}$$

This, in turn, implies that Ward identities have to be satisfied separately for each quantum group. It allows us to generalize Ward identities (52) to,

$$\left\langle X \cdot \left( V_{\ell_1, m_1}(x_1) \dots V_{\ell_n, m_n}(x_n) \right) \right\rangle = 0, \tag{150}$$

with $X = H^\pm, E^\pm, F^\pm$.

**$w$-independence property.** In section 3.2.1, we have provided a direct proof that the integrals (38) are independent of the coordinate $w$. In this section we give just a physical argument relying on the structure of the OPE for the $w$ independence of (146). To build intuition of what could go wrong, let us start with a toy example,

$$\left\langle \mathcal{V}_{2\alpha_0}(w) \mathcal{V}_\alpha(z_1) \mathcal{V}_{-\alpha}(z_2) \right\rangle_{\text{CG}} = \left( \frac{z_1 - w}{z_2 - w} \right)^{4\alpha_0 \alpha} (z_2 - z_1)^{-2\alpha^2}. \tag{151}$$

This correlation function (defined in the free boson theory with a background charge) satisfies the anomalous charge conservation (33). Interestingly, it depends on $w$ even though $\mathcal{V}_{2\alpha_0}$ is a dimension-0 operator. In other words, $\partial \mathcal{V}_{2\alpha_0} = 0$ is not an operator identity in the free boson theory with a background charge. However, this operator identity is only very mildly violated.

---

[16]Had we set all the cocycles to 1, the phase would get an additional sign factor that depends on the quantum number $m^+_{k+1}$ of the second quantum group, which would lead to a different algebraic structure [9].

Namely, $\partial \mathcal{V}_{2\alpha_0}$ has a non-zero two-point function only with $\partial \phi$ (more generally, also with its Virasoro descendants), which is neutral under the $U(1)$ shift symmetry. Note that while $\partial \phi$ is not a primary,

$$L_1(\partial \phi) = 2\sqrt{2}i\alpha_0. \tag{152}$$

It is also not a descendant, since acting with $L_{-1}$ on the identity operator leads to zero,

$$L_{-1}(\mathbb{1}) = 0. \tag{153}$$

This type of behavior is only allowed in non-unitary theories, and it is crucial to have a non-vanishing two-point function with $\partial \mathcal{V}_{2\alpha_0}$. We have

$$\langle \mathcal{V}_{2\alpha_0}(w)\partial \phi(z)\rangle = -\frac{2\sqrt{2}i\alpha_0}{z-w} \quad \Rightarrow \quad \langle \partial \mathcal{V}_{2\alpha_0}(w)\partial \phi(z)\rangle = -\frac{2\sqrt{2}i\alpha_0}{(z-w)^2}. \tag{154}$$

We conclude that any correlator with one insertion of $\partial \mathcal{V}_{2\alpha_0}$ can be nonvanishing only if the OPE of all other insertions has $\partial \phi$ appearing in it. Indeed, in the example (151) we have,

$$\mathcal{V}_\alpha(z_1)\mathcal{V}_{-\alpha}(z_2) = (z_2 - z_1)^{-2\alpha^2}\left[1 + \ldots + A_\alpha(z_2 - z_1)\partial \phi(z_1) + \ldots\right], \tag{155}$$

where the first "..." denotes the Verma module descendants of the identity operator and the second "..." denotes the descendants of $\partial \phi$. The OPE coefficients of the descendants of $\partial \phi$ are all fixed by conformal symmetry and are proportional to $A_\alpha$. One can determine $A_\alpha$ by acting with $L_1$ on both sides of (155). The result is given by

$$A_\alpha = -\sqrt{2}i\alpha. \tag{156}$$

Substituting (155) and (156) into (151) reproduces the correct leading behavior in $z_1 - w$.

Now, to complete the argument for $w$-independence of (146) we only have to show that $\partial \phi$ does not appear in the OPE of all of the rest of the operators. For the purpose of the discussion of the OPE, $\partial \phi$ should be treated as a primary. In other words, even though (152) is satisfied, the appearance of the identity in an OPE does not imply the appearance of $\partial \phi$ in higher orders. For this reason, to detect whether $\partial \phi$ appears in the OPE of $n$ operators, we can work recursively – replacing two operators by any primary (or $\partial \phi$) appearing in their OPE at each step.

We will now give a direct argument that $\partial \phi$ cannot appear in the OPE of $V_{\ell,m}V_{\ell,-m}$ even if their total charge is zero. First, since $\partial \phi$ is in the trivial representation of the quantum group, it only appears in the OPE of two operators which transform under the same quantum group representation. Consider,

$$V_{\ell,m}(0)V_{\ell,-m}(z) = \frac{\widetilde{C}^{(0,0)}_{(\ell,m),(\ell,-m)}}{z^{2h_\ell}}\left[1 + \ldots + A_{\ell,m}z\partial \phi(0) + \ldots\right],$$

and act with $L_1$ once before and once after the OPE (recall that the definition of Virasoro generators in the context of our non-local operators was given around figure 24). This gives,

$$(z^2\partial_z + 2zh_\ell)\left[\frac{1}{(z)^{2h_\ell}}\left[1 + \ldots + A_{\ell,m}z\partial \phi(0) + \ldots\right]\right] = L_1\left[\frac{1}{(z)^{2h_\ell}}\left[1 + \ldots + A_{\ell,m}z\partial \phi(0) + \ldots\right]\right]. \tag{157}$$

Using (152) and matching the coefficients of the identity operator, we find that $A_{\ell,m} = 0$. In any case when the total charge is 0, only vertex operators will appear in the OPE by definition. By iterating the OPE, we verify that $\partial \phi$ does not appear in the OPE of

$$V_{\ell_1,m_1}(z_1)\ldots V_{\ell_n,m_n}(z_n).$$

This completes the general physical argument for the $w$-independence.

**OPE coefficients.** The generalization of the OPE formula (99) to the case with two quantum groups is,

$$V_{\ell_k,m_k}(x_k)\, V_{\ell_{k+1},m_{k+1}}(x_{k+1}) = \sum_{\ell=|m_k+m_{k+1}|}^{\ell_k+\ell_{k+1}} \frac{\widetilde{C}^{(\ell,m_k+m_{k+1})}_{(\ell_k,m_k),(\ell_{k+1},m_{k+1})}}{(x_{k+1}-x_k)^{\Delta_{\ell_k,\ell_{k+1},\ell}}} \left[ V_{\ell,m_k+m_{k+1}}(x_k) + \ldots \right], \quad (158)$$

where bold letters stand for two quantum group versions of the respective quantities. Also,

$$\Delta_{\ell_k,\ell_{k+1},\ell} = h_{2\ell_k^++1,2\ell_k^-+1} + h_{2\ell_{k+1}^++1,2\ell_{k+1}^-+1} - h_{2\ell^++1,2\ell^-+1}, \quad (159)$$

with $h_{r,s}$ defined in (13). The computation of OPE coefficient is the generalization of the steps described in section 3.5 in the presence of two quantum groups. At every step the contour manipulation can be applied separately to the lines associated with $U_{q_-}(sl_2)$ and then to the ones associated with $U_{q_+}(sl_2)$. Similarly to the computation of the coproduct in figure 19, given the cocycles (144) the two types of contours do not affect each other in any way. The result is then obtained by replacing the quantities appearing in (118) by,

$$\begin{aligned}
\boldsymbol{\beta}^{\boldsymbol{n}}_{\boldsymbol{\ell}_{k+1},\boldsymbol{m}_{k+1}} &= \beta^{n^-}_{\ell^-_{k+1},m^-_{k+1}}\Big|_{q=q_-} \beta^{n^+}_{\ell^+_{k+1},m^+_{k+1}}\Big|_{q=q_+}, \\
[\boldsymbol{A}^C_L]^{\boldsymbol{\ell}_k}_{\boldsymbol{h}} &= [A^C_L]^{\ell^-_k}_{h^-}\Big|_{q=q_-} [A^C_L]^{\ell^+_k}_{h^+}\Big|_{q=q_+}, \\
\boldsymbol{\alpha}^{\boldsymbol{\ell}_k,\boldsymbol{\ell}_{k+1}}_{\boldsymbol{t}+\boldsymbol{g},\boldsymbol{g}} &= \alpha^{\ell^-_k,\ell^-_{k+1}}_{t^-+g^-,g^-}\Big|_{q=q_-} \alpha^{\ell^+_k,\ell^+_{k+1}}_{t^++g^+,g^+}\Big|_{q=q_+}, \\
\boldsymbol{\Sigma}(\boldsymbol{t}) &= \Sigma(t^-)\Big|_{q=q_-} \Sigma(t^+)\Big|_{q=q_+}.
\end{aligned} \quad (160)$$

In the last step of the computation we end up with an integral expression which is a generalization of the Selberg integral, but with two distinct sets of integrated variables (A.4). Note that it is not just a product of two Selberg integrals because of the interaction term between the two types of screening charges. The integral was computed in the literature [4], and we provide expression for it in (A.5). The final result for the OPE coefficient is

$$\begin{aligned}
\widetilde{C}^{(\ell,m_k+m_{k+1})}_{(\ell_k,m_k),(\ell_{k+1},m_{k+1})} &= \sum_{n^-=0}^{\min(\ell^-_{k+1}-m^-_{k+1},\,m^-_k+\ell^-_k)} \sum_{n^+=0}^{\min(\ell^+_{k+1}-m^+_{k+1},\,m^+_k+\ell^+_k)} \boldsymbol{\beta}^{\boldsymbol{n}}_{\boldsymbol{\ell}_{k+1},\boldsymbol{m}_{k+1}} [\boldsymbol{A}^C_L]^{\boldsymbol{\ell}_k}_{\boldsymbol{\ell}_k-\boldsymbol{m}_k+\boldsymbol{n}} \\
&\times (-1)^{2t^-\ell^+_k}(-1)^{2t^+\ell^-_{k+1}} i^{4\ell^+_k\ell^-_{k+1}} \\
&\times \sum_{g^-=g^-_0}^{\ell^-_k-m^-_k+n^-} \sum_{g^+=g^+_0}^{\ell^+_k-m^+_k+n^+} \frac{\boldsymbol{\alpha}^{\boldsymbol{\ell}_k,\boldsymbol{\ell}_{k+1}}_{\boldsymbol{t}+\boldsymbol{g},\boldsymbol{g}}}{t^+!t^-!} \boldsymbol{\Sigma}(\boldsymbol{t}) J_{t^-,t^+}(a,b;\alpha^2_-)\delta_{\boldsymbol{\ell}_k-\boldsymbol{m}_k+\boldsymbol{n}-\boldsymbol{g},\boldsymbol{t}}.
\end{aligned} \quad (161)$$

Here, we use the notations,

$$\begin{aligned}
\boldsymbol{t} &= \boldsymbol{\ell}_k + \boldsymbol{\ell}_{k+1} - \boldsymbol{\ell}, \\
g^\pm_0 &= \max\left(0, \ell^\pm_k - m^\pm_k - 2\ell^\pm_{k+1} + n^\pm\right), \\
a &= 1 - 2\ell^-_k\alpha^2_- + 2\ell^+_k, \\
b &= 1 - 2\ell^-_{k+1}\alpha^2_- + 2\ell^+_{k+1}.
\end{aligned} \quad (162)$$

The signs in the second line in (161) come from the effect of the cocycles (144) on the final integral expression.

In the case when 2 out of 3 operators are highest weight w.r.t both quantum groups this again simplifies to,

$$\begin{aligned}
\widetilde{C}^{(\boldsymbol{m}+\boldsymbol{\ell}_{k+1},\boldsymbol{m}+\boldsymbol{\ell}_{k+1})}_{(\ell_k,m),(\ell_{k+1},\ell_{k+1})} &= [A^C_S]^{\boldsymbol{\ell}_k}_{\boldsymbol{\ell}_k-\boldsymbol{m}}(-1)^{2(\ell^-_k-m^-)\ell^+_k+2(\ell^+_k-m^+)\ell^-_{k+1}} i^{4\ell^+_k\ell^-_{k+1}} \\
&\times \frac{\boldsymbol{\alpha}^{\boldsymbol{\ell}_k,\boldsymbol{\ell}_{k+1}}_{\boldsymbol{\ell}_k-\boldsymbol{m},0}}{(\ell^-_k-m^-)!(\ell^+_k-m^+)!} J_{\ell^-_k-m^-,\ell^+_k-m^+}(a,b;\alpha^2_-),
\end{aligned} \quad (163)$$

where $[A_S^C]_{\ell_k-m}^{\ell_k} = [A_L^C]_{\ell_k-m}^{\ell_k}\Sigma(\ell_k-m)$.

Using the explicit expressions of the factors in (163), we get the general OPE coefficient which generalizes the $(1,s)$ result given in (122):

$$
\widetilde{C}_{(\ell_1,\ell_3-\ell_2),(\ell_2,\ell_2)}^{(\ell_3,\ell_3)} \tag{164}
$$

$$
= i^{4\ell_1^+\ell_2^-}\left((-1)^{2\ell_2^-}2\pi i\right)^{\ell_1^++\ell_2^+-\ell_3^+}\left((-1)^{2\ell_1^+}2\pi i\right)^{\ell_1^-+\ell_2^--\ell_3^-}\alpha_+^{4(\ell_1^-+\ell_2^--\ell_3^-)(\ell_1^++\ell_2^+-\ell_3^+)}
$$

$$
\times\prod_{k=1}^{\ell_1^-+\ell_2^--\ell_3^-}\left(\frac{\Gamma\left((\ell_1^-+\ell_2^-+\ell_3^-+2-k)\alpha_-^2-(2\ell_3^++1)\right)\Gamma\left(1-\alpha_-^2\right)}{\Gamma\left((2\ell_1^-+1-k)\alpha_-^2-2\ell_1^+\right)\Gamma\left((2\ell_2^-+1-k)\alpha_-^2-2\ell_2^+\right)\Gamma\left(1-k\alpha_-^2\right)}\right)
$$

$$
\times\prod_{k=1}^{\ell_1^++\ell_2^+-\ell_3^+}\left(\frac{\Gamma\left((\ell_1^++\ell_2^++\ell_3^++2-k)\alpha_+^2-(\ell_1^-+\ell_2^-+\ell_3^-+1)\right)}{\Gamma\left((2\ell_1^++1-k)\alpha_+^2-(\ell_1^--\ell_2^-+\ell_3^-)\right)\Gamma\left((2\ell_2^++1-k)\alpha_+^2-(\ell_2^--\ell_1^-+\ell_3^-)\right)}\right.
$$

$$
\left.\times\frac{\Gamma\left(1-\alpha_+^2\right)}{\Gamma\left(1+\ell_1^-+\ell_2^--\ell_3^--k\alpha_+^2\right)}\right).
$$

Here, $\alpha_+ = -1/\alpha_-$, following from (27). We can also rewrite (164) as

$$
\widetilde{C}_{(\ell_1,\ell_3-\ell_2),(\ell_2,\ell_2)}^{(\ell_3,\ell_3)} = \left[\widetilde{C}_{(\ell_1^+,\ell_3^+-\ell_2^+),(\ell_2^+,\ell_2^+)}^{(\ell_3^+,\ell_3^+)}\right]_+\times\left[\widetilde{C}_{(\ell_1^-,\ell_3^--\ell_2^-),(\ell_2^-,\ell_2^-)}^{(\ell_3^-,\ell_3^-)}\right]_- \tag{165}
$$

$$
\times i^{4\ell_1^+\ell_2^-}(-1)^{(\ell_1^++\ell_2^+-\ell_3^+)(\ell_1^-+\ell_2^--\ell_3^-)+2\ell_1^+(\ell_1^-+\ell_2^--\ell_3^-)+2\ell_2^-(\ell_1^++\ell_2^+-\ell_3^+)}
$$

$$
\times\frac{Y(0,2\ell_1)Y(0,2\ell_2)Y(1,2\ell_3)}{Y(0,\ell_1-\ell_2+\ell_3)Y(0,\ell_2-\ell_1+\ell_3)Y(0,\ell_1+\ell_2-\ell_3)Y(1,\ell_1+\ell_2+\ell_3)},
$$

where the function $Y(n,\ell)$ is defined by

$$
Y(n,\ell) := \prod_{k^+=1+n}^{\ell^++n}\prod_{k^-=1+n}^{\ell^-+n}\left(k^+\alpha_+ + k^-\alpha_-\right), \qquad \ell=(\ell^+,\ell^-). \tag{166}
$$

Further simplification happens for the calculation of the OPE between the operators $V_{\ell,-\ell}$, $V_{\ell,\ell}$ and $\mathbb{1}$. In that case, the OPE coefficient (164) reduces to:

$$
\widetilde{C}_{(\ell,-\ell),(\ell,\ell)}^{(0,0)} = (-i)^{4\ell^+\ell^-}(2\pi i)^{2\ell^++2\ell^-}\frac{(2\ell^++1)\alpha_+ + (2\ell^-+1)\alpha_-}{\alpha_+ + \alpha_-}
$$

$$
\times\prod_{k=1}^{2\ell^+}\left(\frac{\Gamma\left((2\ell^++2-k)\alpha_+^2-1\right)\Gamma\left(1-\alpha_+^2\right)}{\Gamma\left((2\ell^++1-k)\alpha_+^2\right)^2\Gamma\left(1-k\alpha_+^2\right)}\frac{k\alpha_+ + \alpha_-}{(k+1)\alpha_+ + (2\ell^-+1)\alpha_-}\right) \tag{167}
$$

$$
\times\prod_{k=1}^{2\ell^-}\left(\frac{\Gamma\left((2\ell^-+2-k)\alpha_-^2-1\right)\Gamma\left(1-\alpha_-^2\right)}{\Gamma\left((2\ell^-+1-k)\alpha_-^2\right)^2\Gamma\left(1-k\alpha_-^2\right)}\frac{\alpha_+ + k\alpha_-}{(2\ell^++1)\alpha_+ + (k+1)\alpha_-}\right),
$$

which generalizes (123). For the convenience of the discussion below, we rewrite (167) as

$$
\widetilde{C}_{(\ell,-\ell),(\ell,\ell)}^{(0,0)} = \left[\widetilde{C}_{(\ell^+,-\ell^+),(\ell^+,\ell^+)}^{(0,0)}\right]_+\times\left[\widetilde{C}_{(\ell^-,-\ell^-),(\ell^-,\ell^-)}^{(0,0)}\right]_-\times(-i)^{4\ell^+\ell^-}\frac{Y(0,2\ell)}{Y(1,2\ell)}. \tag{168}
$$

Here, the factors $[\ldots]_+$ and $[\ldots]_-$ represent the OPE coefficients in the $(r,1)$ and $(1,s)$ subsectors, respectively, as given in (123) and its "+"-analogue.

**Canonical normalization.**  Let us now convert the results of this section into canonical normalization. The procedure is similar to the one described in section 3.6, but now the group-theoretical structures invariant under $U_{q_+}(sl_2)\otimes U_{q_-}(sl_2)$ are products of two of the structures

appearing in the single quantum group case. We denote the canonically normalized operators $\widetilde{V}_{\ell,m}$ and define them in a way such that their two-point functions are,

$$\left\langle \widetilde{V}_{\ell,m_1}(0)\widetilde{V}_{\ell,m_2}(1)\right\rangle = \begin{bmatrix} \ell^+ & \ell^+ & 0 \\ m_1^+ & m_2^+ & 0 \end{bmatrix}_{q_+} \begin{bmatrix} \ell^- & \ell^- & 0 \\ m_1^- & m_2^- & 0 \end{bmatrix}_{q_-}. \tag{169}$$

This is a generalization of (6). The OPE coefficients of these operators generalize (7) and are given by,

$$\widetilde{V}_{\ell_1,m_1}(0)\widetilde{V}_{\ell_2,m_2}(1) = \sum_{\ell_3=|\ell_1-\ell_2|}^{\ell_1+\ell_2} \begin{bmatrix} \ell_1^+ & \ell_2^+ & \ell_3^+ \\ m_1^+ & m_2^+ & m_1^+ + m_2^+ \end{bmatrix}_{q_+} \begin{bmatrix} \ell_1^- & \ell_2^- & \ell_3^- \\ m_1^- & m_2^- & m_1^- + m_2^- \end{bmatrix}_{q_-} \tag{170}$$
$$\times C_{(r_1,s_1)(r_2,s_2)(r_3,s_3)}\left[\widetilde{V}_{\ell_3,m_1+m_2}(0)+\dots\right].$$

Here, as usual, we use the notation $r_i = 2\ell_i^+ + 1, s_i = 2\ell_i^- + 1$.

The relation between operators in two different normalizations is given by

$$V_{\ell,m} = \widetilde{\mathcal{N}}_{\ell,m}\widetilde{V}_{\ell,m} \equiv \mathcal{N}_\ell \left(\prod_{m=m^++1}^{\ell^+} f_m^{\ell^+}\right)\left(\prod_{n=m^-+1}^{\ell^-} f_n^{\ell^-}\right)\widetilde{V}_{\ell,m}. \tag{171}$$

As before, the normalization constant $\mathcal{N}_\ell$ can be found from comparing the two-point functions,

$$\mathcal{N}_\ell^2 = \frac{\widetilde{C}_{(\ell,-\ell),(\ell,\ell)}^{(0,0)}}{\left(\prod_{m=-\ell^-+1}^{\ell^-} f_m^{\ell^-}\right)\left(\prod_{n=-\ell^++1}^{\ell^+} f_m^{\ell^+}\right)\begin{bmatrix} \ell^+ & \ell^+ & 0 \\ -\ell^+ & \ell^+ & 0 \end{bmatrix}_{q_+}\begin{bmatrix} \ell^- & \ell^- & 0 \\ -\ell^- & \ell^- & 0 \end{bmatrix}_{q_-}}, \tag{172}$$

where the coefficient $\widetilde{C}_{(\ell,-\ell),(\ell,\ell)}^{(0,0)}$ was previously computed (167).

Using (127), (172) and the factorization property (168) of $\widetilde{C}_{(\ell,-\ell),(\ell,\ell)}^{(0,0)}$, we have

$$\mathcal{N}_\ell^2 = \left[\mathcal{N}_{\ell^+}^2\right]_+ \left[\mathcal{N}_{\ell^-}^2\right]_- \times (-i)^{4\ell^+\ell^-}\frac{Y(0,2\ell)}{Y(1,2\ell)}. \tag{173}$$

Here, $\left[\mathcal{N}_{\ell^+}^2\right]_+$ and $\left[\mathcal{N}_{\ell^-}^2\right]_-$ are given by (128) in the "+" and "−" conventions, respectively.

By repeating the procedure of section 3.6, we have here

$$C_{(r_1,s_1)(r_2,s_2)(r_3,s_3)} = \frac{\mathcal{N}_{\ell_3}\mathcal{N}_{\ell_1}^{-1}\mathcal{N}_{\ell_2}^{-1}\widetilde{C}_{(\ell_1,\ell_3-\ell_2),(\ell_2,\ell_2)}^{(\ell_3,\ell_3)}}{\left(\prod_{m^+=\ell_3^+-\ell_2^++1}^{\ell_1^+}f_{m^+}^{\ell_1^+}(q_+)\right)(+\leftrightarrow-)\begin{bmatrix} \ell_1^+ & \ell_2^+ & \ell_3^+ \\ \ell_3^+-\ell_2^+ & \ell_2^+ & \ell_3^+ \end{bmatrix}_{q_+}[+\leftrightarrow-]_{q_-}}. \tag{174}$$

Then using eqs. (137), (165), (173) and (174), we get

$$C_{(r_1,s_1)(r_2,s_2)(r_3,s_3)} = C_{(r_1,1)(r_2,1)(r_3,1)}\, C_{(1,s_1)(1,s_2)(1,s_3)}\sqrt{\frac{Y(0,2\ell_1)Y(0,2\ell_2)Y(0,2\ell_3)}{Y(1,2\ell_1)Y(1,2\ell_2)Y(1,2\ell_3)}} \tag{175}$$
$$\times e^{i\pi(\ell_1^+\ell_1^-+\ell_2^+\ell_2^--\ell_3^+\ell_3^--2\ell_1^+\ell_2^-)}(-1)^{(\ell_1^++\ell_3^+-\ell_2^+)(\ell_2^-+\ell_3^--\ell_1^-)}$$
$$\times \frac{Y(1,2\ell_1)Y(1,2\ell_2)Y(1,2\ell_3)}{Y(0,\ell_1-\ell_2+\ell_3)Y(0,\ell_2-\ell_1+\ell_3)Y(0,\ell_1+\ell_2-\ell_3)Y(1,\ell_1+\ell_2+\ell_3)},$$
$$\text{where}\quad \ell_i = (\ell_i^+,\ell_i^-) = \left(\frac{r_i-1}{2},\frac{s_i-1}{2}\right).$$

Here $C_{(1,s_1)(1,s_2)(1,s_3)}$ is the chiral OPE coefficient given in (138), the function $Y$ is given in (166), and $C_{(r_1,1)(r_2,1)(r_3,1)}$ is the "+"-analogue of (138):

$$
\begin{aligned}
C_{(r_1,1)(r_2,1)(r_3,1)} = \frac{1}{[r_3-1]_{q_+}!} & \sqrt{\frac{\left[\frac{r_1+r_3-r_2-1}{2}\right]_{q_+}!\left[\frac{r_2+r_3-r_1-1}{2}\right]_{q_+}!\left[\frac{r_1+r_2+r_3-1}{2}\right]_{q_+}!}{\left([r_1]_{q_+}[r_2]_{q_+}[r_3]_{q_+}\right)^{1/2}\left[\frac{r_1+r_2-r_3-1}{2}\right]_{q_+}!}} \\
& \times \prod_{k=1}^{\frac{r_1+r_2-r_3-1}{2}} \frac{\Gamma\left(\left(\frac{r_1+r_2+r_3+1}{2}-k\right)\alpha_+^2-1\right)}{\Gamma\left((r_1-k)\alpha_+^2\right)\Gamma\left((r_2-k)\alpha_+^2\right)\Gamma(1-k\alpha_+^2)} \\
& \times \sqrt{\left(\prod_{k=1}^{r_1-1}\frac{\Gamma\left(k\alpha_+^2\right)^2\Gamma(1-k\alpha_+^2)}{\Gamma\left((k+1)\alpha_+^2-1\right)}\right)\left(\prod_{k=1}^{r_2-1}\frac{\Gamma\left(k\alpha_+^2\right)^2\Gamma(1-k\alpha_+^2)}{\Gamma\left((k+1)\alpha_+^2-1\right)}\right)} \\
& \times \sqrt{\prod_{k=1}^{r_3-1}\frac{\Gamma\left((k+1)\alpha_+^2-1\right)}{\Gamma\left(k\alpha_+^2\right)^2\Gamma(1-k\alpha_+^2)}}, \\
\alpha_+ = -\sqrt{\frac{\mu+1}{\mu}}, \qquad & q_+ = e^{i\pi\frac{\mu+1}{\mu}}.
\end{aligned}
\tag{176}
$$

In the right-hand side of (175), the first line is a multi-valued function, while the second and third lines are single-valued. Similar to the case of $C_{(1,s_1)(1,s_2)(1,s_3)}$ in section 3.6, we fix the ambiguity by choosing specific branches for the normalization factor $\mathcal{N}_\ell$ and the $q_\pm$-numbers. After fixing the branch, the first line of (175) is positive at the $c=1$ point. We leave the technical details to appendix E.

Similarly to the single quantum group case, these OPE coefficients are also related to the OPE coefficients of minimal models by

$$
\begin{aligned}
C_{(r_1,s_1)(r_2,s_2)(r_3,s_3)}^{\mathrm{MM}} &= i^{-4(\ell_1^-\ell_1^++\ell_2^-\ell_2^+-\ell_3^-\ell_3^+)}(-1)^{4\ell_1^+\ell_2^-}(C_{(r_1,s_1)(r_2,s_2)(r_3,s_3)})^2 \\
&= C_{(r_1,s_1)(r_2,s_2)(r_3,s_3)}C_{(r_2,s_2)(r_1,s_1)(r_3,s_3)},
\end{aligned}
\tag{177}
$$

where $C^{\mathrm{MM}}$ takes a positive value at $c=1$. Note that in the second line, the indices of the second OPE coefficient are swapped.

**R-matrix.** Although the global symmetry of the above chiral theory is $U_{q_+}(sl_2)\otimes U_{q_-}(sl_2)$, the braid matrix of the theory is not simply the tensor product of two $R$-matrices – one related to $U_{q_+}(sl_2)$ and one related to $U_{q_-}(sl_2)$. By an explicit computation using the highest-weight operators, one can verify that the braid matrix of the theory is given by

$$
\begin{aligned}
[R_{\ell_2,\ell_1}]_{m_2,m_1}^{m_2',m_1'}\widetilde{V}_{\ell_1,m_1}(x_1)\widetilde{V}_{\ell_2,m_2}(x_2) &= \widetilde{V}_{\ell_2,m_2}(x_2)\widetilde{V}_{\ell_1,m_1}(x_1), \\
\text{with}\quad R_{\ell_1,\ell_2} &= e^{-i\pi H_-\otimes H_+}R_{\ell_1^+,\ell_2^+}(q_+)R_{\ell_1^-,\ell_2^-}(q_-).
\end{aligned}
\tag{178}
$$

The additional $\mathbb{Z}_2$ factor $e^{-i\pi H_-\otimes H_+}$ commutes with all other terms and equals one when either $\ell_1^-$ or $\ell_2^+$ is an integer. It modifies the spin selection rule (10) to

$$
\mathrm{spin} = \frac{\ell^+(\ell^++1)}{\pi i}\log q_+ + \frac{\ell^-(\ell^-+1)}{\pi i}\log q_- - \ell^+ - \ell^- - 2\ell^+\ell^- + \mathbb{Z},
\tag{179}
$$

and also affects the transformation rule of the OPE coefficients under permutations of indices. This $\mathbb{Z}_2$ factor does not appear in the construction of XXZ$_q$ operators in the next subsection, since $\ell_-$ is always an integer in that context. However, in the case of chiral CFT, it is essential for reproducing the correct spin $h_{r,s}$ when both $r\equiv 2\ell^+ + 1$ and $s\equiv 2\ell^- + 1$ are even.

**Crossing symmetry and the fusion kernel.** Now let us go back to the question, discussed in [1], of how crossing symmetry of $U_q(sl_2)$ symmetric theories explains the appearance of $U_q(sl_2)$ $6j$-symbols in the fusion kernel of Virasoro blocks. Let us consider the four-point function of canonically normalized operators $\langle \widetilde{V}_{\ell_1,m_1}(x_1)\dots \widetilde{V}_{\ell_4,m_4}(x_4)\rangle$. The condition of crossing symmetry between s-channel and t-channel expansions is

$$\sum_{r,s} C_{(r_1,s_1)(r_2,s_2)(r,s)} C_{(r,s)(r_3,s_3)(r_4,s_4)} T_\ell^{(s)} \mathcal{F}_{r,s}^{(s)}(z) = \sum_{r',s'} C_{(r_2,s_2)(r_3,s_3)(r',s')} C_{(r_1,s_1)(r',s')(r_4,s_4)} T_{\ell'}^{(t)} \mathcal{F}_{r',s'}^{(t)}(z),$$
(180)

where $C$ is the chiral OPE coefficient defined in (170); $T_\ell^{(s)}$ and $T_{\ell'}^{(t)}$ are the s- and t-channel quantum group invariant tensors, given by

$$T_\ell^{(s)} = T_{\ell^+}^{(s)} T_{\ell^-}^{(s)}, \qquad T_\ell^{(t)} = T_{\ell^+}^{(t)} T_{\ell^-}^{(t)},$$
(181)

where the $T_{\ell^+}$ are defined by

$$T_{\ell^+}^{(s)} = \sum_{m_{12}^+} \begin{bmatrix} \ell_1^+ & \ell_2^+ & \ell^+ \\ m_1^+ & m_2^+ & m_{12}^+ \end{bmatrix}_{q_+} \begin{bmatrix} \ell^+ & \ell_3^+ & \ell_4^+ \\ m_{12}^+ & m_3^+ & -m_4^+ \end{bmatrix}_{q_+} \begin{bmatrix} \ell_4^+ & \ell_4^+ & 0 \\ -m_4^+ & m_4^+ & 0 \end{bmatrix}_{q_+},$$

$$T_{\ell^+}^{(t)} = \sum_{m_{23}^+} \begin{bmatrix} \ell_2^+ & \ell_3^+ & \ell^+ \\ m_2^+ & m_3^+ & m_{23}^+ \end{bmatrix}_{q_+} \begin{bmatrix} \ell_1^+ & \ell^+ & \ell_4^+ \\ m_1^+ & m_{23}^+ & -m_4^+ \end{bmatrix}_{q_+} \begin{bmatrix} \ell_4^+ & \ell_4^+ & 0 \\ -m_4^+ & m_4^+ & 0 \end{bmatrix}_{q_+},$$
(182)

and the $T_{\ell^-}$ are defined by substituting $\ell_i^+ \to \ell_i^-$, $m_i^+ \to m_i^-$, $q_+ \to q_-$ in the previous expressions. $\mathcal{F}_{r,s}^{(s)}$ and $\mathcal{F}_{r,s}^{(t)}$ are s- and t-channel Virasoro blocks for an operator with weight $h_{r,s}$; we remind the reader of the identification $r_i = 2\ell_i^+ + 1$ and $s_i = 2\ell_i^- + 1$. The quantum group invariant tensors $T_{r,s}^{(s)}$ and $T_{r',s'}^{(t)}$ are related by the product of $6j$-symbols

$$T_{\ell'}^{(t)} = \sum_{\ell^+,\ell^-} \begin{Bmatrix} \ell_1^+ & \ell_2^+ & \ell^+ \\ \ell_3^+ & \ell_4^+ & \ell'^+ \end{Bmatrix}_{q_+} \begin{Bmatrix} \ell_1^- & \ell_2^- & \ell^- \\ \ell_3^- & \ell_4^- & \ell'^- \end{Bmatrix}_{q_-} T_\ell^{(s)}.$$
(183)

Using (180) and (183), we get the fusing kernel of the Virasoro blocks

$$\mathcal{F}_{r',s'}^{(t)} = \sum_{r,s} \frac{C_{(r_1,s_1)(r_2,s_2)(r,s)} C_{(r,s)(r_3,s_3)(r_4,s_4)}}{C_{(r_2,s_2)(r_3,s_3)(r',s')} C_{(r_1,s_1)(r',s')(r_4,s_4)}} \begin{Bmatrix} \frac{r_1-1}{2} & \frac{r_2-1}{2} & \frac{r-1}{2} \\ \frac{r_3-1}{2} & \frac{r_4-1}{2} & \frac{r'-1}{2} \end{Bmatrix}_{q_+} \begin{Bmatrix} \frac{s_1-1}{2} & \frac{s_2-1}{2} & \frac{s-1}{2} \\ \frac{s_3-1}{2} & \frac{s_4-1}{2} & \frac{s'-1}{2} \end{Bmatrix}_{q_-} \mathcal{F}_{r,s}^{(s)}.$$
(184)

This gives the same structure as the crossing kernel of Virasoro blocks, known in the literature, see e.g. [24, 27, 28], whose explicit form can also be found in section 4 of the companion paper [1].[17] We remind the readers that the Virasoro blocks are just the functions of the central charge and the weights of operators $h_i$. This means that, if we have a $c < 1$ theory with degenerate operators of dimension $h_{r,s}$, no matter its global symmetry, $U_q(sl_2)$ $6j$-symbols will appear via the fusion kernel of Virasoro blocks. We discuss the implications for generalized minimal models at the end of section 4.3.

## 4.2 General OPE coefficients of XXZ$_q$ CFT

Now we would like to apply the Coulomb gas formalism developed in section 4.1 to construct the CFT data of the XXZ$_q$ CFT. Recall that in this theory there is only one quantum group acting non-trivially, $U_{q_-}(sl_2)$, and generic operators are non-chiral, see (15). In order to construct those we combine the chiral theory of section 4.1 with an anti-chiral counterpart, and choose

---

[17]The notation there is slightly different from the notation used here: $q_+$ becomes $\tilde{q}$ and $q_-$ becomes $q$ in [1].

a combination of operators that have a trivial contribution to monodromies from the $R$-matrix corresponding to $U_{q_+}(sl_2)$. The procedure described below realizes a "projection" to operators transforming under only a single quantum group.

Let us start with defining an anti-chiral counterpart of section 4.1. We would like to consider operators $\overline{V}_{\ell,m}$ of dimensions $(0, h_{r,s})$. Their correlation functions can be defined as,

$$\left\langle \overline{V}_{\ell_1,m_1}(\bar{x}_1) \dots \overline{V}_{\ell_n,m_n}(\bar{x}_n) \right\rangle \equiv \int_{\overline{\Gamma}} \prod_{j=1}^{n} d\bar{z}^{(j)} d\bar{u}^{(j)} \Big\langle \overline{\mathcal{V}}_{2\alpha_0}(\bar{w}) \cdot \overline{\mathcal{V}}_{\alpha_1}(\bar{x}_1) \overline{\mathcal{V}}_{\alpha_-}\left(\bar{z}^{(1)}\right) \overline{\mathcal{V}}_{\alpha_+}\left(\bar{u}^{(1)}\right) \dots$$
$$\times \overline{\mathcal{V}}_{\alpha_n}(\bar{x}_n) \overline{\mathcal{V}}_{\alpha_-}\left(\bar{z}^{(n)}\right) \overline{\mathcal{V}}_{\alpha_+}\left(\bar{u}^{(n)}\right) \Big\rangle_{\overline{\mathrm{CG}}}, \quad (185)$$

where $\overline{\Gamma}$ is the mirror image of the contour $\Gamma$ appearing in definition (146) with respect to the real axis. Here, by "$\overline{\mathrm{CG}}$" we mean that we use the inverse phase factor for the OPE of the anti-chiral vertex operator compared to the chiral case (26):

$$\overline{\mathcal{V}}_\alpha(\bar{z}_1) \overline{\mathcal{V}}_\beta(\bar{z}_2) = \varphi(\alpha,\beta)^{-1}(\bar{z}_2 - \bar{z}_1)^{2\alpha\beta} \overline{\mathcal{V}}_{\alpha+\beta}(\bar{z}_1) + \dots \quad (186)$$

The resulting anti-chiral operators transform in representations of $U_{q_+^{-1}}(sl_2) \otimes U_{q_-^{-1}}(sl_2)$. We denote their canonically normalized versions as $\overline{\widetilde{V}}_{\ell,m}$.

For real $c \leqslant 1$, the anti-chiral integrals (185) are the complex conjugations of the chiral integrals (146). Furthermore, since $q_+$ and $q_-$ are on the unit circle, they are mapped to $q_+^{-1}$ and $q_-^{-1}$ after complex conjugation. Therefore, all the related quantities ($q^n$, $q$-numbers, quantum Clebsch-Gordon coefficients) simply change $q_\pm$ to $q_\pm^{-1}$ in their expressions. This leads to the following conclusion for the OPE of canonically normalized anti-chiral operators:

$$\overline{\widetilde{V}}_{\ell_1,m_1}(0) \overline{\widetilde{V}}_{\ell_2,m_2}(1) = \sum_{\ell_3=|\ell_1-\ell_2|}^{\ell_1+\ell_2} \begin{bmatrix} \ell_1^+ & \ell_2^+ & \ell_3^+ \\ m_1^+ & m_2^+ & m_1^+ + m_2^+ \end{bmatrix}_{q_+^{-1}} \begin{bmatrix} \ell_1^- & \ell_2^- & \ell_3^- \\ m_1^- & m_2^- & m_1^- + m_2^- \end{bmatrix}_{q_-^{-1}} \quad (187)$$
$$\times \left( C_{(r_1,s_1)(r_2,s_2)(r_3,s_3)} \right)^* \left[ \overline{\widetilde{V}}_{\ell_3,m_1+m_2}(0) + \dots \right], \quad \text{for real } c \leqslant 1.$$

Here, we use again the notation $r_i = 2\ell_i^+ + 1, s_i = 2\ell_i^- + 1$.

Now we would like to analytically continue (187) to a generic complex central charge $c$. All the factors except for $\left( C_{(r_1,s_1)(r_2,s_2)(r_3,s_3)} \right)^*$ are already analytic in $c$. For $\left( C_{(r_1,s_1)(r_2,s_2)(r_3,s_3)} \right)^*$, we use (177) and the fact that $C^{\mathrm{MM}}$ is real and positive when $c$ is real and very close to (but less than or equal to) 1. This gives

$$\left( C_{(r_1,s_1)(r_2,s_2)(r_3,s_3)} \right)^* = C_{(r_2,s_2)(r_1,s_1)(r_3,s_3)}, \quad \text{for real } c \leqslant 1. \quad (188)$$

Together with the properties of the quantum Clebsch-Gordon coefficients

$$\begin{bmatrix} \ell_1 & \ell_2 & \ell_3 \\ m_1 & m_2 & m_3 \end{bmatrix}_{q^{-1}} = (-1)^{\ell_1+\ell_2-\ell_3} \begin{bmatrix} \ell_1 & \ell_2 & \ell_3 \\ -m_1 & -m_2 & -m_3 \end{bmatrix}_q, \quad (189)$$

we get

$$\overline{\widetilde{V}}_{\ell_1,m_1}(0) \overline{\widetilde{V}}_{\ell_2,m_2}(1) = \sum_{\ell_3=|\ell_1-\ell_2|}^{\ell_1+\ell_2} \begin{bmatrix} \ell_1^+ & \ell_2^+ & \ell_3^+ \\ -m_1^+ & -m_2^+ & -m_1^+ - m_2^+ \end{bmatrix}_{q_+} \begin{bmatrix} \ell_1^- & \ell_2^- & \ell_3^- \\ -m_1^- & -m_2^- & -m_1^- - m_2^- \end{bmatrix}_{q_-}$$
$$\times (-1)^{\ell_1^+ + \ell_2^+ - \ell_3^+ + \ell_1^- + \ell_2^- - \ell_3^-} C_{(r_2,s_2)(r_1,s_1)(r_3,s_3)} \left[ \overline{\widetilde{V}}_{\ell_3,m_1+m_2}(0) + \dots \right]. \quad (190)$$

In (190), all the factors are analytic in $c$.

Now, we construct the operators (15) appearing in the continuum limit of the $\mathrm{XXZ}_q$ spin chain as follows,

$$W^m_{r,2\ell+1}(x,\bar{x}) := e^{i\pi(\ell^++\ell^-)} \sum_{m_+=-\ell^+}^{\ell^+} \widetilde{V}_{(\ell^+,\ell),(m^+,m)}(x)\overline{\widetilde{V}}_{(\ell^+,0),(-m^+,0)}(\bar{x}), \tag{191}$$

for any integer $r = 2\ell^+ + 1$. This definition, together with the OPE of chiral operators (170), the OPE of anti-chiral operators (190), the permutation symmetry of $C_{(r_1,1),(r_2,1),(r_3,1)}$ and the orthogonality relation of Clebsch-Gordan coefficients,

$$\sum_{m_1,m_2} \begin{bmatrix} \ell_1 & \ell_2 & \ell \\ m_1 & m_2 & m \end{bmatrix}_q \begin{bmatrix} \ell_1 & \ell_2 & \ell' \\ m_1 & m_2 & m' \end{bmatrix}_q = \delta_{\ell,\ell'}\delta_{m,m'}, \tag{192}$$

leads to the OPE,

$$W^{m_1}_{r_1,2\ell_1+1}(z_1,\bar{z}_1)W^{m_2}_{r_2,2\ell_2+1}(z_2,\bar{z}_2) = \sum_{\ell=|m_1+m_2|}^{\ell_1+\ell_2} \sum_{\substack{r_3=|r_1-r_2|+1 \\ r_1+r_2+r_3 \text{ odd}}}^{r_1+r_2-1} \frac{C_{(r_1,2\ell_1+1),(r_2,2\ell_2+1),(r_3,2\ell+1)}C_{(r_1,1),(r_2,1),(r_3,1)}}{z_{21}^{h_{123}}\bar{z}_{21}^{\bar{h}_{123}}}$$

$$\times \begin{bmatrix} \ell_1 & \ell_2 & \ell \\ m_1 & m_2 & m_1+m_2 \end{bmatrix}_{q_-} \left(W^{m_1+m_2}_{r_3,2\ell+1}(z_1,\bar{z}_1) + \ldots\right), \tag{193}$$

matching the expected structure of an OPE in a theory with a single quantum group symmetry (7). This structure, together with the values of the spins $h_{r,s} - h_{r,1}$, is consistent with the assumption that the braid matrix of the theory is exactly the $R$-matrix of the quantum group $U_{q_-}(sl_2)$, and the braiding structure under $U_{q_+}(sl_2)$ trivializes (see the discussion below). Note that however the constructed operators are not singlets under $U_{q_+}(sl_2)$, i.e. the action of generators associated with this quantum group transforms them into some other operators, that generically do not belong to our theory.

Comparing the result of (193) with the general OPE formula for the $\mathrm{XXZ}_q$ operators (16), we observe that OPE coefficients are given by

$$C^{\mathrm{XXZ}}_{(r_1,s_1),(r_2,s_2),(r_3,s_3)} = C_{(r_1,s_1),(r_2,s_2),(r_3,s_3)}C_{(r_1,1),(r_2,1),(r_3,1)}, \tag{194}$$

where $s_i$ are taken to be odd, and the expression for $C_{(r_1,s_1),(r_2,s_2),(r_3,s_3)}$ is given in (175). This result matches the predictions from conformal bootstrap (18). Moreover, the Coulomb gas computations provide expressions without any sign ambiguity which fixes the sign for $C_{(r_1,s_1),(r_2,s_2),(r_3,s_3)}$.

By (175), we can rewrite (194) as

$$C^{\mathrm{XXZ}}_{(r_1,s_1),(r_2,s_2),(r_3,s_3)} = i^{2\ell_1^+\ell_1^-+2\ell_2^+\ell_2^--2\ell_3^+\ell_3^-} \left(C_{(r_1,1),(r_2,1),(r_3,1)}\right)^2 \tag{195}$$

$$\times C_{(1,s_1),(1,s_2),(1,s_3)}\sqrt{\frac{Y(0,2\ell_1)Y(0,2\ell_2)Y(0,2\ell_3)}{Y(1,2\ell_1)Y(1,2\ell_2)Y(1,2\ell_3)}}$$

$$\times \frac{(-1)^{(\ell_1^++\ell_3^+-\ell_2^+)(\ell_2^-+\ell_3^--\ell_1^-)-2\ell_1^+\ell_2^-}Y(1,2\ell_1)Y(1,2\ell_2)Y(1,2\ell_3)}{Y(0,\ell_1-\ell_2+\ell_3)Y(0,\ell_2-\ell_1+\ell_3)Y(0,\ell_1+\ell_2-\ell_3)Y(1,\ell_1+\ell_2+\ell_3)}.$$

Also see (E.40) for the explicit but lengthy expression. Here, $C_{(r_1,1),(r_2,1),(r_3,1)}$, $C_{(1,s_1),(1,s_2),(1,s_3)}$ and the square-root factor are positive at $c = 1$. The other factors are unambiguously defined.

Before finishing this subsection, we would like to mention the properties of the OPE coefficients under permutations of indices, given in [1, eq. (3.47)]:

$$C_{ijk} = (-1)^{P(\sigma)(n_i+n_j+n_k)}C_{\sigma(i)\sigma(j)\sigma(k)}, \qquad \sigma \in S_3, \qquad P(\sigma) := \begin{cases} 0, & \sigma \text{ even}, \\ 1, & \sigma \text{ odd}. \end{cases} \tag{196}$$

In the case of XXZ$_q$ CFT, the number $n_i$ in eq. (196) is given by

$$n_{r,s} = -2\ell^+\ell^-, \qquad (r = 2\ell^+ + 1,\ s = 2\ell^- + 1).$$ (197)

Eq. (196) comes from the braid locality condition, assuming that the braid matrix is the same as the $R$-matrix of the quantum group $U_q(sl_2)$. According to (195), to verify (196) for OPE coefficients in the XXZ$_q$ CFT, it suffices to check the following two statements:

- $C_{(r_1,1),(r_2,1),(r_3,1)}$ and $C_{(1,s_1),(1,s_2),(1,s_3)}$ are symmetric under permutations of indices;

- When $s_i$'s are all odd, the factor

$$\Phi(1,2,3) := i^{2\ell_1^+\ell_1^- + 2\ell_2^+\ell_2^- - 2\ell_3^+\ell_3^-}(-1)^{(\ell_1^+ + \ell_3^+ - \ell_2^+)(\ell_2^- + \ell_3^- - \ell_1^-) - 2\ell_1^+\ell_2^-},$$ (198)

  satisfies the same permutation relations as (196), i.e.

$$\Phi(1,2,3) = (-1)^{P(\sigma)(2\ell_1^+\ell_1^- + 2\ell_2^+\ell_2^- + 2\ell_3^+\ell_3^-)}\Phi(\sigma(1),\sigma(2),\sigma(3)).$$ (199)

The first statement is confirmed in section 3.6. The second statement can be verified by explicit computation.

## 4.3 Generalized minimal models

In the previous section we found a way of projecting to the operators of XXZ$_q$ CFT from the combined theory of chiral and anti-chiral Coulomb gas operators. In this section we discuss another projection allowed by the Coulomb gas formalism, that is, a projection to generalized diagonal minimal models with central charge (12). A similar prescription was suggested in the context of WZW models in [26].

To produce the correlation functions of the generalized minimal model, we propose the following construction:

$$\phi_{r,s}(z,\bar{z}) := e^{i\pi(\ell^+ + \ell^-)}\sum_{m^-=-\ell^-}^{\ell^-}\sum_{m^+=-\ell^+}^{\ell^+}\widetilde{V}_{(\ell^+,\ell^-),(m^+,m^-)}(z)\overline{\widetilde{V}}_{(\ell^+,\ell^-),(-m^+,-m^-)}(\bar{z}).$$ (200)

Here $\widetilde{V}$ and $\overline{\widetilde{V}}$ belong to decoupled chiral and anti-chiral theories. For the similar reason to (193), the OPE of $\phi_{r,s}$ is given by

$$
\begin{aligned}
\phi_{r_1,s_1}(z_1,\bar{z}_1)\phi_{r_2,s_2}(z_2,\bar{z}_2) &= \sum_{r_3,s_3}\frac{C_{(r_1,s_1)(r_2,s_2)(r_3,s_3)}C_{(r_2,s_2)(r_1,s_1)(r_3,s_3)}}{z_{21}^{h_{123}}\bar{z}_{21}^{\bar{h}_{123}}}\left[\phi_{r_3,s_3}(z_1,\bar{z}_1) + \dots\right] \\
&\equiv \sum_{r_3,s_3}\frac{C^{\mathrm{MM}}_{(r_1,s_1)(r_2,s_2)(r_3,s_3)}}{z_{21}^{h_{123}}\bar{z}_{21}^{\bar{h}_{123}}}\left[\phi_{r_3,s_3}(z_1,\bar{z}_1) + \dots\right].
\end{aligned}
$$ (201)

In the second line we used the second equality of (177).

Thus, we see that a choice of operators as in (200) reproduced the correct diagonal minimal model OPE coefficients. Moreover, the braiding relations for such operators trivialize under both quantum groups, and they become local operators. Similarly to the previous projection, despite having correct OPE and braiding, these operators are not singlets under quantum group action. Quantum group operators transform them into some non-local operators.

It has been known for a long time that crossing kernels of degenerate Virasoro blocks appearing in generalized minimal models are proportional to $6j$-symbols. This property is sometimes said to follow from a "hidden" quantum group symmetry [11]. The construction

above completely elucidates the origin of this property. Namely, it is a direct consequence of the existence of a crossing invariant set of correlators with 2 quantum group symmetries. Crossing symmetry of this theory imposes the behavior (184), and therefore the appearance of $U_q(sl_2)$ $6j$-symbols in the fusion kernel of Virasoro blocks in minimal models, given that the Virasoro blocks only depend on the weights of operators and the central charge.

Let us summarize the above discussion. In total, a generic two-dimensional CFT of the type we consider here has four QG symmetries – two that act on the chiral and two on anti-chiral sectors. In the terminology of [26], in diagonal minimal models all of these symmetries are "confined". This means that chiral and anti-chiral operators are paired in meson-like states so that topological lines corresponding to left and right QGs cancel each other and the operators are local. None of the QGs, or any of their combinations act within the Hilbert space of the minimal model, however, they still have an imprint on correlators, like the relation between the Virasoro blocks and the $6j$-symbols. This is why the symmetry is "hidden". It is still possible that some discrete set of transformation survives and corresponds to global symmetries of minimal models. In the XXZ$_q$ model only three quantum groups exist to start with due to the choice of operator weights. Two of them are confined, as in diagonal minimal models, and one survives as a global symmetry.

# 5 Conclusions

In this paper we continued to study quantum groups as global symmetries in QFTs. We focused on a particular example of a 2d CFT which has $U_q(sl_2)$ global symmetry – the continuum limit of XXZ$_q$ spin chain. We studied it using the Coulomb gas formalism. Importantly, to describe a theory with quantum group symmetry we introduced some modifications to the standard formalism. Unlike the formal approach to quantum group symmetries that was applied in [1], the Coulomb gas construction provides a more concrete framework.

First, for simplicity, we considered the construction applied only to the chiral subsector of XXZ$_q$ CFT, i.e. to the operators of conformal dimension $(h_{1,s}, 0)$. Following [10], we identified the screening charge $S_-$ with the lowering generator $F$ of $U_q(sl_2)$. We showed that, when acting on appropriate primaries of the free boson theory, this generates finite dimensional representations of operators transforming under the quantum group, see (36) and figure 4. Then, we defined the correlation functions of such operators as integrated correlators of vertex operators, see (38). The integration contour is defined in figure 5. This construction differs from the standard Coulomb gas approach by the introduction of a vertex operator, $\mathcal{V}_{2\alpha_0}(w)$, at the point $w$ where all of the integration contours start and end. We showed that such correlation functions do not depend on the position of $w$. Importantly, such construction of correlation functions makes them satisfy the defining property of having quantum group symmetry – quantum group Ward identities.

Then we proceeded to explicit computations of these correlation functions. We started with the derivation of the formula for two-point correlation functions of the lowest-weight – highest-weight configuration of operators of any spin $\ell$ in (94). Afterwards, we computed the general OPE coefficients (122). Finally, we canonically normalized the operators such that their two-point functions take the form of quantum Clebsch-Gordan coefficients. We found that the OPE coefficients of such normalized operators are given by the square root of minimal model OPE coefficients (140), multiplied by the Clebsch-Gordan coefficients, in agreement with the results of [1].

The Coulomb gas formalism can also be applied to correlation functions of general operators of the XXZ$_q$ CFT. To construct these, we first needed to extend the formalism to chiral operators of dimensions $(h_{r,s}, 0)$. In this case, the two screening charges $S_\pm$ are identified with

$F_\pm$, the lowering generators of two different quantum groups. By introducing appropriate cocycles, which represent the non-locality of chiral vertex operators, we constructed operators that transform under a simple tensor product of these quantum groups, $U_{q_+}(sl_2) \otimes U_{q_-}(sl_2)$. In particular, we showed that the correlation functions of such operators, defined by (146), independently satisfy Ward identities imposed by $U_{q_\pm}(sl_2)$. Then, repeating a similar procedure as for the single quantum group case, we computed general OPE coefficients: first in the normalization given to us from Coulomb gas formalism (164), and then in the canonical normalization (175).

The chiral operators of dimensions $(h_{r,s}, 0)$ transforming under two quantum groups were combined with an anti-chiral counter-part to get the general $XXZ_q$ CFT operators, see (191). In this construction, we projected operators of combined chiral and anti-chiral theories to a subsector which corresponds to the full set of operators of the $XXZ_q$ CFT. This pairing of chiral and anti-chiral sectors leads to non-local operators, so it is different from the usual one of [3,4]. This pairing leaves a single quantum group acting on the operators in a consistent way, and this group survives as a global symmetry of the model. Particularly, this construction provided us with the explicit calculation of OPE coefficients for general operators, (194). This result agrees with the result for OPE coefficients obtained from crossing symmetry in [1]. Finally, we realized another projection which results in generalized minimal model operators, (200). Such a projection correctly reproduces both the locality and the OPE structure of minimal model operators. This choice of contours is different from the one used in [3,4], however, it is equivalent, meaning that it leads to the same correlation functions. It would be interesting to see explicitly how this equivalence holds.

To conclude, we would like to mention that the developed version of the Coulomb gas formalism is a powerful tool for the description of quantum group symmetry in 2d CFTs. The Coulomb gas contours provide an intuitive picture of the non-locality of operators transformed by the quantum group. All of the CFT data can be explicitly computed by evaluating Coulomb gas integrals. We expect a similar construction to be possible starting from free field realizations of other theories: for example, the Wakimoto construction of WZW models. This would lead to new quantum-group invariant theories. For chiral WZW models, this would lead to a generalization of [29].

The Coulomb gas approach gives a more explicit construction of the correlators, allowing us to study various properties of the topological lines. Nonetheless, it is not a fully microscopic definition of the theory since it does not provide us with a description in terms of a Lagrangian or a deformation of a well defined theory. As a result, even at the practical level, there are important questions that are difficult to answer using the Coulomb gas. First, the lines attached to the $XXZ_q$ operators seem to factorize in a way that is neither additive, like an algebra line, nor multiplicative, like the topological defect lines (TDL) [16]. Nevertheless, they have a Hilbert space at their endpoints, like a TDL. It is interesting to understand how the two behaviors are compatible. Related to that, it is not obvious what the modular properties of the partition function of our theory are. Finally, it is fascinating to understand whether the non-local theory studied in this paper can be embedded as a defect sector in a local, and perhaps unitary, theory.

## Acknowledgments

We thank Matthias Gaberdiel, Sylvain Ribault, Slava Rychkov, Raoul Santachiara and Sasha Trufanov for valuable discussions. We are also grateful to the organizers and participants of the workshops "Bootstrap 2024" at Universidad Complutense in Madrid and "28$^e$ rencontre Itzykson: Analytic results in Conformal Field Theory" at Institut de Physique Théorique in Saclay for insightful exchanges.

**Funding information**   The work of BZ has been partially supported by STFC consolidated grants ST/T000694/1 and ST/X000664/1, as well as David Tong's Simons Investigator Award. The work of VG and JQ is supported by Simons Foundation grant 994310 (Simons Collaboration on Confinement and QCD Strings).

# A   Useful formulas to compute Coulomb gas integrals

In this appendix we collect all the useful formulas that we use throughout sections 3 and 4 to compute the Coulomb gas integrals. The following relations are helpful to simplify the expressions with $\Gamma$-functions:

$$
\begin{aligned}
\Gamma(z)\Gamma(1-z) &= \frac{\pi}{\sin(\pi z)}, \\
\Gamma(z)\Gamma\left(z+\frac{1}{2}\right) &= 2^{1-2z}\sqrt{\pi}\Gamma(2z).
\end{aligned}
\tag{A.1}
$$

Typically, the Coulomb gas integrals are reduced to the computation of Selberg-type integrals. The simplest such integral is an integral expression for Euler's Beta function,

$$
B(a,b) = \int_0^1 dt \cdot t^{a-1}(1-t)^{b-1} = \frac{\Gamma(a)\Gamma(b)}{\Gamma(a+b)}.
\tag{A.2}
$$

The generalization of the Beta function is known as a standard Selberg's integral:

$$
\begin{aligned}
S_n(\alpha,\beta,\gamma) &:= \int_0^1 dt_1 \cdots \int_0^1 dt_n \prod_{i=1}^n t_i^{\alpha-1}(1-t_i)^{\beta-1} \prod_{i<j} |t_i - t_j|^{2\gamma} \\
&= \prod_{j=0}^{n-1} \frac{\Gamma(\alpha+j\gamma)\Gamma(\beta+j\gamma)\Gamma(1+(j+1)\gamma)}{\Gamma(\alpha+\beta+(n+j-1)\gamma)\Gamma(1+\gamma)}.
\end{aligned}
\tag{A.3}
$$

In particular, for $n = 1$ it is reduced to Beta function (A.2).

There is also a useful generalization of the Selberg integral [4, appendix A],

$$
\begin{aligned}
J_{n,m}(\alpha,\beta;\gamma) &:= \prod_{i=1}^n \int_0^1 dt_i\, t_i^{\alpha-1}(1-t_i)^{\beta-1} \prod_{i=1}^m \int_0^1 d\tau_i\, \tau_i^{-(\alpha-1)/\gamma}(1-\tau_i)^{-(\beta-1)/\gamma} \\
&\quad \times \prod_{1 \leq i < j \leq n} |t_i - t_j|^{2\gamma} \prod_{1 \leq i < j \leq m} |\tau_i - \tau_j|^{2/\gamma} \prod_{i=1}^n \prod_{j=1}^m \frac{P}{(t_i - \tau_j)^2},
\end{aligned}
\tag{A.4}
$$

where $P$ stands for principal value integration. It integrates to

$$
\begin{aligned}
J_{n,m}(\alpha,\beta;\gamma) &= m!\, \gamma^{-(2n+1)m} \prod_{i=1}^n \left( \frac{\Gamma(\alpha+(i-1)\gamma)\Gamma(\beta+(i-1)\gamma)\Gamma(1+i\gamma)}{\Gamma(-2m+\alpha+\beta+(n-2+i)\gamma)\Gamma(1+\gamma)} \right) \\
&\quad \times \prod_{j=1}^m \left( \frac{\Gamma(-n+j\gamma^{-1})\Gamma(1-n+(j-\alpha)\gamma^{-1})\Gamma(1-n+(j-\beta)\gamma^{-1})}{\Gamma(1+\gamma^{-1})\Gamma(2-n+(m+j-\alpha-\beta)\gamma^{-1})} \right).
\end{aligned}
\tag{A.5}
$$

We see that (A.5) reduces to (A.3) when $m = 0$.

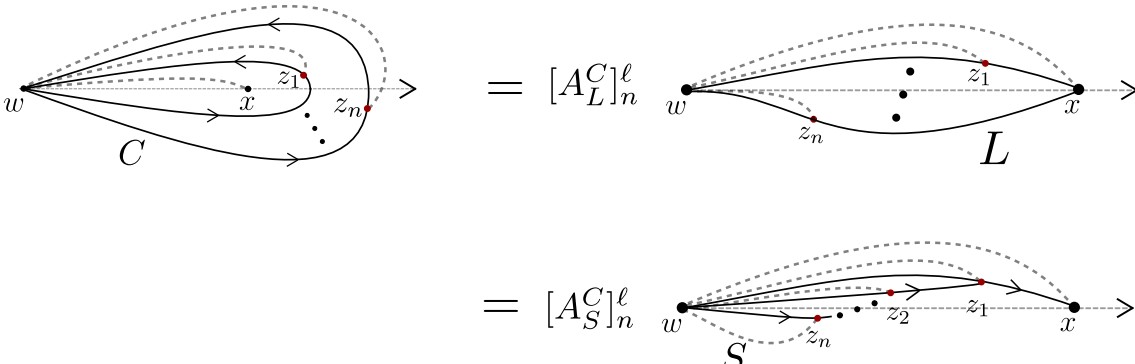

Figure 20: As a reminder, the integrand is completely specified by ordering of the vertex operators (26). The interpretation of the ordering for the first figure (defining representation) is that the integrand is real and positive when all the vertex operators are located on the real line with the following order: $w < x < z_1 < \ldots < z_n$.

## B  Different representations of Coulomb gas integrals

In this appendix we compute the constants associated with conversions of integrals between different representations. In this paper we utilize all 3 representations appearing in figure 20. The dots in the figures represent other screening charges. The goal of this appendix is to compute the constants $A_L^C$ and $A_S^C$.

Let us start with the computation of the coefficient $[A_L^C]_n^\ell$, that is, the conversion of the integral on the LHS of figure 20 to the top RHS of figure 20. More precisely, we perform the following transformation of the relevant piece of the integral:

$$\int_C dz_1 \ldots dz_n \mathcal{V}_{-\ell\alpha_-}(x) \mathcal{V}_{\alpha_-}(z_1) \ldots \mathcal{V}_{\alpha_-}(z_n) = [A_L^C]_n^\ell \int_{\gamma_1} dz_1 \ldots \int_{\gamma_n} dz_n \mathcal{V}_{\alpha_-}(z_n) \ldots \mathcal{V}_{\alpha_-}(z_1) \mathcal{V}_{-\ell\alpha_-}(x),$$
(B.1)

where $\gamma_k$'s are paths from $w$ to $x$ parallel to the real line. These paths do not intersect except at $w$ and $x$. The full contour appearing on the RHS including all $\gamma_k$ is called $L$. Here each $z_i$ goes from $w$ to $x$, but the integrand is not symmetric under permutations of the $z_i$'s. Namely, it acquires phases depending on the relative ordering of $z_i$'s.

We first consider the simplest case with a single integral ($n = 1$), with the operator $\mathcal{V}_{\alpha_-}(z_1)$ being integrated. Figure 21 represents the contour deformations. In the second line of the figure we split the integral into two pieces: the one from $w$ to $x$ and the second one from $x$ to $w$. Now, figure 3 allows to exchange the dashed lines and yields the phases $e^{-i\pi\cdot(-2\ell\alpha_-^2)}$ and $e^{i\pi\cdot(-2\ell\alpha_-^2)}$ respectively for the first and second integrals. Using the notation $q = e^{i\pi\alpha_-^2}$, the overall coefficient in the last line becomes $(q^{2\ell} - q^{-2\ell})$. What we have just computed is the coefficient $[A_L^C]_1^\ell$.

Now, we compute the coefficient $[A_L^C]_n^\ell$ for arbitrary $n$. Assume that $(n-1)$ contours are already deformed, and now we deform the last contour, see figure 22. The first step is the same, we split the contour into two pieces. Then, again we use figure 3 to exchange the dashed lines. In the second line of figure 22 we see a difference between the first and the second integrals in the brackets. For the first integral the dashed line that goes to $z_n$ has to be exchanged only with the line going to $x$ (after that, one can relabel the integration variables $z_i$ to be from $z_1$ to $z_n$ from the top line to the bottom line respectively). Whereas for the second integral the dashed line that goes to $z_n$ has to be exchanged with all other dashed lines. Using the same notation for $q$, we get the following relation:

$$[A_L^C]_n^\ell = \left(q^{2\ell-2n+2} - q^{-2\ell}\right)[A_L^C]_{n-1}^\ell.$$
(B.2)

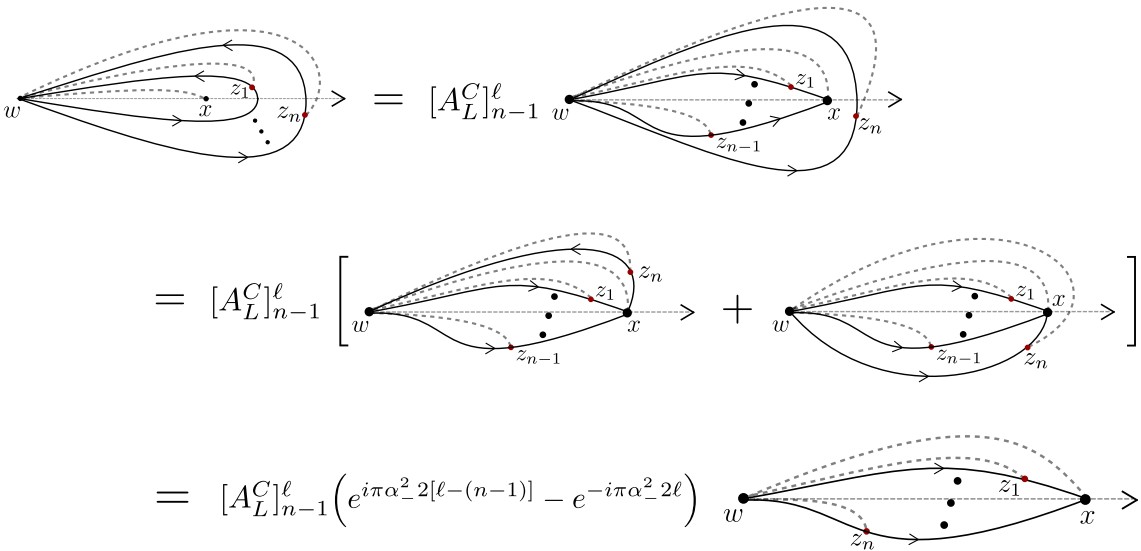

Figure 21: This figure represents the necessary contour deformations to compute the coefficient $[A_L^C]_1^\ell$.

Figure 22: This figure represents the necessary contour deformations to compute the coefficient $[A_L^C]_n^\ell$ assuming that $(n-1)$ contours were already deformed.

From these contour manipulations we conclude that,

$$[A_L^C]_n^\ell = \prod_{j=0}^{n-1}\left(q^{2\ell-2j} - q^{-2\ell}\right). \tag{B.3}$$

Now, we move on to the computation of the coefficient $[A_S^C]_n^\ell/[A_L^C]_n^\ell$. In other words, we start from the top RHS of figure 20 and work our way to the bottom RHS. The latter is defined as the following transformation of the integral

$$\int_C dz_1 \ldots dz_n \mathcal{V}_{-\ell\alpha_-}(x)\mathcal{V}_{\alpha_-}(z_1)\ldots \mathcal{V}_{\alpha_-}(z_n) = [A_S^C]_n^\ell \int_w^x dz_1 \int_w^{z_1} dz_2 \ldots$$
$$\times \int_w^{z_{n-1}} dz_n \mathcal{V}_{\alpha_-}(z_n)\ldots \mathcal{V}_{\alpha_-}(z_1)\mathcal{V}_{-\ell\alpha_-}(x). \tag{B.4}$$

Here $z_1$ is integrated along the real line from $w$ to $x$, $z_2$ from $w$ to $z_1$ and so on. The integrand is defined to be real for

$$w < z_n < \ldots < z_1 < x. \tag{B.5}$$

We denote the contour corresponding to this collection of integrals $S$.

To compute this quantity, we perform a similar trick to the one we used before. We assume that $(n-1)$ contours are already deformed as needed, and we deform the $n$-th contour. This

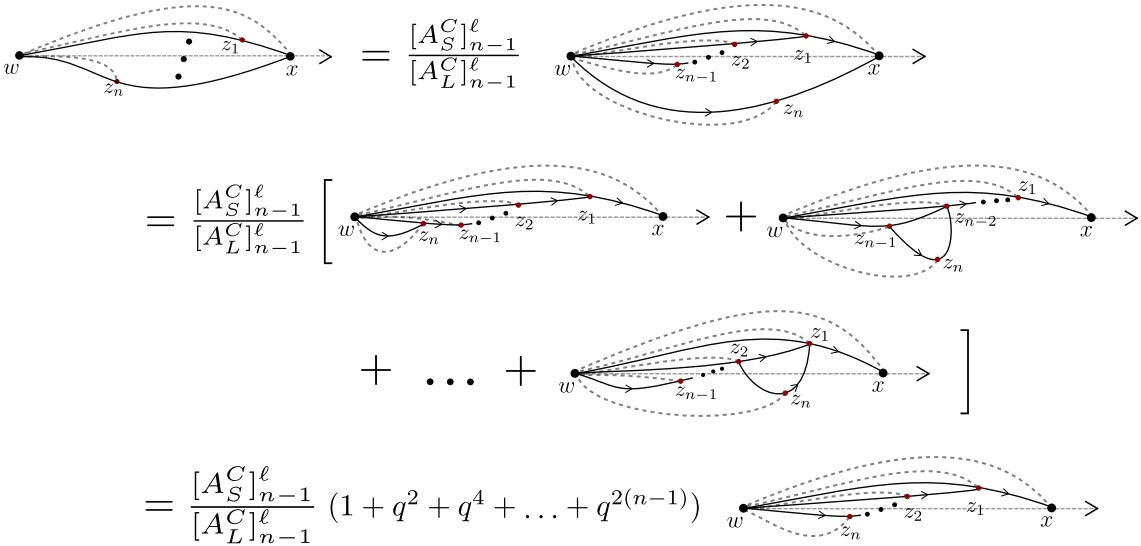

Figure 23: This figure represents the necessary contour deformations to compute the coefficient $[A_S^C]_n^\ell / [A_L^C]_n^\ell$ assuming that $(n-1)$ contours are already deformed.

is represented in figure 23. Clearly, when the integration over $z_n$ is between $w$ and $z_{n-1}$, the integrand is real and no phase is needed: this is given by the first term in the second line. Each next term needs one more exchange of operators $\mathcal{V}_{\alpha_-}$, which gives phases $q^2, q^4, \ldots$. Then the recursion provides us with the following relation

$$\frac{[A_S^C]_n^\ell}{[A_L^C]_n^\ell} = \left(1 + q^2 + q^4 + \ldots + q^{2(n-1)}\right) \frac{[A_S^C]_{n-1}^\ell}{[A_L^C]_{n-1}^\ell} = \frac{1-q^{2n}}{1-q^2} \frac{[A_S^C]_{n-1}^\ell}{[A_L^C]_{n-1}^\ell}, \tag{B.6}$$

from which we can write the final formula

$$\frac{[A_S^C]_n^\ell}{[A_L^C]_n^\ell} = \Sigma_n = \prod_{k=1}^n \frac{1-q^{2k}}{1-q^2}. \tag{B.7}$$

Note that the same ratio was computed in [4].

Let us now discuss the possible divergences of the integrals in all representations. The original integral appearing in the definition on the LHS of figure 20 is finite in a subset of the physical region, $\mathrm{Re}(\alpha_-^2) > \frac{1}{2}$, with the only potential divergence coming from the vicinity of $w$. Conversely, for generic spin $\ell$, the second and third representations in figure 20 are formal integrals that diverge close to $x$. In principle, in the modification of the contour when shrinking the contours around point $x$, one has to keep the integration along a small circle around this point. It is divergent in the small circle limit and cancels the divergence of line integrals. However, this contribution is strictly divergent – it has no finite piece. This means that we can assign an unambiguous value to the line integrals. The precise statement is that any regularization of the integral, such as point-splitting, will introduce a cutoff scale $\delta$. Then, in order to cancel the divergences we add contributions where we remove $n$ of the integrals and replace the operator $\mathcal{V}_{-\ell\alpha_-}(x)$ with a linear combination of all operators with charge $(n-\ell)\alpha_-$ and an appropriate power of $\delta$ to compensate for the difference in dimensions. The coefficients of operators coming with negative powers of delta are tuned to cancel divergences, and then $\delta$ is taken to 0. Since the computation should be valid in any reasonable scheme, there can only be ambiguities if there are operators appearing in any of the sums accompanied by $\delta^0$. This is allowed only for operators of a very specific charge to dimension relation (Specifically, operator of dimensions $h_{1,2\ell+1}$ that have charge $-\ell\alpha_- + k\alpha_-$ for $k > 0$) that do not exist in our formalism.

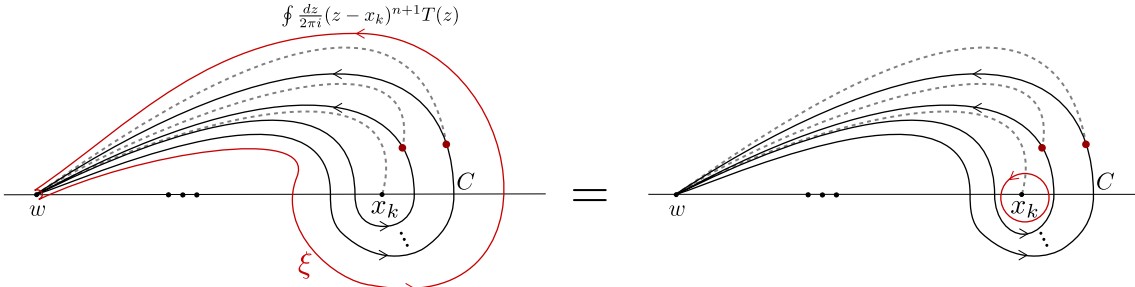

Figure 24: The contour in this figure gives a consistent definition of the Virasoro generators, such that they commute with the quantum group generators. It is important that the contour passes infinitesimally close to the point $w$.

## C  Commutation relations with Virasoro

In this appendix, we define the Virasoro generators acting on operators $V_{\ell,m}$. The integrand of Virasoro generator takes the usual form, but the integration contour $\xi$,

$$L_n = \oint_\xi \frac{dz}{2\pi i}(z - x_k)^{n+1} T(z), \tag{C.1}$$

has to be taken to pass infinitesimally close to the point $w$ (see the LHS of figure 24).

We will now show that with this definition the commutator of the generators of Virasoro with the generator $F$ vanishes. Showing this is equivalent to showing that one can move the red line, past the black lines, to act solely on $\mathcal{V}_{\alpha_{1,2\ell+1}}(x_k)$, as in the RHS of figure 24. To do this, let us notice that the full expression represented by the LHS of figure 24 has two types of contours: a red one, $\xi$, corresponding to the action of a Virasoro generator and a black one, $C$, representing the integration of the screening charges. First, let us split the integration over the contour $C$ into two parts: one is an infinitesimal piece from the point $w$ to the red line, and the second one is fully inside the red contour. The first part vanishes by analytic continuation from $\alpha_-^2 > 1/2$. For the second part, we continue by shrinking the red contour, separately for each value of $z_i$ (under the sign of the integral corresponding to the black contours). The result is a sum of small circular contours around $x_k$ and around each of the $z_i$'s. By subtracting the contour around $x_k$ we get the commutator $[L_n, F^{\ell_k-m_k}]$. We then conclude that the commutator is equal to the sum of small contours around the $z_i$'s, taking the form,

$$\sum_i \oint \frac{dz}{2\pi i}\left((z_i - x_k)^{n+1} + (1+n)(z_i - x_k)^n(z - z_i) + \dots\right) T(z). \tag{C.2}$$

This operator should be inserted in the full correlator and under the integral sign of the black contour $C$. The first term generates a translation, and the second a dilatation. The ellipses stand for terms with higher powers of $(z - z_i)$, which vanish when acting on a primary. For each term in the sum, the two non-vanishing terms add up to,

$$(z_i - x_k)^{n+1}\partial_{z_i}\mathcal{V}_{\alpha_-}(z_i) + (1+n)(z_i - x_k)^n(z - z_i)\mathcal{V}_{\alpha_-}(z_i) = \partial_{z_i}\left[(z_i - x_k)^{n+1}\mathcal{V}_{\alpha_-}(z_i)\right]. \tag{C.3}$$

We see that the commutator of $L_n$ and the $i$-th $F$ line is the replacement of $\mathcal{V}_{\alpha_-}(z_i)$ with the total derivative above. Then, we can preform the $F$-line integral immediately – it localizes to its boundary (in figure 24 it is where it meets the red contour). By the OPE between $\mathcal{V}_{\alpha_-}$ at that point and $\mathcal{V}_{2\alpha_0}(w)$, this boundary term is a fractional power of the radius of the small red semi-circle around $w$ in figure 24. For generic values of $\alpha_-$, it vanishes by analytic continuation from $\alpha_-^2 > 1$. We conclude that the generators of Virasoro commute with the quantum group

generators. Note that if the contour did not pass infinitesimally close to the point $w$, the boundary term of the total derivative would not vanish and the commutator with $F$ would be nontrivial.

# D   Proof of Lemma 3.1

Dividing both sides of (43) by $f$, the problem is reduced to proving

$$\frac{\partial}{\partial w} \log f = \sum_{r=1}^{N} g_r \frac{\partial}{\partial z_r} \log(g_r f). \tag{D.1}$$

By explicit computation, one can see that both sides of the above equation only depend on $\alpha_-$ through a factor of $2(\alpha_-^2 - 1)$, and above equation is equivalent to the following identity which is independent of $\alpha_-$:

$$\sum_{r=1}^{N} \frac{1}{w - z_r} - \sum_{i=1}^{n} \frac{\ell_i}{w - x_i} = \sum_{r=1}^{N} g_r \left( -\frac{1}{w - z_r} + \sum_{\substack{s=1 \\ s \neq r}}^{N} \frac{1}{z_r - z_s} - \sum_{i=1}^{n} \frac{\ell_i}{z_r - x_i} \right). \tag{D.2}$$

Notice that in (D.2), there are no singularities when $x_i - x_j = 0$, so nothing bad happens when we take $x_i = x_j$. Therefore, it suffices to prove the case when $n = 2N$ and $\ell_1 = \ell_2 = \ldots = \ell_n = \frac{1}{2}$, all other cases can be reproduced by setting some of the $x_i$'s equal. In other words, the problem is reduced to proving the following identity

$$\sum_{r=1}^{N} \frac{1}{w - z_r} - \sum_{i=1}^{2N} \frac{1/2}{w - x_i} = \sum_{r=1}^{N} g_r^{(N)} \left( -\frac{1}{w - z_r} + \sum_{\substack{s=1 \\ s \neq r}}^{N} \frac{1}{z_r - z_s} - \sum_{i=1}^{2N} \frac{1/2}{z_r - x_i} \right), \tag{D.3}$$

where

$$g_r^{(N)}(w, x_1, \ldots, x_{2N}, z_1, \ldots, z_N) = -\left( \prod_{i=1}^{2N} \left( \frac{z_r - x_i}{w - x_i} \right) \right) \left( \prod_{\substack{s=1 \\ s \neq r}}^{N} \left( \frac{w - z_s}{z_r - z_s} \right)^2 \right). \tag{D.4}$$

Let us define the RHS of (D.3) as $G_N(w, x_1, \ldots, x_{2N}, z_1, \ldots, z_N)$. For any fixed variables $(x_1, \ldots, x_{2N}, z_1, \ldots, z_N)$, $G_N$ has the following properties as a function of $w$:

- $G_N(w)$ is a meromorphic function of $w$ on $\mathbb{C}$.

- $G_N(w)$ has poles only at $w = z_r$ and $w = x_i$.

- $G_N(w)$ is bounded in the regime where $|w| \geqslant \max\{|z_1|, \ldots, |z_N|, |x_1|, \ldots, |x_{2N}|\} + 1$.

- $G_N(\infty) = 0$.

Therefore, to show (D.3), it suffices to show that $G_N(w)$ (as a function of $w$) has simple poles at $w = z_r$ and $w = x_i$, with the same residues as the LHS of (D.3).

Let us first look at the pole at $w = z_r$. The functions $g_1, g_2, \ldots, g_N$ (as functions of $w$) are regular at $w = z_r$. Furthermore, we have $g_r(w = z_r) = -1$. Therefore, the singularity of $G(w)$ at $w = z_r$ comes from the $-\frac{1}{w - z_r}$ term in the RHS of (D.3), and we have

$$G_N(w) \overset{w \to z_r}{=} \frac{1}{w - z_r} + O(1), \tag{D.5}$$

which matches the LHS of (D.3).

Then let us look at the pole at $w = x_{2N}$. To match the LHS of (D.3), we need to show that $w = x_{2N}$ is a simple pole of $G_N(w)$, and with the residue equal to $-\frac{1}{2}$:

$$
\begin{aligned}
F_N &\equiv \operatorname{Res} G_N(w)\big|_{w=x_{2N}} \\
&= \sum_{r=1}^{N-1} \frac{(z_r - x_{2N-1})(z_r - x_{2N})(x_{2N} - z_N)^2}{(x_{2N} - x_{2N-1})(z_r - z_N)^2} g_r^{(N-1)}(x_{2N}, x_1, \dots, x_{2N-2}, z_1, \dots, z_{N-1}) \\
&\quad \times \left( -\frac{1}{2(x_{2N} - z_r)} - \sum_{i=1}^{2N-1} \frac{1}{2(z_r - x_i)} + \sum_{\substack{s=1 \\ s \neq r}}^{N} \frac{1}{z_r - z_s} \right) \\
&\quad + (z_N - x_{2N}) \prod_{i=1}^{2N-1} \left( \frac{z_N - x_i}{x_{2N} - x_i} \right) \prod_{s=1}^{N-1} \left( \frac{x_{2N} - z_s}{z_N - z_s} \right)^2 \\
&\quad \times \left( \frac{1}{2(x_{2N} - z_N)} + \sum_{i=1}^{2N-1} \frac{1}{2(z_N - x_i)} - \sum_{s=1}^{N} \frac{1}{z_N - z_s} \right) \\
&\stackrel{?}{=} -\frac{1}{2}.
\end{aligned}
\tag{D.6}
$$

Let us view $F_N$ as a function of $x_{2N}$. When the other variables are fixed, $F_N(x_{2N})$ is a meromorphic function of $x_{2N}$, bounded when $|x_{2N}|$ is sufficiently large, and has potential poles at $x_{2N} = x_i$ ($i = 1, 2, \dots, 2N-1$). To prove (D.6), we need to show that

- $F_N(x_{2N} = \infty) = -\frac{1}{2}$.

- $F_N(x_{2N})$ is actually regular at $x_{2N} = x_i$ ($i = 1, 2, \dots, 2N-1$).

**Proof of $F_N(x_{2N} = \infty) = -\frac{1}{2}$**

By (D.6), $F_N(x_{2N} = \infty)$ is given by the following expression

$$
\begin{aligned}
H_N &\equiv F_N(x_{2N} = \infty) \\
&= -\sum_{r=1}^{N} \left( \prod_{i=1}^{2N-1} (z_r - x_i) \right) \left( \prod_{\substack{s=1 \\ s \neq r}}^{N} \frac{1}{(z_r - z_s)^2} \right) \left( \sum_{i=1}^{2N-1} \frac{1}{2(z_r - x_i)} - \sum_{\substack{s=1 \\ s \neq r}}^{N} \frac{1}{z_r - z_s} \right).
\end{aligned}
\tag{D.7}
$$

We would like to show that $H_N \equiv -\frac{1}{2}$. Let us view $H_N$ as a function of $z_N$ (i.e. fix other variables). $H_N(z_N)$ is a meromorphic function of $z_N$ with potential triple poles at $z_N = z_r$ ($r = 1, \dots, N-1$), and bounded around $z_N = \infty$. So it suffices to show that (a) $H_N(z_N = \infty) = -\frac{1}{2}$ and (b) $H_N(z_N)$ is regular at $z_N = z_r$.

To see (a), we take the limit $z_N \to \infty$ for $H_N$. The only non-vanishing contributions come from the $r = N$ term in (D.7). This gives

$$
\lim_{z_N \to \infty} H_N(z_N) = -\left( \frac{2N-1}{2} - (N-1) \right) = -\frac{1}{2}.
\tag{D.8}
$$

Then let us prove (b). Since $H_N$ is permutation symmetric in $z_1, \dots, z_N$, it suffices to prove the regularity of $H_N(z_N)$ at $z_N = z_{N-1}$. Around $z_N = z_{N-1}$ we have

$$
H_N(z_N) = \frac{h_{-3}}{(z_N - z_{N-1})^3} + \frac{h_{-2}}{(z_N - z_{N-1})^2} + \frac{h_{-3}}{z_N - z_{N-1}} + O(1),
\tag{D.9}
$$

where $h_{-3}, h_{-2}$ and $h_{-1}$ are functions of $x_1, \dots, x_{2N-1}$ and $z_1, \dots, z_{N-1}$. We would like to show that they are actually zero. Let us do explicit computation.

The contributions to $h_{-3}$ come from $r = N-1$ and $r = N$ in (D.7):

$$h_{-3} = -\left(\prod_{i=1}^{2N-1}(z_{N-1} - x_i)\right)\left(\prod_{s=1}^{N-2}\frac{1}{(z_{N-1} - z_s)^2}\right) + \left(\prod_{i=1}^{2N-1}(z_N - x_i)\right)\left(\prod_{s=1}^{N-2}\frac{1}{(z_N - z_s)^2}\right)$$
$$\overset{z_N = z_{N-1}}{=} 0. \tag{D.10}$$

The contributions to $h_{-2}$ also come from $r = N-1$ and $r = N$ in (D.7):

$$h_{-2} = -\left(\prod_{i=1}^{2N-1}(z_{N-1} - x_i)\right)\left(\prod_{s=1}^{N-2}\frac{1}{(z_{N-1} - z_s)^2}\right)\left(\sum_{i=1}^{2N-1}\frac{1}{2(z_{N-1} - x_i)} - \sum_{s=1}^{N-2}\frac{1}{z_{N-1} - z_s}\right)$$
$$-\left(\prod_{i=1}^{2N-1}(z_N - x_i)\right)\left(\prod_{s=1}^{N-2}\frac{1}{(z_N - z_s)^2}\right)\left(\sum_{i=1}^{2N-1}\frac{1}{2(z_N - x_i)} - \sum_{s=1}^{N-2}\frac{1}{z_N - z_s}\right) \tag{D.11}$$
$$+ \frac{\partial}{\partial z_N}\left[\left(\prod_{i=1}^{2N-1}(z_N - x_i)\right)\left(\prod_{s=1}^{N-2}\frac{1}{(z_N - z_s)^2}\right)\right]$$
$$\overset{z_N = z_{N-1}}{=} 0.$$

Here the first line comes from $r = N-1$, and the second and the third lines come from $r = N$ in (D.7).

The contributions to $h_{-1}$ only come from $r = N$ in (D.7):

$$h_{-1} = -\frac{\partial}{\partial z_N}\left[\left(\prod_{i=1}^{2N-1}(z_N - x_i)\right)\left(\prod_{s=1}^{N-2}\frac{1}{(z_N - z_s)^2}\right)\left(\sum_{i=1}^{2N-1}\frac{1}{2(z_N - x_i)} - \sum_{s=1}^{N-2}\frac{1}{z_N - z_s}\right)\right]$$
$$+ \frac{1}{2}\frac{\partial^2}{\partial z_N^2}\left[\left(\prod_{i=1}^{2N-1}(z_N - x_i)\right)\left(\prod_{s=1}^{N-2}\frac{1}{(z_N - z_s)^2}\right)\right] \tag{D.12}$$
$$= 0.$$

This finishes the proof of $H_N(z_N) \equiv -\frac{1}{2}$.

**Proof of regularity of $F_N(x_{2N})$ at $x_{2N} = x_i$**

By (D.6), $F_N$ is permutation symmetric in $x_1, \ldots, x_{2N-1}$, so it suffices to prove the regularity of $F_N(x_{2N})$ at $x_{2N} = x_{2N-1}$. The residue of $F_N(x_{2N})$ at $x_{2N} = x_{2N-1}$ is given by

$$\text{Res}\, F_N(x_{2N})\big|_{x_{2N} = x_{2N-1}} = \sum_{r=1}^{N-1}\frac{(z_r - x_{2N-1})^2(x_{2N-1} - z_N)^2}{(z_r - z_N)^2}$$
$$\times g_r^{(N-1)}(x_{2N-1}, x_1, \ldots, x_{2N-2}, z_1, \ldots, z_{N-1})$$
$$\times\left(-\sum_{i=1}^{2N-2}\frac{1}{2(z_r - x_i)} + \sum_{\substack{s=1 \\ s \neq r}}^{N}\frac{1}{z_r - z_s}\right) \tag{D.13}$$
$$+ (z_N - x_{2N-1})^2\prod_{i=1}^{2N-2}\left(\frac{z_N - x_i}{x_{2N-1} - x_i}\right)\prod_{s=1}^{N-1}\left(\frac{x_{2N-1} - z_s}{z_N - z_s}\right)^2$$
$$\times\left(\sum_{i=1}^{2N-2}\frac{1}{2(z_N - x_i)} - \sum_{s=1}^{N}\frac{1}{z_N - z_s}\right).$$

We observe that the residue function has an overall factor of

$$\prod_{r=1}^{N}(x_{2N-1} - z_r)^2 \times \prod_{i=1}^{2N-2}\frac{1}{x_{2N-1} - x_i},$$

and the proof of $\operatorname{Res} F_N(x_{2N})\big|_{x_{2N}=x_{2N-1}} \equiv 0$ is reduced to proving the following identity:

$$R_N \equiv \sum_{r=1}^{N}\left(\prod_{i=1}^{2N-2}(z_r - x_i)\right)\left(\prod_{\substack{s=1\\s\neq r}}^{N}\frac{1}{(z_r - z_s)^2}\right)\left[\sum_{i=1}^{2N-2}\frac{1}{z_r - x_i} - \sum_{\substack{s=1\\s\neq r}}^{N}\frac{2}{z_r - z_s}\right] \overset{?}{=} 0. \quad \text{(D.14)}$$

Comparing $R_N$ with the definition of $H_N$ in (D.7), we have

$$R_N = \lim_{x_{2N-1}\to\infty}\frac{\partial}{\partial x_{2N-1}}H_N(x_{2N-1}). \quad \text{(D.15)}$$

Since we have proven that $H_N(x_{2N-1}) \equiv -\frac{1}{2}$, the above relation implies that $R_N \equiv 0$. This finishes the proof of regularity of $F_N(x_{2N})$ at $x_{2N} = x_{2N-1}$ and the proof of lemma 3.1.

# E  Chiral OPE coefficients in the canonical normalization

This appendix provides the technical details on the calculations related to the general chiral OPE coefficients:

$$C_{(r_1,s_1),(r_2,s_2),(r_3,s_3)}.$$

Here, we normalized the operators such that

$$C_{(r,s),(r,s),(1,1)} = 1, \quad \text{(E.1)}$$

holds for all positive integers $r$ and $s$. This is what we mean by "canonical normalization".

At the same central charge, these OPE coefficients are related to the ones in the $\text{XXZ}_q$ CFT (denoted as $C^{\text{XXZ}}_{(r_1,s_1),(r_2,s_2),(r_3,s_3)}$) and the ones in the diagonal generalized minimal model CFT (denoted as $C^{\text{MM}}_{(r_1,s_1),(r_2,s_2),(r_3,s_3)}$) as follows:

$$\begin{aligned}
C^{\text{XXZ}}_{(r_1,s_1),(r_2,s_2),(r_3,s_3)} &= C_{(r_1,s_1),(r_2,s_2),(r_3,s_3)}C_{(r_1,1),(r_2,1),(r_3,1)} & (r_i \in \mathbb{N}+1,\ s_i \in 2\mathbb{N}+1),\\
C^{\text{MM}}_{(r_1,s_1),(r_2,s_2),(r_3,s_3)} &= C_{(r_1,s_1),(r_2,s_2),(r_3,s_3)}C_{(r_2,s_2),(r_1,s_1),(r_3,s_3)} & (r_i,s_i \in \mathbb{N}+1).
\end{aligned} \quad \text{(E.2)}$$

In particular, the OPE coefficients in the chiral subsector of $\text{XXZ}_q$ CFT are simply given by

$$C^{\text{XXZ}}_{(1,s_1),(1,s_2),(1,s_3)} = C_{(1,s_1),(1,s_2),(1,s_3)} \qquad (s_i \in 2\mathbb{N}+1). \quad \text{(E.3)}$$

The appendix is organized as follows. In section E.1, we compute the chiral OPE coefficients in the $(1,s)$ subsector, $C_{(1,s_1),(1,s_2),(1,s_3)}$. In section E.2, we check the relation (E.2) for $C_{(1,s_1),(1,s_2),(1,s_3)}$ (which is (142) in the main text). The results for the chiral OPE coefficients in the $(r,1)$ subsector is similar, so we give the result in the main text instead of discussing it here. In section E.3, we compute the general chiral OPE coefficients, $C_{(r_1,s_1),(r_2,s_2),(r_3,s_3)}$ and check the relation (E.2). In section E.4 we give a full expression of OPE coefficients in $\text{XXZ}_q$ CFT.

## E.1  Computing $C_{(1,2\ell_1+1),(1,2\ell_2+1),(1,2\ell_3+1)}$

We choose an arbitrary triplet of $U_q(sl_2)$ spins $(\ell_1,\ell_2,\ell_3)$. $\ell_i$'s can be integers or half-integers, and they should satisfy the following conditions required by the representation theory of the quantum group $U_q(sl_2)$:

- (triangle inequality) $|\ell_1 - \ell_2| \leqslant \ell_3 \leqslant \ell_1 + \ell_2$;

- $\ell_1 + \ell_2 + \ell_3 \in \mathbb{N}$.

Our starting point is (136), with a specific choice of $m_1$ and $m_2$:

$$m_1 = \ell_3 - \ell_2, \qquad m_2 = \ell_2. \tag{E.4}$$

Under this choice, (136) simplifies to:

$$C_{(1,2\ell_1+1),(1,2\ell_2+1),(1,2\ell_3+1)} = \frac{\mathcal{N}_{\ell_3}\, \widetilde{C}^{(\ell_3,\ell_3)}_{(\ell_1,\ell_3-\ell_2),(\ell_2,\ell_2)}}{\mathcal{N}_{\ell_1}\mathcal{N}_{\ell_2}\left(\displaystyle\prod_{m=\ell_3-\ell_2+1}^{\ell_1} f_m^{\ell_1}\right)\begin{bmatrix}\ell_1 & \ell_2 & \ell_3 \\ \ell_3-\ell_2 & \ell_2 & \ell_3\end{bmatrix}_q}. \tag{E.5}$$

The explicit expressions for $\mathcal{N}_\ell$, $\widetilde{C}^{(\ell_3,\ell_3)}_{(\ell_1,\ell_3-\ell_2),(\ell_2,\ell_2)}$, and $f_m^\ell$ are given in eqs. (128), (122), and (4), respectively.

The final ingredient is the Clebsch-Gordan coefficient. The general expression for the Clebsch-Gordan coefficient is provided in [30] (see also [1, section 2.2.2]). In this special case, it simplifies to:

$$\begin{bmatrix}\ell_1 & \ell_2 & \ell_3 \\ \ell_3-\ell_2 & \ell_2 & \ell_3\end{bmatrix}_q = (-1)^{\ell_1+\ell_2-\ell_3} q^{-\frac{1}{2}(\ell_1+\ell_2-\ell_3)(\ell_1-\ell_2+\ell_3+1)}\sqrt{\frac{[2\ell_2]![2\ell_3+1]!}{[\ell_2+\ell_3-\ell_1]![\ell_1+\ell_2+\ell_3+1]!}}, \tag{E.6}$$

where $[n]!$ denotes the $q$-deformed factorial.

Substituting eqs. (4), (128), (122), and (E.6) into eq. (E.5), we arrive at the following expression for the OPE coefficient $C_{(1,2\ell_1+1),(1,2\ell_2+1),(1,2\ell_3+1)}$:

$$\begin{aligned}
C_{(1,2\ell_1+1),(1,2\ell_2+1),(1,2\ell_3+1)} &= \frac{1}{(\ell_1+\ell_2-\ell_3)!\,[2\ell_3]!}\sqrt{\frac{(2\ell_1)!\,(2\ell_2)!}{(2\ell_3)!}} \\
&\quad \times \frac{1}{\left[[2\ell_1+1]_q\,[2\ell_2+1]_q\,[2\ell_3+1]_q\right]^{1/4}} \\
&\quad \times \sqrt{\frac{[\ell_1-\ell_2+\ell_3]!\,[\ell_2-\ell_1+\ell_3]!\,[\ell_1+\ell_2+\ell_3+1]!}{[\ell_1+\ell_2-\ell_3]!}} \\
&\quad \times \prod_{k=1}^{\ell_1+\ell_2-\ell_3}\frac{\Gamma\big((\ell_1+\ell_2+\ell_3-k+2)\alpha_-^2-1\big)}{\Gamma\big((2\ell_1-k+1)\alpha_-^2\big)\Gamma\big((2\ell_2-k+1)\alpha_-^2\big)\Gamma(-k\alpha_-^2)} \\
&\quad \times \sqrt{\left(\prod_{k=1}^{2\ell_1}\frac{\Gamma(k\alpha_-^2)^2\Gamma(-k\alpha_-^2)}{\Gamma((k+1)\alpha_-^2-1)}\right)\left(\prod_{k=1}^{2\ell_2}\frac{\Gamma(k\alpha_-^2)^2\Gamma(-k\alpha_-^2)}{\Gamma((k+1)\alpha_-^2-1)}\right)} \\
&\quad \times \sqrt{\prod_{k=1}^{2\ell_3}\frac{\Gamma((k+1)\alpha_-^2-1)}{\Gamma(k\alpha_-^2)^2\Gamma(-k\alpha_-^2)}}. \tag{E.7}
\end{aligned}$$

In the above expression, the standard factorials $(\ell_1+\ell_2-\ell_3)!$, $(2\ell_1)!$, $(2\ell_2)!$ and $(2\ell_3)!$ can be absorbed into the Gamma functions. Finally, we get

$$\begin{aligned}
C_{(1,2\ell_1+1),(1,2\ell_2+1),(1,2\ell_3+1)} &= \frac{1}{[2\ell_3]!}\sqrt{\frac{[\ell_1-\ell_2+\ell_3]!\,[\ell_2-\ell_1+\ell_3]!\,[\ell_1+\ell_2+\ell_3+1]!}{\left([2\ell_1+1]_q\,[2\ell_2+1]_q\,[2\ell_3+1]_q\right)^{1/2}[\ell_1+\ell_2-\ell_3]!}} \\
&\quad \times \prod_{k=1}^{\ell_1+\ell_2-\ell_3}\frac{\Gamma\big((\ell_1+\ell_2+\ell_3-k+2)\alpha_-^2-1\big)}{\Gamma\big((2\ell_1-k+1)\alpha_-^2\big)\Gamma\big((2\ell_2-k+1)\alpha_-^2\big)\Gamma(1-k\alpha_-^2)}
\end{aligned}$$

$$\times \sqrt{\left(\prod_{k=1}^{2\ell_1} \frac{\Gamma\left(k\alpha_-^2\right)^2 \Gamma(1-k\alpha_-^2)}{\Gamma\left((k+1)\alpha_-^2-1\right)}\right)\left(\prod_{k=1}^{2\ell_2} \frac{\Gamma\left(k\alpha_-^2\right)^2 \Gamma(1-k\alpha_-^2)}{\Gamma\left((k+1)\alpha_-^2-1\right)}\right)}$$
$$\times \sqrt{\prod_{k=1}^{2\ell_3} \frac{\Gamma\left((k+1)\alpha_-^2-1\right)}{\Gamma\left(k\alpha_-^2\right)^2 \Gamma(1-k\alpha_-^2)}}. \tag{E.8}$$

### E.2 Check relation (142)

Let us now verify the relation (142):

$$C^2_{(1,s_1),(1,s_2),(1,s_3)} = C^{\mathrm{MM}}_{(1,s_1),(1,s_2),(1,s_3)}, \tag{E.9}$$

where the minimal-model OPE coefficient $C^{\mathrm{MM}}_{(1,s_1),(1,s_2),(1,s_3)}$ is given in (140).

To see this, it suffices to show that

- (E.9) holds at the level of absolute value for any complex $q$ (as long as the OPE coefficients are finite),

- (E.9) holds at $q = -1$;

#### E.2.1 Check absolute value

In this subsection show that (E.9) holds at least up to some phase factor. Until the end of this subsection, we will not be very careful about the choices of branches of the square root functions, so the conclusion here will hold up to a phase factor. The phase factor will be fixed in the next subsection.

Using the properties of the Gamma function, we have the following identities for $\gamma$:

$$\frac{\gamma\left(s-\alpha^{-2}\right)}{\gamma\left(1-\alpha^{-2}\right)} = \alpha^{-4(s-1)} \prod_{k=1}^{s-1} \left(k\alpha^2-1\right)^2,$$
$$\frac{\gamma\left(-s+\alpha^{-2}\right)}{\gamma\left(-1+\alpha^{-2}\right)} = \alpha^{4(s-1)} \prod_{k=1}^{s-1} \left((k+1)\alpha^2-1\right)^{-2}. \tag{E.10}$$

Using (E.10), some factors in the expression of $C^{\mathrm{MM}}$ (140) are rewritten as follows:

$$\frac{\gamma\left(s_1-\alpha_-^{-2}\right)\gamma\left(s_2-\alpha_-^{-2}\right)\gamma\left(-s_3+\alpha_-^{-2}\right)}{\gamma\left(1-\alpha_-^{-2}\right)} = -\left(\alpha_-^2\right)^{-2(s_1+s_2-s_3-2)}\left[\frac{\prod_{k=1}^{s_1-1}\left(k\alpha_-^2-1\right)\prod_{k=1}^{s_2-1}\left(k\alpha_-^2-1\right)}{\prod_{k=1}^{s_3}\left(k\alpha_-^2-1\right)}\right]^2. \tag{E.11}$$

We also have the following relations:

$$\frac{1}{\gamma\left(-1+s\alpha_-^2\right)} \equiv \frac{\Gamma\left(2-s\alpha_-^2\right)}{\Gamma\left(-1+s\alpha_-^2\right)} = -\frac{\Gamma\left(\alpha_-^2\right)\Gamma\left(1-\alpha_-^2\right)}{\Gamma\left(-1+s\alpha_-^2\right)^2 [s]_q}, \tag{E.12}$$

and

$$\frac{1}{\gamma\left(1-s\alpha_-^2\right)} \equiv \frac{\Gamma\left(s\alpha_-^2\right)}{\Gamma\left(1-s\alpha_-^2\right)} = \frac{\Gamma\left(1-s\alpha_-^2\right)^2 [s]_q}{\Gamma\left(\alpha_-^2\right)\Gamma\left(1-\alpha_-^2\right)}. \tag{E.13}$$

Using eqs. (E.11), (E.12), and (E.13), the square-root factor in $C^{\mathrm{MM}}_{(1,s_1),(1,s_2),(1,s_3)}$ can be expressed as:

$$\frac{\gamma(\alpha_-^2-1)\gamma(s_1-\alpha_-^{-2})\gamma(s_2-\alpha_-^{-2})\gamma(-s_3+\alpha_-^{-2})}{\gamma(1-\alpha_-^{-2})\gamma(-1+s_1\alpha_-^2)\gamma(-1+s_2\alpha_-^2)\gamma(1-s_3\alpha_-^2)} \tag{E.14}$$

$$= \frac{[s_3]_q}{[s_1]_q[s_2]_q}\left(\alpha_-^2\right)^{-2(s_1+s_2-s_3-2)}\left[\frac{\Gamma(\alpha_-^2-1)\Gamma(s_3\alpha_-^2-1)}{\Gamma(s_1\alpha_-^2-1)\Gamma(s_2\alpha_-^2-1)}\frac{\prod_{k=1}^{s_1-1}(k\alpha_-^2-1)\prod_{k=1}^{s_2-1}(k\alpha_-^2-1)}{\prod_{k=1}^{s_3-1}(k\alpha_-^2-1)}\right]^2.$$

The factors in the last line of (140) can be rewritten as products of Gamma functions and $q$-numbers as follows:

$$\gamma(k\alpha_-^2)\gamma(-1+(k+s_3)\alpha_-^2)\gamma(1+(k-s_1)\alpha_-^2)\gamma(1+(k-s_2)\alpha_-^2)$$

$$= -\frac{[s_3+k]_q}{[k]_q[s_1-k]_q[s_2-k]_q}\left(\frac{\Gamma(\alpha_-^2)\Gamma(1-\alpha_-^2)\Gamma((s_3+k)\alpha_-^2-1)}{\Gamma(1-k\alpha_-^2)\Gamma((s_1-k)\alpha_-^2)\Gamma((s_2-k)\alpha_-^2)}\right)^2. \tag{E.15}$$

Recalling that $s_i = 2\ell_i + 1$, and taking the product over $k$, we get:

$$\prod_{k=1}^{\ell_1+\ell_2-\ell_3}\gamma(k\alpha_-^2)\gamma(-1+(k+s_3)\alpha_-^2)\gamma(1+(k-s_1)\alpha_-^2)\gamma(1+(k-s_2)\alpha_-^2)$$

$$= (-1)^{\ell_1+\ell_2-\ell_3}\frac{[\ell_1-\ell_2+\ell_3]![\ell_2-\ell_1+\ell_3]![\ell_1+\ell_2+\ell_3+1]!}{[\ell_1+\ell_2-\ell_3]![2\ell_1]![2\ell_2]![2\ell_3+1]!} \tag{E.16}$$

$$\times \prod_{k=1}^{\ell_1+\ell_2-\ell_3}\left(\frac{\Gamma(\alpha_-^2)\Gamma(1-\alpha_-^2)\Gamma((2\ell_3+1+k)\alpha_-^2-1)}{\Gamma(1-k\alpha_-^2)\Gamma((2\ell_1+1-k)\alpha_-^2)\Gamma((2\ell_2+1-k)\alpha_-^2)}\right)^2.$$

Substituting (E.14) and (E.16) into (140), we obtain the following equivalent expression for $C^{\mathrm{MM}}_{(1,s_1),(1,s_2),(1,s_3)}$:

$$C^{\mathrm{MM}}_{(1,s_1),(1,s_2),(1,s_3)} = (-1)^{\ell_1+\ell_2-\ell_3}\frac{[\ell_1-\ell_2+\ell_3]![\ell_2-\ell_1+\ell_3]![\ell_1+\ell_2+\ell_3+1]!}{[\ell_1+\ell_2-\ell_3]![2\ell_1]![2\ell_2]![2\ell_3+1]!}$$

$$\times \sqrt{\frac{[2\ell_3+1]_q}{[2\ell_1+1]_q[2\ell_2+1]_q}}$$

$$\times \prod_{k=1}^{\ell_1+\ell_2-\ell_3}\left(\frac{\Gamma(\alpha_-^2)\Gamma(1-\alpha_-^2)\Gamma((2\ell_3+1+k)\alpha_-^2-1)}{\Gamma(1-k\alpha_-^2)\Gamma((2\ell_1+1-k)\alpha_-^2)\Gamma((2\ell_2+1-k)\alpha_-^2)}\right)^2 \tag{E.17}$$

$$\times \frac{\Gamma(\alpha_-^2-1)\Gamma(s_3\alpha_-^2-1)}{\Gamma(s_1\alpha_-^2-1)\Gamma(s_2\alpha_-^2-1)}\frac{\prod_{k=1}^{2\ell_1}(k\alpha_-^2-1)\prod_{k=1}^{2\ell_2}(k\alpha_-^2-1)}{\prod_{k=1}^{2\ell_3}(k\alpha_-^2-1)}.$$

The expression (E.17) looks almost the same as $\left(C_{(1,2\ell_1+1),(1,2\ell_2+1),(1,2\ell_3+1)}\right)^2$ given in (138).

After eliminating the common factors of them, it remains to show the following identity:

$$
\frac{1}{[2\ell_3]!} \left( \prod_{k=1}^{2\ell_1} \frac{\Gamma\left(k\alpha_-^2\right)^2 \Gamma(1-k\alpha_-^2)}{\Gamma\left((k+1)\alpha_-^2-1\right)} \right) \left( \prod_{k=1}^{2\ell_2} \frac{\Gamma\left(k\alpha_-^2\right)^2 \Gamma(1-k\alpha_-^2)}{\Gamma\left((k+1)\alpha_-^2-1\right)} \right) \left( \prod_{k=1}^{2\ell_3} \frac{\Gamma\left((k+1)\alpha_-^2-1\right)}{\Gamma\left(k\alpha_-^2\right)^2 \Gamma(1-k\alpha_-^2)} \right)
$$

$$
= \frac{1}{[2\ell_1]! \, [2\ell_2]!} \left( \Gamma\left(\alpha_-^2\right) \Gamma\left(1-\alpha_-^2\right) \right)^{2(\ell_1+\ell_2-\ell_3)} \frac{\Gamma\left(\alpha_-^2-1\right) \Gamma\left((2\ell_3+1)\alpha_-^2-1\right)}{\Gamma\left((2\ell_1+1)\alpha_-^2-1\right) \Gamma\left((2\ell_2+1)\alpha_-^2-1\right)}
$$

$$
\times \prod_{k=1}^{2\ell_1} \left(k\alpha_-^2-1\right) \prod_{k=1}^{2\ell_2} \left(k\alpha_-^2-1\right) \prod_{k=1}^{2\ell_3} \left(k\alpha_-^2-1\right)^{-1} \times e^{i\theta(\ell_1,\ell_2,\ell_3)}, \tag{E.18}
$$

where $e^{i\theta(\ell_1,\ell_2,\ell_3)}$ is the phase factor which we do not care about in this subsection. To see (E.18), we simplify the LHS as follows

$$
\prod_{k=1}^{2\ell} \frac{\Gamma\left(k\alpha_-^2\right)^2 \Gamma(1-k\alpha_-^2)}{\Gamma\left((k+1)\alpha_-^2-1\right)} = \prod_{k=1}^{2\ell} \frac{\Gamma\left(k\alpha_-^2\right) \Gamma\left(\alpha_-^2\right) \Gamma(1-\alpha_-^2)}{\Gamma\left((k+1)\alpha_-^2-1\right) [k]_q}
$$

$$
= \frac{\left(\Gamma\left(\alpha_-^2\right) \Gamma(1-\alpha_-^2)\right)^{2\ell} \Gamma\left(\alpha_-^2-1\right)}{[2\ell]! \, \Gamma\left((2\ell+1)\alpha_-^2-1\right)} \prod_{k=1}^{2\ell} \frac{\Gamma\left(k\alpha_-^2\right)}{\Gamma\left(k\alpha_-^2-1\right)} \tag{E.19}
$$

$$
= \frac{\left(\Gamma\left(\alpha_-^2\right) \Gamma(1-\alpha_-^2)\right)^{2\ell} \Gamma\left(\alpha_-^2-1\right)}{[2\ell]! \, \Gamma\left((2\ell+1)\alpha_-^2-1\right)} \prod_{k=1}^{2\ell} \left(k\alpha_-^2-1\right).
$$

With this, the identity (E.18) becomes evident with $\theta(\ell_1,\ell_2,\ell_3)=0$, completing the derivation of the relation (E.9) at the level of absolute values.

### E.2.2  Check relation (E.9) at $q = -1$

The previous subsection shows that relation (E.9) holds in absolute values. There is the potential possibility that two sides of (E.9) may differ by a phase factor.

Since both sides of (E.9) are locally analytic in $q$ (or equivalently, $\alpha_-^2$), their ratio, i.e. the phase factor must be a constant. Therefore, it remains to show that (E.9) holds at $q = -1$. In this subsection, we will perform this analysis by sticking to the convention we chose in section 3.6 and analytically continue all the quantities involved to $q = -1$. Our goal is to show the following claim:

- Under the convention of section 3.6, at $q = -1$, both $C_{(1,s_1),(1,s_2),(1,s_3)}$ and $C^{\mathrm{MM}}_{(1,s_1),(1,s_2),(1,s_3)}$ are real and positive.

This claim and the fact that $\left|C_{(1,s_1),(1,s_2),(1,s_3)}\right|^2 = \left|C^{\mathrm{MM}}_{(1,s_1),(1,s_2),(1,s_3)}\right|$ imply the relation (E.9).

Let us first compute $C_{(1,s_1),(1,s_2),(1,s_3)}$ at $q = -1$. Our starting point is again eq. (E.5). We would like to compute the quantities in the right-hand side of (E.5) individually and combine them together in the end.

The normalization factor $\mathcal{N}_\ell$ at $q = -1$ has already been discussed in section 3.6 and is given in eq. (133):

$$
\mathcal{N}_\ell\big|_{q=-1} = (2\pi)^\ell (2\ell+1)^{1/4}. \tag{E.20}
$$

The OPE coefficient in the Coulomb gas normalization, $\widetilde{C}^{(\ell_3,\ell_3)}_{(\ell_1,\ell_3-\ell_2),(\ell_2,\ell_2)}$, is single valued by its

explicit expression (122). We just need to evaluate it at $\alpha_-^2 = 1$, which gives

$$\left.\widetilde{C}^{(\ell_3,\ell_3)}_{(\ell_1,\ell_3-\ell_2),(\ell_2,\ell_2)}\right|_{q=-1} = (-1)^{\frac{1}{2}(\ell_1+\ell_2-\ell_3)(\ell_1+\ell_2-\ell_3-1)}(2\pi i)^{\ell_1+\ell_2-\ell_3}(\ell_1+\ell_2-\ell_3)!$$
$$\times \prod_{k=1}^{\ell_1+\ell_2-\ell_3} \frac{\Gamma(\ell_1+\ell_2+\ell_3-k+1)\,\Gamma(k)}{\Gamma(2\ell_1-k+1)\,\Gamma(2\ell_2-k+1)}. \tag{E.21}$$

The quantity $f_m^\ell$ involves $q$-numbers. By (4) and the convention (131), we get

$$\left.f_m^\ell\right|_{q=-1} = -i(-1)^{\ell-m}\sqrt{(\ell+m)(\ell-m+1)}. \tag{E.22}$$

The Clebsch-Gordan coefficient in (E.5) is given in (E.6). Using our convention of the $q$-numbers, we get

$$\left.\begin{bmatrix} \ell_1 & \ell_2 & \ell_3 \\ \ell_3-\ell_2 & \ell_2 & \ell_3 \end{bmatrix}\right|_{q=-1} = (-1)^{\ell_1+\ell_2-\ell_3}\sqrt{\frac{(2\ell_2)!\,(2\ell_3+1)!}{(\ell_2+\ell_3-\ell_1)!\,(\ell_1+\ell_2+\ell_3+1)!}}. \tag{E.23}$$

Plugging the above expressions into (E.5), all the minus signs and $i$'s cancel out, and we get

$$\left.C_{(1,2\ell_1+1),(1,2\ell_2+1),(1,2\ell_3+1)}\right|_{q=-1} = \sqrt{\frac{(\ell_1+\ell_2-\ell_3)!(\ell_1+\ell_3-\ell_2)!(\ell_2+\ell_3-\ell_1)!(\ell_1+\ell_2+\ell_3+1)!}{(2\ell_1)!(2\ell_2)!(2\ell_3)!\sqrt{(2\ell_1+1)(2\ell_2+1)(2\ell_3+1)}}}$$
$$\times \prod_{k=1}^{\ell_1+\ell_2-\ell_3} \frac{\Gamma(k)\Gamma(\ell_1+\ell_2+\ell_3-k+1)}{\Gamma(2\ell_1-k+1)\Gamma(2\ell_2-k+1)}. \tag{E.24}$$

This expression shows the positivity of $C_{(1,s_1),(1,s_2),(1,s_3)}$ at $q=-1$.

Next, let us compute $C^{\text{MM}}_{(1,s_1),(1,s_2),(1,s_3)}$ at $q=-1$ using (140). As described around (140), $C^{\text{MM}}_{(1,s_1),(1,s_2),(1,s_3)}$ involves square root functions and we choose the principal branch of the square root at $\alpha_-^2 = 1$.

Evaluating the function inside the square root at $\alpha_-^2 = 1$, we get

$$\lim_{\alpha_-^2 \to 1} \frac{\gamma(\alpha_-^2-1)\gamma(s_1-\alpha_-^{-2})\gamma(s_2-\alpha_-^{-2})\gamma(-s_3+\alpha_-^{-2})}{\gamma(1-\alpha_-^{-2})\gamma(-1+s_1\alpha_-^2)\gamma(-1+s_2\alpha_-^2)\gamma(1-s_3\alpha_-^2)} = \frac{s_3}{s_1 s_2}. \tag{E.25}$$

We see that it is positive, so its square root (in the principal branch) is also positive.

Then we evaluate the factor in the product:

$$\gamma(k\alpha_-^2)\gamma\left(-1+(k+s_3)\alpha_-^2\right)\gamma\left(1+(k-s_1)\alpha_-^2\right)\gamma\left(1+(k-s_2)\alpha_-^2\right).$$

At $\alpha_-^2 = 1$, we get

$$\lim_{\alpha_-^2 \to 1} \gamma(k\alpha_-^2)\gamma\left(-1+(k+s_3)\alpha_-^2\right)\gamma\left(1+(k-s_1)\alpha_-^2\right)\gamma\left(1+(k-s_2)\alpha_-^2\right)$$
$$= \frac{k!(k-1)!(s_3+k-2)!(s_3+k)!}{(s_3+k-1)(s_1-k-1)!(s_1-k)!(s_2-k-1)!(s_2-k)!}, \tag{E.26}$$

which is also positive in the allowed range of $k$.

Therefore, $C^{\text{MM}}_{(1,s_1),(1,s_2),(1,s_3)}$ is positive at $\alpha_-^2 = 1$:

$$\left.C^{\text{MM}}_{(1,s_1),(1,s_2),(1,s_3)}\right|_{\alpha_-^2=1} = \sqrt{\frac{s_3}{s_1 s_2}} \prod_{k=1}^{\frac{s_1+s_2-s_3-1}{2}} \frac{k!(k-1)!(s_3+k-2)!(s_3+k)!}{(s_3+k-1)(s_1-k-1)!(s_1-k)!(s_2-k-1)!(s_2-k)!}. \tag{E.27}$$

This finishes the derivation of the relation (E.9).

### E.3 General OPE coefficients $C_{(r_1,s_1)(r_2,s_2)(r_3,s_3)}$

#### E.3.1 Fixing the phase ambiguity

This section provides details on how we fix the ambiguities of the quantities related to the general OPE coefficients $C_{(r_1,s_1),(r_2,s_2),(r_3,s_3)}$. Since the ambiguities come from the multi-valuedness of the square-root functions, it suffices to fix them at

$$c = 1, \quad \text{i.e.} \quad \alpha_- = -1, \quad \alpha_+ = 1. \tag{E.28}$$

At this point, $q_+$ and $q_-$ coincide: $q_\pm \equiv e^{i\pi\alpha_\pm^2} = -1$.

Let us first fix the normalization factor $\mathcal{N}_\ell$ in (173). As demonstrated in section 3.6, we let $\mathcal{N}_{\ell_-}$ be positive at $q_- = -1$. Using the same rule, we also let $\mathcal{N}_{\ell_+}$ be positive there. For taking the square root of the extra factors in (173), we always take

$$\sqrt{(-i)^{4\ell^+\ell^-}} = e^{-i\pi\ell^+\ell^-}, \tag{E.29}$$

and the principal branch for the ratio of the $Y$-functions near the $c = 1$ point. It happens to be true that the ratio of the $Y$-functions in (173) is positive at $c = 1$:

$$\left.\frac{Y(0,2\ell)}{Y(1,2\ell)}\right|_{\alpha_\pm = \mp 1} = \lim_{\alpha_+^2 \to 1} \prod_{k^+=1}^{2\ell^+} \prod_{k^-=1}^{2\ell^-} \frac{k^+\alpha_+^2 - k^-}{(k^++1)\alpha_+^2 - (k^-+1)}. \tag{E.30}$$

Here we have used the fact that

$$\lim_{\alpha_+^2 \to 1} \frac{k^+\alpha_+^2 - k^-}{(k^++1)\alpha_+^2 - (k^-+1)} = \begin{cases} \frac{k^+ - k^-}{(k^++1)-(k^-+1)} > 0, & (k^+ \neq k^-), \\ \frac{k^+}{k^++1} > 0, & (k^+ = k^-). \end{cases} \tag{E.31}$$

This fixes the normalization factor:

- At $c = 1$, $\mathcal{N}_\ell = e^{-i\pi\ell^+\ell^-} \times$ (positive factors).

Under this normalization, the OPE coefficient $C_{(r_1,s_1),(r_2,s_2),(r_3,s_3)}$ is given by

$$C_{(r_1,s_1)(r_2,s_2)(r_3,s_3)} = C_{(r_1,1)(r_2,1)(r_3,1)} C_{(1,s_1)(1,s_2)(1,s_3)} \sqrt{\frac{Y(0,2\ell_1)Y(0,2\ell_2)Y(0,2\ell_3)}{Y(1,2\ell_1)Y(1,2\ell_2)Y(1,2\ell_3)}} \tag{E.32}$$

$$\times e^{i\pi(\ell_1^+\ell_1^- + \ell_2^+\ell_2^- - \ell_3^+\ell_3^- - 2\ell_1^+\ell_2^-)} (-1)^{(\ell_1^+ + \ell_3^+ - \ell_2^+)(\ell_2^- + \ell_3^- - \ell_1^-)}$$

$$\times \frac{Y(1,2\ell_1)Y(1,2\ell_2)Y(1,2\ell_3)}{Y(0,\ell_1-\ell_2+\ell_3)Y(0,\ell_2-\ell_1+\ell_3)Y(0,\ell_1+\ell_2-\ell_3)Y(1,\ell_1+\ell_2+\ell_3)},$$

which is the same as (175).

#### E.3.2 Compare to OPE coefficients of generalized minimal models

The OPE coefficients of the generalized minimal model are given in [5, 15]:

$$C^{\mathrm{MM}}_{(r_1,s_1),(r_2,s_2),(r_3,s_3)} = \rho^{4st+2t-2s-1} \sqrt{\frac{\gamma(\rho-1)\gamma(s_1-r_1\rho^{-1})\gamma(s_2-r_2\rho^{-1})\gamma(-s_3+r_3\rho^{-1})}{\gamma(1-\rho^{-1})\gamma(-r_1+s_1\rho)\gamma(-r_2+s_2\rho)\gamma(r_3-s_3\rho)}}$$

$$\times \prod_{i=1}^{r} \prod_{j=1}^{s} [(i-j\rho)(i-r_1-(j-s_1)\rho)(i-r_2-(j-s_2)\rho)(i+r_3-(j+s_3)\rho)]^{-2}$$

$$\times \prod_{i=1}^{r} \gamma(i\rho^{-1})\gamma(s_1+(i-r_1)\rho^{-1})\gamma(s_2+(i-r_2)\rho^{-1})\gamma(-s_3+(i+r_3)\rho^{-1})$$

$$\times \prod_{j=1}^{s} \gamma(j\rho)\gamma(r_1+(j-s_1)\rho)\gamma(r_2+(j-s_2)\rho)\gamma(-r_3+(j+s_3)\rho), \tag{E.33}$$

where

$$\gamma(x) \equiv \frac{\Gamma(x)}{\Gamma(1-x)}, \qquad \rho = \alpha_-^2, \qquad r = \frac{r_1 + r_2 - r_3 - 1}{2}, \qquad s = \frac{s_1 + s_2 - s_3 - 1}{2}. \quad \text{(E.34)}$$

Now we would like to check the second identity of (E.2). The argument in section E.2 already shows that the identity holds for the $(1,s)$ subsector. Since the OPE coefficients of the $(r, 1)$ sector are computed in a similar way, the identity holds for the $(r, 1)$ subsector for the same reason. So we have

$$C^2_{(r_1,1),(r_2,1),(r_3,1)} = C^{\text{MM}}_{(r_1,1),(r_2,1),(r_3,1)}, \qquad C^2_{(1,s_1),(1,s_2),(1,s_3)} = C^{\text{MM}}_{(1,s_1),(1,s_2),(1,s_3)}. \quad \text{(E.35)}$$

Therefore, to show the second identity of (E.2), it suffices to show the following identity

$$\frac{C^{\text{MM}}_{(r_1,s_1),(r_2,s_2),(r_3,s_3)}}{C^{\text{MM}}_{(r_1,1),(r_2,1),(r_3,1)} C^{\text{MM}}_{(1,s_1),(1,s_2),(1,s_3)}} = \frac{C_{(r_1,s_1),(r_2,s_2),(r_3,s_3)} C_{(r_2,s_2),(r_1,s_1),(r_3,s_3)}}{C^2_{(r_1,1),(r_2,1),(r_3,1)} C^2_{(1,s_1),(1,s_2),(1,s_3)}}. \quad \text{(E.36)}$$

Let us first look at the RHS of (E.36). By (E.32) and the fact that the OPE coefficients in the $(r, 1)$ and $(1,s)$ sectors are permutation symmetric, we have

$$\text{RHS of (E.36)} = \frac{Y(0, 2\ell_1) Y(0, 2\ell_2) Y(0, 2\ell_3) Y(1, 2\ell_1) Y(1, 2\ell_2) Y(1, 2\ell_3)}{Y(0, \ell_1 - \ell_2 + \ell_3)^2 Y(0, \ell_2 - \ell_1 + \ell_3)^2 Y(0, \ell_1 + \ell_2 - \ell_3)^2 Y(1, \ell_1 + \ell_2 + \ell_3)^2}. \quad \text{(E.37)}$$

Here we also used the fact that the allowed values of $\ell_1^+ + \ell_2^+ + \ell_3^+$ and $\ell_1^- + \ell_2^- + \ell_3^-$ must be integers, so that the phase factors in (E.32) cancel out in the RHS of (E.36). The final result is positive for real $c$ around $c = 1$.

Now we would like to show that the LHS of (E.36) is equal to the expression in (E.37). Using (E.33), we have

$$\text{LHS of (E.36)} = \rho^{4st} \sqrt{\frac{\gamma(s_1 - r_1\rho^{-1})\gamma(s_2 - r_2\rho^{-1})\gamma(-s_3 + r_3\rho^{-1})}{\gamma(1 - r_1\rho^{-1})\gamma(1 - r_2\rho^{-1})\gamma(-1 + r_3\rho^{-1})}}$$

$$\times \sqrt{\frac{\gamma(-1 + s_1\rho)\gamma(-1 + s_2\rho)\gamma(1 - s_3\rho)}{\gamma(-r_1 + s_1\rho)\gamma(-r_2 + s_2\rho)\gamma(r_3 - s_3\rho)}}$$

$$\times \sqrt{\frac{\gamma(-r_1 + \rho)\gamma(-r_2 + \rho)\gamma(r_3 - \rho)}{\gamma(-1 + \rho)\gamma(-1 + \rho)\gamma(1 - \rho)}} \times \sqrt{\frac{\gamma(1 - \rho^{-1})\gamma(1 - \rho^{-1})\gamma(-1 + \rho^{-1})}{\gamma(s_1 - \rho^{-1})\gamma(s_2 - \rho^{-1})\gamma(-s_3 + \rho^{-1})}}$$

$$\times \prod_{i=1}^{r} \prod_{j=1}^{s} ((i - j\rho)(i + r_3 - (j + s_3)\rho) (i - r_1 - (j - s_1)\rho)(i - r_2 - (j - s_2)\rho))^{-2}$$

$$\times \prod_{i=1}^{r} \frac{\gamma(-s_3 + (i + r_3)\rho^{-1})\gamma(s_1 + (i - r_1)\rho^{-1})\gamma(s_2 + (i - r_2)\rho^{-1})}{\gamma(-1 + (i + r_3)\rho^{-1})\gamma(1 + (i - r_1)\rho^{-1})\gamma(1 + (i - r_2)\rho^{-1})}$$

$$\times \prod_{j=1}^{s} \frac{\gamma(-r_3 + (j + s_3)\rho)\gamma(r_1 + (j - s_1)\rho)\gamma(r_2 + (j - s_2)\rho)}{\gamma(-1 + (j + s_3)\rho)\gamma(1 + (j - s_1)\rho)\gamma(1 + (j - s_2)\rho)}. \quad \text{(E.38)}$$

To simplify (E.38), we use the identity

$$\frac{\gamma(n + a)}{\gamma(1 + a)} = (-1)^{n-1} \prod_{k=1}^{n-1} (k + a)^2, \qquad \frac{\gamma(-1 + a)}{\gamma(-n + a)} = (-1)^{n-1} \prod_{k=2}^{n} (k - a)^2, \qquad (n \geq 1), \quad \text{(E.39)}$$

which shows that all the factors in (E.38) can be rewritten as products of form

$$-k\rho^{-1/2} + l\rho^{1/2} = k\alpha_+ + l\alpha_-.$$

The prefactor $\rho^{4st}$ happens to cancel the total extra power of $\rho$. Then one can easily verify that (E.38) is the reshuffling of the product in (E.37), so they are equal. This finishes the check of (E.36). Note that in this argument, we do not need to worry about "±" ambiguity from the square-root function because both sides of (E.36) are already fixed to be positive at $c = 1$.

Therefore, we conclude that the second identity of (E.2) holds for any $c$.

### E.4  OPE coefficients in the XXZ$_q$ CFT

Now that the chiral OPE coefficients $C_{(r_1,s_1),(r_2,s_2),(r_3,s_3)}$ have been computed, using the first relation of (E.2), we get the final expression of the OPE coefficients in the XXZ$_q$ CFT:

$$
\begin{aligned}
C^{\text{XXZ}}_{(r_1,s_1),(r_2,s_2),(r_3,s_3)} = {}& i^{2\ell_1^+\ell_1^-+2\ell_2^+\ell_2^--2\ell_3^+\ell_3^-}\sqrt{\frac{Y(0,2\ell_1)Y(0,2\ell_2)Y(0,2\ell_3)}{Y(1,2\ell_1)Y(1,2\ell_2)Y(1,2\ell_3)}}\\[4pt]
&\times \frac{(-1)^{(\ell_1^++\ell_3^+-\ell_2^+)(\ell_2^-+\ell_3^--\ell_1^-)-2\ell_1^+\ell_2^-}\,Y(1,2\ell_1)Y(1,2\ell_2)Y(1,2\ell_3)}{Y(0,\ell_1-\ell_2+\ell_3)Y(0,\ell_2-\ell_1+\ell_3)Y(0,\ell_1+\ell_2-\ell_3)Y(1,\ell_1+\ell_2+\ell_3)}\\[4pt]
&\times \frac{1}{[2\ell_3^-]_q!}\sqrt{\frac{[\ell_1^--\ell_2^-+\ell_3^-]_q!\,[\ell_2^--\ell_1^-+\ell_3^-]_q!\,[\ell_1^-+\ell_2^-+\ell_3^-+1]_q!}{\left([2\ell_1^-+1]_q[2\ell_2^-+1]_q[2\ell_3^-+1]_q\right)^{1/2}[\ell_1^-+\ell_2^--\ell_3^-]_q!}}\\[4pt]
&\times \prod_{k=1}^{\ell_1^-+\ell_2^--\ell_3^-}\frac{\Gamma\!\left((\ell_1^-+\ell_2^-+\ell_3^--k+2)\alpha_-^2-1\right)}{\Gamma\!\left((2\ell_1^--k+1)\alpha_-^2\right)\Gamma\!\left((2\ell_2^--k+1)\alpha_-^2\right)\Gamma(1-k\alpha_-^2)}\\[4pt]
&\times \sqrt{\left(\prod_{k=1}^{2\ell_1^-}\frac{\Gamma\!\left(k\alpha_-^2\right)^2\Gamma(1-k\alpha_-^2)}{\Gamma\!\left((k+1)\alpha_-^2-1\right)}\right)\left(\prod_{k=1}^{2\ell_2^-}\frac{\Gamma\!\left(k\alpha_-^2\right)^2\Gamma(1-k\alpha_-^2)}{\Gamma\!\left((k+1)\alpha_-^2-1\right)}\right)}\\[4pt]
&\times \sqrt{\prod_{k=1}^{2\ell_3^-}\frac{\Gamma\!\left((k+1)\alpha_-^2-1\right)}{\Gamma\!\left(k\alpha_-^2\right)^2\Gamma(1-k\alpha_-^2)}}\,.\\[4pt]
&\times \frac{1}{\left([2\ell_3^+]_{\tilde q}!\right)^2}\frac{[\ell_1^+-\ell_2^++\ell_3^+]_{\tilde q}!\,[\ell_2^+-\ell_1^++\ell_3^+]_{\tilde q}!\,[\ell_1^++\ell_2^++\ell_3^++1]_{\tilde q}!}{\left([2\ell_1^++1]_{\tilde q}[2\ell_2^++1]_{\tilde q}[2\ell_3^++1]_{\tilde q}\right)^{1/2}[\ell_1^++\ell_2^+-\ell_3^+]_{\tilde q}!}\\[4pt]
&\times \prod_{k=1}^{\ell_1^++\ell_2^+-\ell_3^+}\left(\frac{\Gamma\!\left((\ell_1^++\ell_2^++\ell_3^+-k+2)\alpha_+^2-1\right)}{\Gamma\!\left((2\ell_1^+-k+1)\alpha_+^2\right)\Gamma\!\left((2\ell_2^+-k+1)\alpha_+^2\right)\Gamma(1-k\alpha_+^2)}\right)^2\\[4pt]
&\times \left(\prod_{k=1}^{2\ell_1^+}\frac{\Gamma\!\left(k\alpha_+^2\right)^2\Gamma(1-k\alpha_+^2)}{\Gamma\!\left((k+1)\alpha_+^2-1\right)}\right)\left(\prod_{k=1}^{2\ell_2^+}\frac{\Gamma\!\left(k\alpha_+^2\right)^2\Gamma(1-k\alpha_+^2)}{\Gamma\!\left((k+1)\alpha_+^2-1\right)}\right)\\[4pt]
&\times \prod_{k=1}^{2\ell_3^+}\frac{\Gamma\!\left((k+1)\alpha_+^2-1\right)}{\Gamma\!\left(k\alpha_+^2\right)^2\Gamma(1-k\alpha_+^2)}\,.
\end{aligned}
\tag{E.40}
$$

Here, the function $Y$ is defined in (166), and other quantities are given by

$$
\begin{aligned}
& r_i = 2\ell_i^+ + 1\,, && s_i = 2\ell_i^- + 1\,, && \ell_i = (\ell_i^+, \ell_i^-)\,,\\
& \alpha_-^2 = 1/\alpha_+^2 = \frac{\mu}{\mu+1}\,, && q = e^{i\pi\frac{\mu}{\mu+1}}\,, && \tilde q = e^{i\pi\frac{\mu+1}{\mu}}\,.
\end{aligned}
\tag{E.41}
$$

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
