# Peer review of "Quantum Groups as Global Symmetries II. Coulomb Gas Construction"

_SciPost Physics, doi:SciPost Phys. 19, 070 (2025)_

## Round 3 · Referee Report · Anonymous (Referee 1) · 2025-5-27

Strengths

  1. Calculations are very clear and succinct.
  2. Motivation is clear.

Weaknesses

  1. Slightly missing is some more discussion about the relation to the generalized symmetries program. This is mentioned in passing in some footnotes and in brief in the conclusions but there must be closer relations, particularly to noninvertible symmetries.
  2. Also slightly missing is a more detailed comparison to existing results and computations using similar methods in older literature. The authors note that this is not a new subject but mention only some results, and it could be a bit clearer where their methods diverge from previous studies.

Report

This paper discusses a specific example of a CFT with a quantum group symmetry, and analyzes it from a complimentary point of view using the Coulomb gas formalism. Several results that are expected from a quantum group analysis are obtained independently, including some results which the quantum group is not enough to fix.

The paper is interesting and well written. It serves as a good companion paper to the other paper by these authors, with clear computations which corroborate their results. I recommend it for publication.

Recommendation

Publish (easily meets expectations and criteria for this Journal; among top 50%)

---

## Round 3 · Referee Report · Connor Behan (Referee 2) · 2025-6-1

Report

This paper, the second in a series, fits broadly within the generalized symmetry program which aims to exploit symmetry structures other than finite and finite-dimensional groups in QFT. The focus here is on quantum groups, a certain infinite-dimensional structure which can be written in terms of finitely many generators by passing to the universal enveloping algebra. The first paper in this series developed general tools such as Ward identities and bootstrap equations which are useful for studying CFTs that have quantum groups as genuine internal symmetries acting on the Hilbert space. Previously, most works about quantum groups in QFT encountered them as auxiliary objects.

In order to check consistency of this toolkit, the present paper solves a particular CFT in two dimensions which has $U_q(sl_2)$ symmetry. It is defined as the critical point of the $XXZ_q$ spin chain in the continuum limit. Although the $XXZ_q$ spin chain is integrable, it is quite non-trivial compared to the ordinary $XXZ$ spin chain, which flows to a free boson CFT. To compute correlation functions, the authors exploit the fact that all Verma modules in this theory are degenerate and can therefore be given a Coulomb gas representation. The well known Coulomb gas formalisms which compute correlators in generalized minimal models and critical Potts models are both inadequate for this purpose so significant time is spent developing a new one. The main novelty is that correlation functions obeying quantum group Ward identities are generically multivalued. This is not a contradiction because operators charged under the quantum group are attached to topological lines.

The heart of the paper is section 3 which works extensively with contour integrals of screening charges. In this new approach, the integrals should be interpreted as full correlation functions instead of blocks. As such, there is no notion of "channel" and different choices for the contour control which operators are hit by lowering operators for the quantum group. Nevertheless, Virasoro blocks have a close relation to correlators in this CFT and this leads to one explanation for why $U_q(sl_2)$ 6j symbols appear in the Virasoro crossing kernel. Section 3 builds up to a derivation of chiral OPE coefficients for operators on the border of the Kac table. These initially take an unwieldy form as a sum of many terms where the coefficients are defined implicitly by a recursion relation. While this expression allows various checks to be performed, the authors are able to derive a much more compact expression for the same OPE coefficients. The key property (which is assured by the Ward identities but looks remarkable from the computational point of view) is that the complicated sum becomes independent of $U(1)$ charges once it is normalized in the proper way. All results therefore follow from the simplest $U(1)$ charge configuration which collapses the sum to a single term.

Section 4 essentially repeats this construction four times in order to study correlators in some auxiliary theory which is invariant under four quantum groups. The single quantum group from before is doubled by introducing anti-chiral vertex operators and then doubled again by allowing both Kac indices to be non-trivial. OPE coefficients in this theory take on a factored form up to a cocycle which is fixed by the Ward identities and some gamma functions related to the structure of the Selberg integral. The final step is then an index contraction which eliminates two copies of $U_q(sl_2)$ at a time. Performing it twice leads to generalized minimal models. Performing it once leads to $XXZ_q$ which effectively has only one copy of $U_q(sl_2)$ because its operators are all singlets under the second one. By propagating the coycles through this projection, the authors are able to determine the phases of the $XXZ_q$ OPE coefficients for the first time.

This paper is a great example of how modern methods based on symmetry can streamline old results and solve open problems. Although it is full of technical calculations, they are explained in a self-contained way. Some of the subtleties addressed are crucial for the current work while others relate to potential extensions of it. Overall, it is clear that the authors paid attention to many details and I hope the suggestions below are compatible with that goal.

Requested changes

  1. The introduction mentions the monodromy condition for "local CFTs". Readers used to a slightly different definition would say this theory is also local because it has a stress tensor.

  2. I do not know what "as well as from some additional defects" means at the end of the introduction.

  3. On page 4, it might be best to say that correlation functions of several operators bring about the need to act on a tensor product of this many representations. Saying "two representations (operators)" might lead people to think that an operator carries two representation labels instead of one.

  4. The $[]_q$ notation is used in equation (2.4) without being defined.

  5. Although there is no issue with using (2.14) to encode the spectrum, calling it the "torus partition function" sounds like an abuse of terminology since it is not modular invariant.

  6. Page 10 says that vertex operators with charges $\alpha_{r,s}$ and $\alpha_{-r,-s}$ are equivalent in generalized minimal models but not here. If there is no quick example which demonstrates this, it might be nice to say that it will be made clear by section 4.

  7. Even for generalized minimal models, I think this statement is not always true. If $V_{2\alpha_0} = V_{\alpha_+ + \alpha_-}$ were always equivalent to the identity, it would be possible to insert it arbitrarily many times without changing a correlation function. This would reduce all screening integrals to those with only $S_+$ or only $S_-$. Never needing both sounds too good to be true. Perhaps this is related to the nice discussion around equation (4.10).

  8. Although figure 4 and figure 20 should have the same first diagram, the latter is more clear because it places all topological lines above their integration contours. There is no need to change either figure but it would be worth emphasizing that one can move a line to make it cross a contour. Only lines crossing lines and contours crossing contours are illegal.

  9. A minor point is that page 50 says the dots in figure 20 represent other operator insertions. It looks like they represent other screening charges instead.

  10. Also, under (B.1) it might be better to say the contours are "parallel to the real line". If they were actually "along the real line" then they would intersect.

  11. I think the last diagram on the first line of Figure 21 should either have a minus sign with the arrow shown or a plus sign with the arrow in the opposite direction.

  12. A discussion in Appendix B refers to "a difference between the first and second integrals". But it sounds like the first is actually the second in Figure 22 and vice versa.

  13. When solving for $\Sigma_n$ in this appendix, it might be worth mentioning that this is a calculation Dotsenko and Fateev already did in reference [3] to relate the unordered $J$ integrals to the ordered $I$ integrals.

  14. On the second line of (3.19), the product with $1 \leq r < s \leq \ell$ should say $1 \leq r < s \leq N$.

  15. Although they are understood as living inside a correlation function with many points, section 3.5.1 only refers to two operators directly. The notation $x_k$ and $x_{k + 1}$ for their positions therefore looks needlessly heavy.

  16. A superscript is missing from $q_-^{2m^-_{k + 1}}$ in Figure 19.

  17. The text refers to the first line of equation (4.26) but there is only one line.

  18. The second last line of (D.13) has $x_{2N}$ appearing. This should be replaced by $x_{2N - 1}$.

  19. There seems to be an unwanted space in equation (E.3).

  20. The paper sometimes says "OPE expansion" which is redundant. It is also switches between saying l.h.s./r.h.s. and LHS/RHS.

  21. Typos I found are "reversed engineered" on page 9, "it turns that" on page 20, "operators involves the simplest" on page 20, "sum of two line integral" on page 22, "what phase to the integrand" on page 22, "so it different from" on page 50", "is has to be" on page 53, "we do not care in" on page 63 and "evaluate the function" on page 65.

Recommendation

Ask for minor revision

---

## Round 4 · Author Response

List of changes
Addressing the comments of the referee report 2, we have made the following improvements to the paper:
-
A sentence about the existence of stress-energy tensor and locality of CFT was added in the introduction.
-
A phrase
as well as some additional defect lines'' was replaced with a phraseand topological defect lines coming from free boson CFT'' in the last paragraph of the introduction. The aim of this paragraph is to briefly announce the construction of section 3.2. We hope this would be enough for the introduction. -
Phrasing changed.
-
Notation added.
-
The word ``torus'' referring to partition function was removed.
6-7. We thank the referee for pointing out the subtleties with operators $\mathcal{V}_\alpha$ and $\mathcal{V}_{2\alpha_0-\alpha}$. A paragraph after eq. (3.12) was modified as well as the paragraph in the end of section 3.2.1 was added. The reference to section 4.1 was also added. We hope this discussion is now made more clear.
-
A phrase in the description of figure 4 was added, emphasizing that dashed lines can cross contours.
-
Typo corrected.
-
Phrasing changed.
-
Typo corrected.
-
More explanation of figure 22 was added.
-
Mentioning of reference [3] was added.
-
Typo corrected.
-
We thank the referee for suggestion of changing the notation of $x_k, x_{k+1}$. However, we believe that the notation we chose is consistent, and moreover changing the notation in the entire section at this point can lead to additional typos. We hope the referee agrees to keep the notation as it is.
16-21. Typos corrected.

---

## Round 4 · List of Changes

Addressing the comments of the referee report 2, we have made the following improvements to the paper:
-
A sentence about the existence of stress-energy tensor and locality of CFT was added in the introduction.
-
A phrase
as well as some additional defect lines'' was replaced with a phraseand topological defect lines coming from free boson CFT'' in the last paragraph of the introduction. The aim of this paragraph is to briefly announce the construction of section 3.2. We hope this would be enough for the introduction. -
Phrasing changed.
-
Notation added.
-
The word ``torus'' referring to partition function was removed.
6-7. We thank the referee for pointing out the subtleties with operators $\mathcal{V}_\alpha$ and $\mathcal{V}_{2\alpha_0-\alpha}$. A paragraph after eq. (3.12) was modified as well as the paragraph in the end of section 3.2.1 was added. The reference to section 4.1 was also added. We hope this discussion is now made more clear.
-
A phrase in the description of figure 4 was added, emphasizing that dashed lines can cross contours.
-
Typo corrected.
-
Phrasing changed.
-
Typo corrected.
-
More explanation of figure 22 was added.
-
Mentioning of reference [3] was added.
-
Typo corrected.
-
We thank the referee for suggestion of changing the notation of $x_k, x_{k+1}$. However, we believe that the notation we chose is consistent, and moreover changing the notation in the entire section at this point can lead to additional typos. We hope the referee agrees to keep the notation as it is.
16-21. Typos corrected.

---

## Editorial Decision

published